# A hydrological framework for persistent pools along non-perennial rivers

**Sarah A. Bourke[1], Margaret Shanafield[2], Paul Hedley[3], Sarah Chapman[1,3], Shawan Dogramaci[1,3]**

[1]School of Earth Sciences, University of Western Australia, Crawley WA 6009 Australia

[2]College of Science and Engineering, Flinders University, Bedford Park, SA 5042, Australia

[3]Rio Tinto Iron Ore, Perth, WA 6000 Australia

*Correspondence to*: Sarah A. Bourke (sarah.bourke@uwa.edu.au)

## Abstract

Persistent surface water pools along non-perennial rivers represent an important water resource for plants, animals, and humans. While ecological studies of these features are not uncommon, these are rarely accompanied by a rigorous examination of the hydrological and hydrogeological characteristics that create or support persistent river pools. Here we present an overarching framework for understanding the hydrology of persistent pools. Perched surface water, alluvial water through-flow and groundwater discharge are the key hydraulic mechanisms that control the persistence of pools along river channels. Groundwater discharge can be further categorized into that controlled by a geological contact or barrier, and discharge controlled by topography. Emphasis is put on clearly defining through-flow of alluvial water and the different drivers of groundwater discharge. The suite of regional-scale and pool-scale diagnostic tools available for elucidating these hydraulic mechanisms are summarized and critiqued. Water fluxes to pools supported by through-flow alluvial and groundwater discharge can vary spatially and temporally and quantitatively resolving pool water balance components is commonly non-trivial. This framework allows the evaluation of the susceptibility of persistent pools along river channels to changes in climate or groundwater withdrawals. Finally, we demonstrate the application of

this framework using a suite of the available tools to conduct a regional and pool-scale assessment of the hydrology of persistent river pools in the Hammersley Basin of north-western Australia.

## 1 Introduction

Permanent or almost permanent water features along non-perennial rivers (hereafter referred to as "persistent pools") represent an important water resource for plants, animals, and humans. These persistent pools typically hold residual water from periodic surface flows, but also may receive input from underlying aquifers, and have alternately been termed pools (Bogan and Lytle, 2011; Jaeger and Olden, 2011; John, 1964), springs (Cushing and Wolf, 1984), waterholes (Arthington et al., 2005; Bunn et al., 2006; Davis et al., 2002; Hamilton et al., 2005; Knighton and Nanson, 2000; Rayner et al., 2009), and wetlands (Ashley et al., 2002). Non-perennial streams are globally distributed across all climate types (Shanafield et al., 2021; Messager et al., 2021). The occurrence of persistent pools along non-perennial streams has been well-documented (Bonada et al., 2020), particularly in the arid southwest of the U.S. (Bogan and Lytle, 2011) and across Australia (Arthington et al., 2005; Bunn et al., 2006; Davis et al., 2002).

Several studies have confirmed that persistent river pools support a highly diverse community of flora and fauna (Shepard, 1993; Bonada et al., 2020) and can vary significantly in water quality (Stanley et al., 1997). Persistent pools are also often of cultural significance (Finn and Jackson, 2011; Yu, 2000), providing key connectivity across landscapes for biota (Sheldon et al., 2010; Goodrich et al., 2018), and early hominid migration (Cuthbert et al., 2017). Paradoxically, the unique ecosystems they support are also sensitive to changing climate and human activities (Bunn et al., 2006; Jaeger and Olden, 2011). Persistent pools may dry out naturally after successive dry years (Shanafield et al., 2021) and recent studies have shown that persistent pools are also changing over time in response to alterations in climate and sediment transport (Pearson et al., 2020, Bishop-Taylor et al., 2017). However, their hydrology is typically poorly understood, and the treatment of the hydrology of persistent river pools in published literature to date has been largely descriptive, vague, or tangential to the main theme of the paper

(Thoms and Sheldon, 2000). As a result, effective water resource management is limited by a lack of
understanding of the mechanisms and water sources that support these persistent pools.
By far, the published literature on persistent pools focuses on the ecological processes and patterns.
They have received attention for the role they play as a seasonal refuge (Goodrich et al., 2018), and
with regards to connectivity between riparian ecosystems (Godsey and Kirchner, 2014). For example,
they have been shown to host unique fish assemblages (Arthington et al., 2005; Labbe and Fausch,
2000), macroinvertebrate communities (Bogan and Lytle, 2011), and play a vital role in primary
productivity (Cushing and Wolf, 1984). Recently, it was shown that the structure, but not composition,
of persistent pools mirrors that of perennial rivers (Kelso and Entrekin, 2018). However, rarely are these
ecological studies accompanied by a rigorous examination of the hydrological and hydrogeological
characteristics that provide a setting for these ecologic communities. Although there are isolated studies
that examine the composition of water and propose sources within specific pools (Hamilton et al.,2005;
Fellman et al., 2011), more frequently they simply describe the seasonal persistence of flow and basic
hydrologic parameters (typically temperature and salinity, sometimes also oxygen).
From a hydrogeological perspective, classification of persistent pools, and springs in general, dates back
to the early 20[th] Century, when geological drivers such as faults and interfaces between bedrock and the
overlying alluvial sediments were first discussed in relation to springs (Bryan, 1919; Meinzer, 1927).
Subsequently, a diverse, modern toolbox of hydrologic and hydrogeologic field and analysis methods
to analyse water source, age, and composition has evolved. Yet contemporary work on springs (Alfaro
and Wallace, 1994; Kresic, 2010), and hydrogeology textbooks (e.g. Fetter, 2001; Poeter et al., 2020)
are still based primarily on these early classifications. More recent classifications, moreover, are either
descriptive or focus on the context (karst vs desert) or observable spring water quality (Springer and
Stevens, 2009; Shepard, 1993; Alfaro and Wallace, 1994) and are not readily applied to understand the
hydrology of persistent river pools (not all persistent pools are springs). There has also been a robust
body of literature developed around surface water – groundwater interaction of the past 20 years (e.g.
Stonedahl et al., 2010; Winter et al., 1998), some of which informs our understanding of persistent river
pools, but has not yet been explicitly applied in this context. Similarly, our understanding of the
hydrology of non-perennial streams and their links to groundwater systems continues to expand
(Costigan et al., 2015; Gutiérrez-Jurado et al., 2019; Blackburn et al., 2021; Bourke et al., 2021). The
existence of rain-fed freshwater rock-pools that are not connected to the groundwater system has also
been documented in the context of understanding their ecology (Joque et al., 2010) but the discussion
of their hydrology is limited. Thus, while many of the hydrologic concepts relevant to persistent river
pools can be found in existing literature, a comprehensive hydrologic framework is lacking (Costigan
et al. 2016; Leibowitz et al., 2018). Such a framework should incorporate the relevant elements of
literature on groundwater springs and surface - groundwater interaction, along with the modern suite of
diagnostic tools, to provide a robust platform for understanding the hydraulic mechanisms that support
persistent river pools.
The aim of this paper is to consolidate the hydrologic processes and observational diagnostic tools
within existing literature into a cohesive framework to support the characterization the hydrology of
persistent pools along non-perennial rivers. To this end, we i) identify the range of hydraulic
mechanisms supporting river pool persistence during periods of no-flow and show how these
mechanisms can manifest in the landscape, ii) discuss the resulting susceptibility of pools to changing
climate or groundwater withdrawals and iii) present and critique field-based observational tools
available for identifying these hydraulic mechanisms. The application of this framework is
demonstrated a regional-scale assessment and three pool-scale studies from the Hamersley Basin of
north-western Australia.

**2 Hydraulic mechanisms supporting the persistence of in-stream pools**

Here we propose a framework for classifying the key hydraulic mechanisms that support the persistence
of pools along non-perennial rivers in environments (summarized in Table 1). Geologically, we start by
considering the general case of a non-perennial river along an alluvial channel (inundated and/or
flowing during contemporary flood events) within valley-fill sediments deposited over bedrock, where
tributaries flow either across the bedrock or the valley-fill sediments (Sections 2.1 and 2.2).
We then move onto a discussion of the ways in which geological structures and outcrops can underpin
the persistence of river pools by facilitating the outflow of regional groundwater (Section 2.3). The
range of geological settings for non-perennial streams is vast (Shanafield et al., 2021); we have
endeavoured to provide sufficient general guidance so that the principles can be applied to specific river
systems as required. Hydrologically, we only consider the water balance of residual river pools after
surface flows have ceased. Any water that has infiltrated to the subsurface saturated zone (which may
be a perched aquifer) is considered to be groundwater, irrespective of the residence time of that water
in the subsurface. This groundwater may be alluvial groundwater, stored within the alluvium beneath
and adjacent to the contemporary river, or regional groundwater stored within regional aquifers. The
conceptual diagrams presented in this section are intended as generalized diagrams to represent key
hydrological features; they do not represent specific locations and are not drawn to scale.


Table 1 Summary of hydrological framework for persistent pools

| Mechanism supporting pool persistence and water balance* | Physical characteristics | Hydrochemical characteristics | Susceptibility to stressors |
|---|---|---|---|
| **Perched water**<br><br>$\frac{\partial V}{\partial t} = EA$ | Topographic low that catches rainfall/runoff. Present in i) elevated hard-rock headwaters of catchments and ii) regionally low-lying topographic location. Water levels in aquifer lower than pool water levels. Vertical head gradient between pool and aquifer with unsaturated zone below pool. | Highly variable; hydrochemistry is a function of rainfall and subsequent evaporation. Substantial enrichment of solutes and water isotopes during dry season. Precipitated salts usually wash away in next flood, (or do not form because of low solute concentrations in streamflow source). enrichment in nutrients and dissolved organic matter by evpo-concentration and the vegetation-related accumulation may contribute to the eutrophication of the pool water. | Relies on surface flows and overland runoff, which is directly tied to precipitation. Sensitive to climate but largely independent of groundwater use. Where infiltration capacity is high pools in downstream areas are more vulnerable to reduced rainfall. Water ingestion by animals and/or transpiration of riparian vegetation can dry out the pool. |
| **Alluvium through-flow**<br><br>$\frac{\partial V}{\partial t} = Q_i - Q_o - EA$ | Expression of river alluvium water table and through-flow. Head gradient reflects water table in alluvium. Water levels in pool coincident with water level in adjacent alluvium (cm-scale gradients expected at influent or effluent zones). Bank storage is important for pool water balance. Absence of surface geological features (e.g. hard-rock ridges) or waterfalls. Physical location may migrate as flood-scour re-shapes alluvium bedform. | Pool hydrochemically similar to local alluvial water; enrichment of solutes and water isotopes in pool during dry season limited by through-flow. Enrichment of solutes and isotopes in successive pools as you move down-stream. Flood water flushes through the alluvium and replaces or mixes with any residual stored water (i.e. hydrochemically flood and alluvial groundwater are the same after a flood). More through-flow means shorter pool residence time and less enrichment. | Relatively small changes in rainfall or groundwater level can result in pool drying if the water level in the unconfined (alluvial) aquifer is reduced to below the base of the pool. Impact of withdrawals (either pumping or uptake by phreatophytes) from alluvium depends on volume and proximity to pool. Abstraction from regional aquifers that are hydraulically connected to alluvium may also affect pool water levels by inducing downward leakage from alluvium. |
| **Groundwater discharge**<br><br>1) Geological contacts and barriers to flow<br><br>$\frac{\partial V}{\partial t} = Q_i - Q_o - EA$ | Two sub-types: i) Catchment constriction across ridges, or ii) aquifer thinning due to geological barrier intersecting topography. Presence of waterfalls or surface geological features (hard-rock ridges). Hydraulic head step-changes across pool feature. Carbonate deposits if source aquifer has sufficient alkalinity. | Consistent hydrochemical composition at point of contact/barrier. Evapo-concentration and evaporative enrichment down-gradient of discharge point. Initial pulse of water from runoff may be saline, pool salinity equilibrates with groundwater at low water levels. | Susceptibility to groundwater abstraction depends on scale of source groundwater reservoir (if large then potentially more resilient) and location of groundwater abstraction. Water persistence is less susceptible to changes in rainfall than other pool types. Presence of geological barrier between pool and groundwater abstraction may limit impacts. |
| 2) Topographically controlled seepage from regional aquifer<br><br>$\frac{\partial V}{\partial t} = Q_i - EA$ | Topography intersects i) water table or ii) preferential flow from artesian aquifer. Standing water persists during dry season due to groundwater discharge in absence of rainfall. Negligible recharge to aquifer during flood event (pool is regional discharge zone). Carbonate deposits if source aquifer has sufficient alkalinity. | Consistent hydrochemical composition at point of seepage. Initial pulse of water from runoff may be saline, pool salinity equilibrates with groundwater at low water levels | Susceptibility to groundwater abstraction depends on scale of source groundwater reservoir (if large then potentially more resilient) and location of groundwater abstraction. Hydraulic gradient supporting pools may be similar to pool depth. No geological barrier to limit susceptibility. |

*Water balance of residual pool when disconnected from surface water flows in the absence of rainfall and if only one mechanism is operating.
$V$ is the volume of water in the pool ($L^3$), $t$ is time (T), $Q_i$ is the water flux from the subsurface into the pool ($L^3T^{-1}$), $Q_o$ is the water flux out
of the pool into the subsurface ($L^3T^{-1}$), $E$ is the evapotranspiration rate ($LT^{-1}$) and $A$ is the surface area of the pool ($L^2$).

## 2.1 Perched surface water

Perched surface water can persist in topographic lows that retain rainfall and runoff during the dry
season but are disconnected from the groundwater system (Fig. 1). This can be the case if the pool
directly overlies impermeable bedrock, or if there is a low-permeability layer between the pool and the
water table (Brunner et al., 2009). The presence of this low-permeability layer is essential to maintain
a surface water body that is disconnected from the groundwater system. In the absence of a low-
permeability layer, the surface water will slowly infiltrate into the subsurface (Shanafield et al., 2021).
This low-permeability layer typically consists of clay, cemented sediments (e.g. calcrete) or bedrock
(Melly et al., 2017, Joque et al., 2010). The persistence of water in these pools will depend on a) shading
from direct sunlight and/or, b) sufficient water volume so that it is not completely depleted by
evapotranspiration during the dry season (which will be largely a function of pool depth). Given the
lack of a water replenishment mechanism during the dry season these pools may persist for some, but
not all, of the dry season.
The occurrence and biological significance of such perched pools has been described particularly for
rivers in inland Australia, where contribution of groundwater has been ruled out on the basis of pool
hydrochemistry (e.g. Bunn et al.., 2006, Fellman et al., 2016). For example, along Cooper Creek in
central Australia, geochemical and isotopic studies revealed a lack of connection to groundwater, and
that convergence of flows at the surface and subsequent evaporative water loss-controlled water
volumes in many pools (Knighton and Nanson, 1994; Hamilton et al., 2005). These pools are situated
in depressions caused by erosion through sandy subsurface layers (note that the low-conductivity layer
for perching was not elucidated). It should be noted that definitive characterization of perched surface
water (i.e. disconnected from the groundwater system) requires the measurement of a vertical hydraulic
gradient between the water level in the pool and local groundwater, as well as identification of a low-
permeability layer at the base of the surface water (Brunner et al. 2009). Although the ecological
significance of perched in-stream pools is documented within the literature (Boulton et al., 2003;
Arthington et al., 2005; Bonada et al, 2020), there is typically no detailed analysis of the hydrology and
sampling is synoptic, so the mechanism of persistence is unclear.

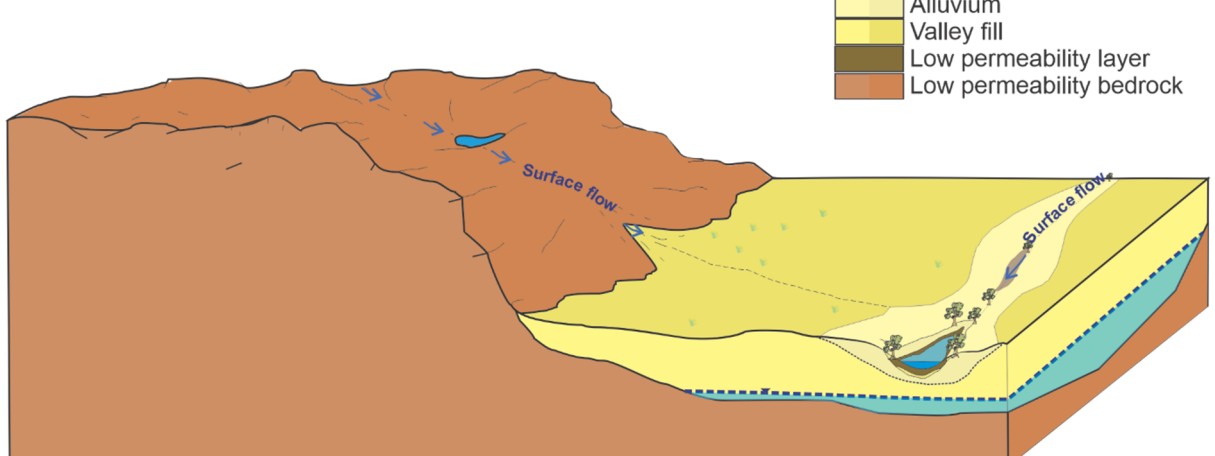

**Figure 1 Schematic illustration of perched pools where rainfall-runoff collects in a depression that has morphology**
**that limits evaporation and/or low permeability lithology beneath the pool that limit infiltration, allowing water to be**
**retained for an extended duration.**

**2.2 Through-flow of alluvial groundwater**
During rainfall events, increases in water levels in rivers result in water storage and flow within the
unconsolidated alluvial sediments in the beds and banks of stream channels (Cranswick and Cook.,
2015). As the streamflow recedes after a flood, continuous surface flow ceases, resulting in isolated
pools along the river channel. Some vertical thickness of the alluvial sediments that line the stream
channel will remain saturated with water beyond the period of surface water flow; this water is hereafter
referred to as alluvial groundwater. Although the subsurface residence times of this alluvial water may
be on the order of months to years (Doble et al., 2012), this water can be accurately described as
groundwater. As such, alluvial water can be considered as a groundwater storage (Leibowitz and Brooks
2008). Persistent river pools can be expressions of this water within streambed sediments (Fig. 2); this
source of water, limited to the floodplain, distinguishes the through-flow mechanism from regional
groundwater discharge.
Alluvial groundwater can be either perched above, or connected to, the regional unconfined aquifer
depending on the depth of the regional water table and the presence of a low- permeability layer to
enable perching (Villeneuve et al. 2015, Rhodes et al., 2017). Once within the alluvial sediments, this
water can subsequently 1) flow laterally through the alluvial sediments towards the bottom of the
catchment 2) be lost to the atmosphere through evapotranspiration, or 3) migrate vertically downward
into lower geological layers (Brooks and Hayashi, 2002; Shanafield et al., 2021; Leibowitz and Brooks,
2008). The recession of groundwater levels in the alluvium will therefore depend on the amount of
water stored, hydraulic gradients and the permeability of the sediments (which control lateral
groundwater flow), hydraulic connections to underlying aquifers (facilitating vertical leakage) and
evpo-transpiration (Brooks and Hayashi 2002; Doble et al., 2012; McCallum and Shanafield, 2016,).
The water level and hydraulic gradients adjacent to persistent pools supported by alluvial through-flow
can therefore change seasonally in response to alluvial recharge by rainfall events, and subsequent
depletion of water stored in the sediments through water flowing into the pool, evapotranspiration and
subsurface groundwater flow (Käser et al., 2009; McCallum and Shanafield, 2016).
The storage and movement of water within alluvial sediments beneath and adjacent to streams has been
described extensively in literature on hyporheic exchange (e.g. Stonedahl 2010) with water fluxes
across temporal (days to weeks) and spatial scales (centimetres to tens of metres). From a hydrological
perspective, the key feature of the hyporheic zone, and hyporheic exchange fluxes, is that it is a zone of
mixing between surface water and shallow groundwater. The scales and mechanisms of hydraulic fluxes
(water movement in and out of the streambed) are determined by streamflow and channel morphology,
which control hydraulic gradients (Stonedahl et al., 2010, Bourke et al., 2014a). Thus, when the stream
is not flowing, the in-and-out hyporheic exchange fluxes that are driven by streamflow are not operating.
However, there can still be exchange of water between surface pools and alluvial water driven by
hydraulic gradients between the pool and alluvial water.
Some authors have considered this exchange through the lens of the hyporheic zone (Käser et al., 2009;
Rau et al., 2017, del Vecchia et al., 2022). However, the hydraulic gradients controlling water fluxes
between pool and alluvium in a stream that is not flowing are not related to changes in stream elevation
along pool-riffle sequences; rather, they are controlled by hydraulic gradient between the pool and water
within the alluvium. While alluvial water that is perched above, and not connected to, the regional
aquifer, does not fit the dominant conceptualization of hyporheic exchange, the physical process that
links streambed elevation changes to flow paths beneath pool-riffle sequences in flowing streams can
be relevant to persistent in-stream pools, regardless of connection status (del Vecchia et al., 2022).
When considering the hydraulic gradient between the pool and the water beside it (as opposed to
beneath it) this can be described as parafluvial flow (Bourke et al., 2014a).  The transient process of
alluvial groundwater recharge and subsequent draining of stored water is also analogous to "bank
storage" adjacent to flowing streams.
An alternative lens through which to consider the exchange of water between persistent river pools and
alluvial groundwater is provided by through-flow lakes, a well-established concept in literature on
surface water – groundwater interaction (Winter et al., 1998). There is a comprehensive body of
literature on the dynamics of through-flow lakes (Pidwirny et al., 2006; Zlotnik et al., 2009; Ong et al.,
2010; Befus et al., 2012). Based on this conceptualization, alluvial water will flow into the pools from
the subsurface across the up-stream portion, and out of the pool, into the subsurface across the down-
stream portion (Townley and Trefry, 2000; Zlotnik et al., 2009). This conceptualization has the
advantage of concisely describing theoretical hydraulic gradients and water exchange between the pool
and the entirety of the surrounding alluvial groundwater, while also accounting for the modification of
the hydraulic head distribution of alluvial groundwater caused by the pool itself. The water level in
pools supported by this alluvial groundwater is effectively a window into the water table within the
streambed sediments (Townley and Trefry, 2000). The rate of inflow to (and outflow from) the pool is
dependent on the hydraulic conductivity of the sediments (Käser et al., 2009) and the balance of inflow
and outflow controls the depth and residence time of water in the pools (Cardenas and Wilson, 2007).
The duration of persistence of the pool will also depend on the storage capacity of the alluvial sediments
that support it; these pools may dry seasonally (Rau et al. 2017) or persist throughout the dry season if
the water level in the alluvial sediments remains above the elevation of the pool.

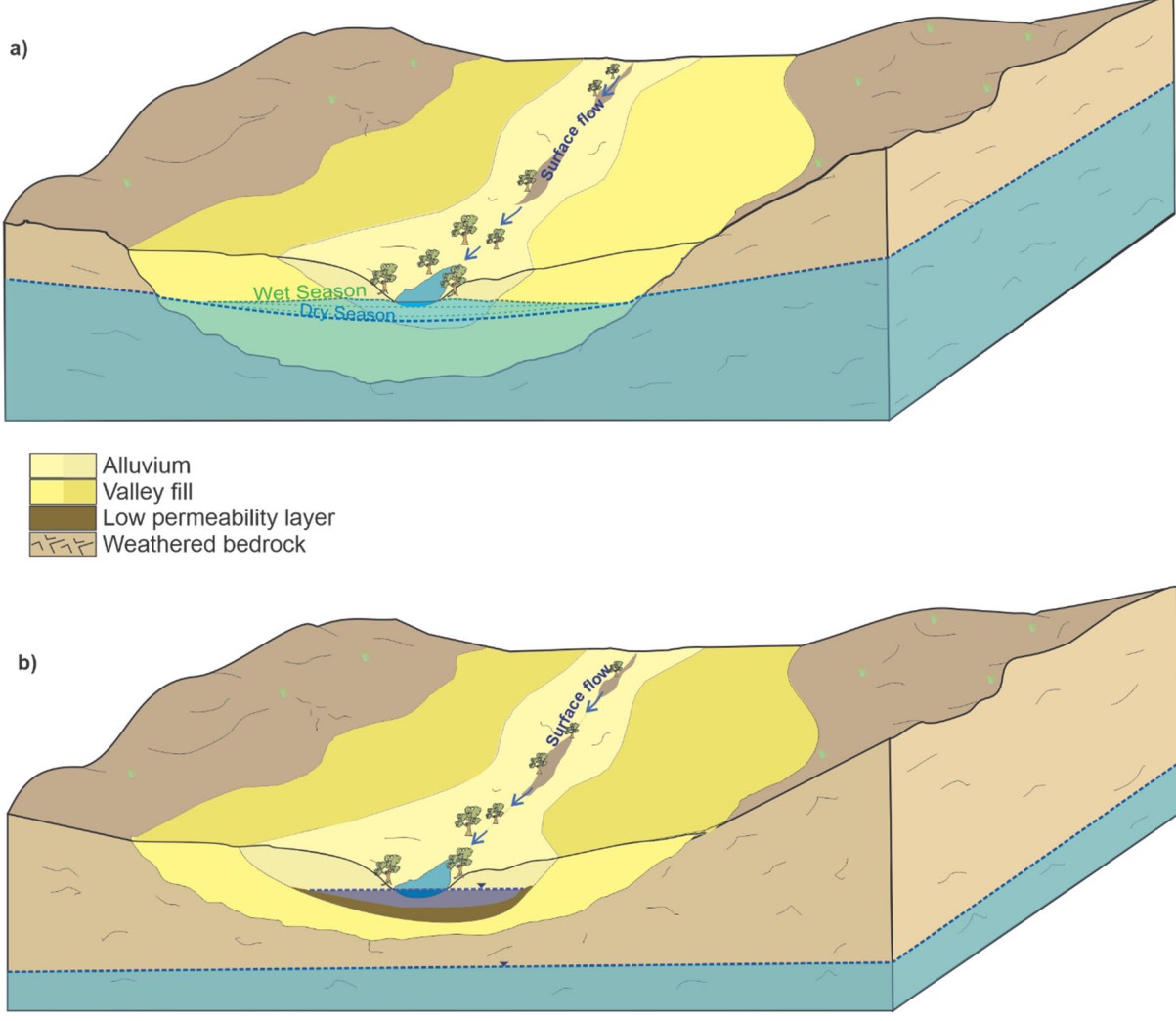


**Figure 2 Schematic illustration of pools that are maintained by through-flow from the adjacent alluvial sediments. The**
**water in these alluvial sediments can be either a) connected to the unconfined aquifer, or b) form a perched aquifer if**
**the water is stored over a low permeability layer.**

## 2.3 Regional groundwater discharge

Similar to springs, rivers can be discharge points for regional groundwater, and this discharge can support the persistence of in-stream pools during periods without surface flow. Groundwater discharge through springs has been articulated into a range of detailed and complex categories, which are not consistent within the literature (Bryan, 1919; Springer and Stevens, 2009; Kresic and Stevanovic, 2010). These existing spring classifications are based on geological mechanism, hydrochemical properties, landscape setting, or a combination of all three, leading to broad categories such as thermal or artesian, as well as nuanced distinctions based on detailed geological structures (Alfaro 1994). For the purposes of understanding persistent river pools, this array of categories is both overly complex and incomplete from a hydraulic point of view. For example, Springer (2009) presents a classification of springs based on their "sphere of influence", which is the setting into which the groundwater flows. A "limnocrene spring" is simply any groundwater that discharges to a pool, as distinct from say a "cave spring", which emerges into a cave. On this basis, one might consider all persistent pools that are not perched as limnocrene springs. However, the schema also articulates 'helocrene springs" which are associated with wetlands and "rheocrene springs" that emerge into stream channels. These also seem to be potentially fitting labels for persistent river pools, which does one choose? And what would it matter for water resource management and the conservation of pool ecosystems if you chose one category over the other?

We suggest two broad categories can encompass the range of hydraulic mechanisms supporting persistent pools in intermittent stream channels; geological features (i.e. lithologic contacts and barriers to flow), and topographic lows. This distinction is valuable because it facilitates an understanding of the source of groundwater discharge (shallow, near-water table vs deeper groundwater) and the size of the reservoir supporting the pool, both of which contribute to the susceptibility of pool persistence to groundwater pumping. This distinction can also be useful for identifying the dominant hydrogeological control on the influx of regional groundwater to the pool; in hard-rock settings with geological contacts and barriers the influx may be limited by the effective hydraulic conductivity, whereas in a topographic

low the influx will be controlled by hydraulic head gradient between the pool and the groundwater
source (see Case Studies below).
**2.3.1 Geological contacts and barriers to flow**
Geological contacts are well-established as potential drivers of groundwater discharge through springs
(Bryan, 1919; Meinzer, 1927). For example, contact springs occur where groundwater discharges over
a low-permeability layer, commonly associated with springs along the side of a hill or mountain (Kresic
and Stevanovic, 2010; Bryan, 1919). Similarly, pool peristence can be supported by groundwater
discharge into a stream channel over a low-permeability geological layer caused by the reduced the
vertical span of the aquifer (Fig. 3a); where this vertical span reduces to zero is known colloquially as
the aquifer "pinching out". This mechanism has been identified as driving regional groundwater
discharge to streams (Gardener et al., 2011), but to our knowledge has not yet been explicitly discussed
in the context of persistent river pools.
Outflow of groundwater where a catchment is constrained by hard-rock ridges that constrict
groundwater flow (by reducing the lateral span of surface flow and the aquifer) can also support the
persistence of surface water pools (Fig. 3b). Although the importance of catchment constriction has
been identified by practitioners (e.g. Queensland Government, 2015), to our knowledge the discharge
of groundwater caused by catchment constriction as a mechanism for surface water generation has not
previously been described in published literature (springs or otherwise).

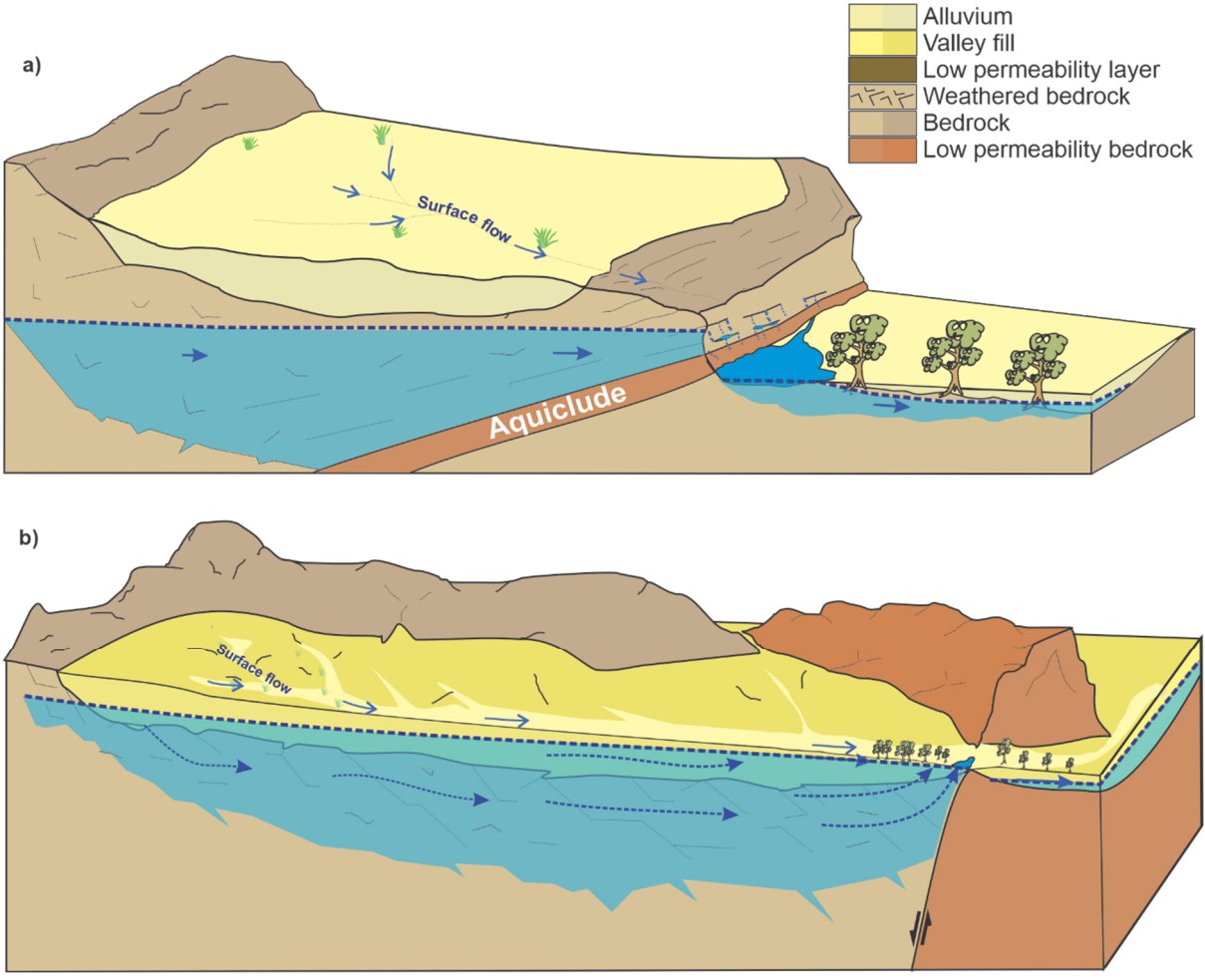

**Figure 3 Schematic illustration of a groundwater discharge pools where surface water persistence is driven by geological barriers that a) cause a regional aquifer to pinch out vertically, or b) form a lateral constraint on the catchment and underlying regional aquifer.**

### 2.3.2 Topographically controlled seepage from regional aquifers

Pool persistence can be sustained by groundwater seepage from regional aquifers in the absence of geological barriers or contacts if there is a topographic low that intersects the regional water table (Fig. 4). This mechanism will generally occur where differential erosion causes a difference in topography, which is equivalent to depression springs (Kresic and Stevanovic, 2010; Bryan, 1919) and analogous to the lakes that form in pit voids left after mining ceases (McJannet et al., 2017). For example, pools likely supported by this mechanism have been identified within the Adelaide region of South Australia where erosion within a syncline has exposed bedrock, facillitating groundwater discharge (Lamontagne

et al., 2021). Within the humid landscape of south-eastern USA, Deemy and Rasmussen (2017) also
describe a vast number of pools along intermittent streams. These pools, which are seasonally connected
by surface flows during the wet season, are expressions of the karst groundwater networks that underlie
them and may be considered special cases of topographically-controlled groundwater discharge pools.
Topographic depressions that fill seasonally with water, known as "sloughs" on the North American
prairie, operate similarly hydraulically (seasonal snow melt inputs, evaporation induces groundwater
inflow), but these sloughs are not within river channels and commonly reside within low-permeability
glacial clays so that they are supported by the local-scale groundwater system (Van der Kamp and
Hayashi, 2009).  Even some Arctic lakes, formed in shallow topographic depressions, receiving
groundwater input and seasonally situated within a stream of snowmelt runoff (Gibson, 2002) can be
considered as pools supported by topographically-controlled groundwater discharge.
Pools may also be sustained by topographically controlled seepage from confined aquifers if there is a
fault or fissure that acts as a conduit to groundwater flow (different to Fig. 3a because there is no
geological transition to sustain a hydraulic gradient across the pool). Topographically controlled
discharge from a confined aquifer is analogous to artesian mound springs like those found in the Great
Artesian Basin of central Australia (Ponder, 1986), but these do not reside within non-perennial streams.
Groundwater discharge along fractures or faults has been identified as an important mechanism for
groundwater discharge to the Fitzroy River in northern Australia (Harrington et al., 2013), but the
significance of this regional groundwater discharge to individual persistent pools is not yet known.

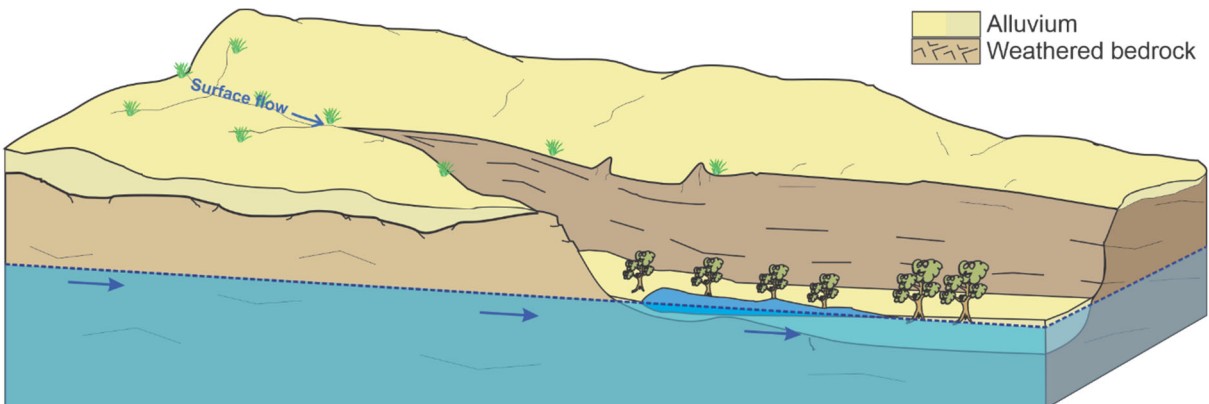


**Figure 4 Schematic illustration of a pool receiving topographically controlled groundwater outflow from a**
**regional aquifer.**

## 3 Management implications: Susceptibility of persistent pools to changing hydrological regimes

Robust water resource management in semi-arid regions requires an understanding of the ways in which
human activities or shifting climates can alter water balances and/or the duration of pool water
persistence (Caldwell et al., 2020; Huang et al., 2020). In the absence of published literature quantifying
the susceptibility of persistent pools, we present general guidance on the susceptibility of pools to
changes in rainfall and groundwater withdrawals based on hydrologic principles (Table 1).
Intuitively, the size of the reservoir (surface catchment or groundwater storage) that supplies water to
the pool should be a key factor in determining the susceptibility of persistent pools to changing
hydrological regimes. However, the patchiness of rainfall and substantial transmission losses typical of
semi-arid zone intermittent river catchments (Shanafield and Cook, 2014) mean that for pools reliant
on surface catchments (perched or supported by alluvial through-flow), catchment size alone is unlikely
to be a robust predictor of resilience. As has been demonstrated for arid zone wetlands in Australia
(Roshier et al., 2001), pools that are storage-limited can be highly sensitive to climate variability.
However, increasing heavy rainfall events may not necessarily result in increased pool persistence
(particularly in pools closest to the location of rainfall) if subsurface storage up-gradient of the pool is
already filling during the wet season. In this case, subsequent rainfall will increase streamflow
downstream, but not result in increased subsurface storage in the reservoir supporting the pool.
Moreover, recent work has shown that groundwater response times are sensitive to aridity, with longer
response times associated with increased aridity (Cuthbert et al., 2019), so that there may be substantial
time-lags between climate variability and hydrologic response in pools supported by groundwater
discharge.
We have distinguished between geological or topographic control on groundwater discharge, but this
distinction may not always be critical from a management perspective. In any system connected to
groundwater, perturbation of the dynamic equilibrium between groundwater recharge and discharge can
impact surface water-groundwater interactions; the timing and extent of the change will depend on the
magnitude and rate of alteration (Winter et al., 1998). The hydraulic head gradients (and groundwater
discharge rates) supporting persistent river pools may be small ($\Delta h$ on the order of cms), so that small
decreases in groundwater level (either due to successive low-rainfall years, or groundwater
withdrawals) can potentially have a detrimental impact on the pool and cause the pool to dry out
(particularly for topographically controlled groundwater discharge to pools).
For pools supported by alluvial through-flow, the water balance is dominated by water outflow from
contemporary fluvial deposits but withdrawals from regional groundwater could impact the pool if these
two subsurface reservoirs are hydraulically connected. The volume of groundwater storage in the source
reservoir can indicate the resilience of pools to hydrological change (i.e. a longer groundwater system
response time), but impacts will also depend on the distance from the recharge zone or groundwater
withdrawals (which could include pumping or uptake by phreatophytes) (Cook et al., 2003). The time-
lag prior to a decrease in groundwater outflow to the pool, and shape of the response (i.e. a slow decline
or sharp decrease), will also depend on the spatial distribution of the forcing (pool distance from
recharge or groundwater withdrawals) (Cook et al., 2003; Manga, 1999). Thus, focussed groundwater
withdrawals close to a pool will cause a larger and faster reduction in groundwater outflow than diffuse
withdrawals across the aquifer, or withdrawals further away (Cook et al., 2003; Theis, 1940). For
example, groundwater withdrawals from within 1 km of a pool will result in a rapid decrease in
discharge (months to years) but the same volume of withdrawal distributed throughout the catchment
will result in a more gradual decline in groundwater discharge to the pool (years to decades).
Susceptibility can be further modified by geological barriers, which may not be obvious from the
surface topography or regional geological maps (Bense et al., 2013), but can isolate pools from the
regional groundwater system and either i) increase susceptibility to pumping within the connected
aquifer, or ii) reduce susceptibility if the pumping is on the other side of the barrier (Marshall et al.

349 2019).

## 4 Diagnostic tools for elucidating hydraulic mechanisms supporting pool persistence

Several tools in the hydrologist's toolbox are appropriate for gathering the data needed to distinguish between the hydraulic mechanisms that support pool-persistence as outlined in the previous section. For most of these, there are no examples in published literature that are specific to persistent pools along intermittent rivers. Therefore, this section provides general background and suggested considerations for the application of these methods to characterize the hydrology of persistent pools (Table 2). A selection of these tools may be deployed at a given site to characterize a) the relationship of the pool to the groundwater system, and b) the relative contributions of evaporation, transpiration, and groundwater fluxes (alluvial and regional) to the pool water balance.

**Table 2 Summary of pros and cons of available diagnostic tools for assessing the hydraulic mechanisms supporting persistent river pools.**

| Diagnostic tool | Strengths | Limitations |
|---|---|---|
| **Regional scale** | | |
| Landscape position and geological context | Low cost, can be assessed using publicly available data. | May be misleading if interpretation made in the absence of robust understanding of subsurface geology and groundwater system. Water balance components not quantified. Surface geology maps may not adequately capture subsurface structures that are important drivers of groundwater discharge. |
| Hydrogeological context | Low cost if the regional hydrogeological system has been previously characterized and water table map (or data) are publicly available. | Hydrogeological maps are not as ubiquitous as surface geology maps and may have been developed based on sparse data sets so that surface water -groundwater interaction is not adequately captured. |
| Remote sensing | Existing data sets available. Requires expertise in spatial data analysis. | Spatial or temporal resolution of data may not be adequate to capture pool hydrology. Water balance components not quantified. |
| Pool hydrography | Water level measuring equipment relatively low cost and readily available. | Equipment may be washed out during flood events. Pool water levels need to be combined with adjacent alluvial and groundwater level data to enable quantification of water balance components. Needs to be combined with bathymetry data to quantify water storage volume in pool. |
| **Pool scale** | | |
| Pool hydrochemistry | Salinity (electrical conductivity) can be measured as a time-series using relatively inexpensive equipment (approx. double the cost of a water level logger). | Multiple discrete samples required to develop time-series. Overlapping values between end-members and spatio-temporal variation can complicate interpretation. |

| Stable isotopes of water | Readily available, low-cost analyses. Mixing and fractionation processes relatively well understood. | Snapshot data interpreted in the absence of an understanding of pool water volumes may be misleading. Sample preservation required for some analytes. Overlapping values between end-members and spatio-temporal variation can complicate interpretation. |
| --- | --- | --- |
| Radon-222 | Distinct end-members between surface and subsurface waters. Can measure in-situ time-series using a portable sampler Rad-7. | Some rock types have naturally low radon concentrations. Requires specialist equipment to measure. Cannot easily distinguish between alluvial and regional groundwater. |
| Temperature | Sensors are relatively cheap, well-established technique for inferring vertical water fluxes in streambeds. Can rapidly and cheaply measure pool-bed temperature. Relatively simple to collect time-series data at multiple depths and estimate vertical water fluxes. | Alluvial and regional groundwater fluxes not easily separated. Can indicate locations of water inflow but not outflow. While vertical fluxes are relatively simple to estimate, lateral fluxes are often overlooked. |

365

366

## 4.1 Regional-scale tools

### 4.1.1 Remote sensing and image analysis

Mapping the persistence of vegetation and water in the landscape based on remotely sensed data (i.e. NDVI or NDWI) or aerial photos can be useful to identify river pools that persist in the absence of rainfall (Haas et al., 2009; Soti et al., 2009; Alaibakhsh et al., 2017). This can be valuable for identifying hydrologic assets that may require risk assessment and protection but does not elucidate the hydraulic mechanism supporting the persistence of the pool. Interpretation of the key hydraulic mechanism(s) supporting a pool requires additional information on the landscape position and hydrogeological context of the pool. In combination, these regional-scale approaches are likely to be particularly useful in remote regions that are difficult to access and pool-specific data collection is challenging.

### 4.1.2 Landscape position and geological context

Landscape position can provide some clues as to the mechanism controlling the persistence of a given pool. For example, a pool located high in the catchment on impermeable basement rock is likely to be a perched pool. Pools that reside within extensive alluvial deposits are likely to be supported at least in part by through-flow of alluvial groundwater. The presence or absence of alluvial deposits capable of hosting a significant volume of groundwater can often be determined by visual inspection, but geophysical tools or drilling are required to confirm the vertical extent of the alluvial aquifer. Similarly,

a pool that is immediately prior to a topographic ridge that constrains the catchment is likely to be
supported by geologically constrained groundwater discharge. Lateral catchment constriction can
commonly be identified from publicly available aerial imagery, but identification of vertical catchment
constriction will usually require geological data from drilling or regional-scale geophysical surveys.
Aerial geophysics (e.g. AEM) in particular can aid in identifying subsurface lithologic geometries and
low-permeability layers that can be important controls on groundwater outflow, but may not be obvious
from aerial photographs or surface geology maps (Bourke et al., 2021). While the locations of
geological can be evident from readily available maps of surface geology, the hydraulic properties of
geological contacts are not always known a-priori. Geological transitions can be zones of high
permeability or barriers, or a combination of both (e.g. faults with high permeability in the vertical, low
permeability laterally) depending on the depositional and deformational history of the area (Bense et
al., 2013).
**4.1.3 Hydrogeological context**
Regional groundwater mapping can provide insights into the mechanisms supporting persistent pools,
particularly if the geology has also been well-characterized (see Case Studies below for examples).
Water table maps can articulate areas of groundwater recharge and discharge, and steep hydraulic
gradients that may (but not definitely) reflect the presence of geological barriers (e.g. Fitts, 2013).
Hydraulic head gradients can provide valuable insights; a step-change in hydraulic head can be a key
indicator for the presence of a hydraulic barrier. If an interpreted water table surface suggests that the
regional water table is tens of meters below ground in the vicinity of a pool, then the surface water is
likely (but not definitely) perched. If a pool is situated in a region that has been identified as a regional
groundwater discharge zone, then this groundwater discharge is likely to be supporting pool persistence.
The presence of active deposition of geological precipitates can also be indicative of pool mode of
occurrence with carbonates associated with groundwater discharge and subsequent degassing of $CO_2$
(Mather et al., 2019).

## 4.2 Pool-scale tools

### 4.2.1 Pool hydrography and water balance

**4.2 Pool-scale tools**
**4.2.1 Pool hydrography and water balance**
If instrumentation can be installed in the pool, then it may be possible to characterize the pool water
balance. Once a pool becomes isolated from the flowing river, a general pool water balance is given by;
$\frac{\partial V}{\partial t} = Q_i - Q_o - EA$, where $V$ is the volume of water in the pool ($L^3$), $t$ is time (T), $Q_i$ is the water flux
from the subsurface into the pool ($L^3T^{-1}$), $Q_o$ is the water flux out of the pool into the subsurface ($L^3T^{-1}$)
$^{-1}$), $E$ is the evapotranspiration rate ($LT^{-1}$) and $A$ is the surface area of the pool ($L^2$). Here we neglect
rainfall on the basis that a significant rainfall event is likely to initiate streamflow, but if this is not the
case, then rainfall can be included as an additional term $PA$, where $P$ is the precipitation rate ($LT^{-1}$). The
water level in the pool, $h_p$ (L), can be routinely measured by installing pressure transducers, but
conversion of water levels to pool water volume (and/or pool area) requires knowledge of pool
bathymetry, and the relationship between $h_p$ and $V$ will change during the dry season as the pool water
level recedes (reducing the pool area $A$), or if pool bathymetry is altered by scour and/or sediment
deposition during flood events. Evapotranspiration rates can be taken from regional data or empirical
equations, but actual losses can vary depending on solar shading, wind exposure and transpiration
(McMahon et al., 2016). For pools with visible surface inflow or outflow, these rates can potentially be
measured using flow gauging (or dilution gauging), but relatively small flow rates and bifurcation of
flow can make this challenging.
Modified versions of this general water balance can be defined for particular pools, depending on the
hydraulic mechanism(s) supporting pool persistence (see Table 1). For perched pools, which are
disconnected from the groundwater system, $Q_i = Q_o = 0$, so that the only component of the water balance
is water loss through evaporation. Pools that are supported by alluvial through-flow are hydraulically
connected to the water stored in the streambed alluvium. Water levels within this alluvium will be more
dynamic than regional groundwater levels, so that influx and efflux rates that can change over time in
response to rainfall events or seasonal drying (of the near-subsurface). Where uptake of water by
riparian vegetation from the groundwater store is substantial this may need to be accounted for explicitly
in a water balance model. For pools supported by groundwater discharge, influx will dominate over
efflux ($Q_i > Q_o$). If the groundwater discharge is over an impermeable aquiclude (see Fig. 3b) there
will commonly be a seepage zone up-gradient of the pool so that water influx is via surface inflow, but
outflow to the subsurface can form a source of groundwater recharge to the adjacent (down-gradient)
aquifer. If the groundwater discharge is controlled by topography, then the pool will be a site of regional
groundwater discharge so that local groundwater recharge (and $Q_o$) should be negligible. An
understanding of pool water balances can be particularly important for interpreting hydrochemical data
(see 4.3)
If a pool is connected to the groundwater system $Q_i$ (or $Q_o$) can be estimated from Darcy's Law; $Q_i =$
$K\frac{\Delta h}{\Delta x}A_i$, where $K$ is hydraulic conductivity, $\frac{\Delta h}{\Delta x}$ is the hydraulic gradient between the pool and the source
aquifer, and $A_i$ is the area over which the groundwater inflow occurs (which will usually be less than
the total area of the base of the pool). The major limitations of this approach are that $K$ of natural
sediments varies by ten orders of magnitude (Fetter, 2001), and that the area of groundwater inflow
needs to be assumed or estimated using a secondary method. Hydraulic gradients between pools and
streambed sediments can be measured using monitoring wells or temporary drive points, with $\Delta h$
usually on the order of centimetres at most. Determination of the hydraulic gradient between regional
aquifers requires that the water level in the pool has been surveyed to a common datum and there is a
monitoring well near the pool to measure the groundwater level relative to that datum. In shallow,
groundwater dominated lakes, geophysical methods have also been used to determine local hydraulic
gradients, and therefore the direction of the water flux(es) between groundwater and surface water (Ong
et al., 2010; Befus et al., 2012). Blackburn et al (2021) similarly applied shallow geophysical surveys,
combined with mapping of hydraulic conductivities, to identify they key structures and processes
controlling water fluxes between groundwater systems and the streams that host persistent pools
(Blackburn et al., 2021).

### 4.2.2 Pool hydrochemistry and salinity (electrical conductivity)

Numerous studies of streams and lakes have employed hydrochemical and mass balance approaches to quantify water sources (Cook, 2013; Sharma and Kansal, 2013) and groundwater recharge (Scanlon et al., 2006). Some of these methods are also applicable in persistent pools, but may require modification, or an iterative approach that allows for refinement of the methods as the mechanism supporting the pool is elucidated. In its simplest form, snapshot measurements of pool hydrochemistry (salinity, pH, major ions) can help distinguish pools that are connected to groundwater from those that are not (Williams and Siebert, 1963). Dissolved ions are relatively cheap and easy to measure and have been used extensively to estimate recharge/discharge, groundwater flow, and ecohydrology in arid climates (Herczeg and Leaney, 2011). Electrical conductivity (EC) as an indicator of salinity can be measured at high temporal resolution using readily available loggers, which can be connected to telemetry systems if required. Time series of EC through flood-recession cycles can indicate relative rates of evaporation and through-flow (Siebers et al., 2016; Fellman et al., 2011) and allow identification of the hydraulic mechanism(s) supporting pool persistence. For example, if a pool is supported by regional groundwater discharge, the EC will re-equilibrate towards the groundwater EC value during the dry-season (see case studies in Section 5); in a perched pool, the pool EC will not plateau, but continue to evapo-concentrate until the next flood event. However, in systems with large flood events, loggers can regularly become lost as the flood moves through, so EC loggers may need to be collected and downloaded prior to anticipated flood events, which isn't always practical.

### 4.2.3 Stable isotopes of water

Stable isotopic values of pool water ($\delta^{18}O$ and $\delta^2H$) can be interpreted similarly to electrical conductivity; groundwater seepage from a regional aquifer will have a relatively consistent isotopic value, while a pool isolated from the groundwater source will experience isotopic enrichment through evaporation (Hamilton et al., 2005), as demonstrated in Case Study 2 (Section 5.2.2). Pools receiving alluvial throughflow will have isotopic values that reflect the balance of inputs (from alluvial groundwater) and outputs (evapotranspiration and outflow to alluvial groundwater). However, the

interpretation stable isotopic values can be limited by overlapping ranges of values across different
water sources (Bourke et al., 2015), and spatiotemporal variability (see Case Studies). The isotopic
values in the alluvial water itself can become enriched through evapotranspiration during the dry season
resulting in variability over time, along the stream and throughout the catchment (Dogramaci et al.,
2015), so that end-member values should be defined locally. In one case, strontium isotopes were found
to be more useful than stable isotopes of water for identifying groundwater contributions to in-stream
pools because the strontium values in the groundwater end-member was far more constrained than
salinity or stable isotope values (Bestland et al., 2017). Importantly, although these data are relatively
easy to measure, their interpretation should ideally be supported by a robust understanding of the pool
geometry, water flow paths and the surrounding geology to ensure that the hydraulic mechanisms
identified are physically plausible (see Discussion).
**4.2.4 Radon-222 and groundwater age indicators**
Radon-222 is a commonly applied tracer in studies of surface water – groundwater interaction, and
$^{222}$Rn mass balances have been effective for quantifying groundwater contributions to streams and lakes
(Cook, 2013; Cook et al., 2008). Preliminary measurements of $^{222}$Rn in persistent pools indicate
substantial spatial variability in $^{222}$Rn activity along the pools, reflecting the spatial distribution of
groundwater influx and gas exchange. This spatial variability will limit quantification of groundwater
discharge based on the $^{222}$Rn mass balance but can allow for hot-spots of groundwater discharge to be
identified (see Case Studies).
Other groundwater age indicators ($^{3}$H, $^{4}$He, $^{14}$C) have been measured along streams to identify
groundwater sources (Gardener et al., 2011; Bourke et al., 2014b), but their applicability in pools is yet
to be determined. Given that shallow, stagnant water is common, tracers such as $^{14}$C or $^{3}$H, which don't
rapidly equilibrate with the atmosphere (Bourke et al., 2014b; Cook and Dogramaci, 2019), are likely
to be better than gaseous isotopic tracers (e.g. $^{4}$He) that equilibrate rapidly (Gardner et al., 2011). If a
mass balance approach is applied, then hydraulic measurements to constrain the pool water balance
should be made in conjunction with hydrochemical sampling to ensure that the water balance is
appropriately reflected in the mass balance.

**4.2.5 Temperature as a tracer**

Temperature measurements have been used extensively to identify and quantify water fluxes across
streambeds and lakebeds (e.g. Shanafield et al.,2010; Lautz, 2012). Diel amplitudes of subsurface
temperatures have been used to identify the transition from flowing stream to dry channel (with isolated
pools) in ephemeral systems (Rau et al., 2017). In persistent pools, temperatures at the water sediment
interface can be used to map zones of groundwater inflow (Conant, 2004). In arid zones, groundwater
temperatures will often be warmer than pool temperatures and this type of survey is best conducted at
dawn when the temperature gradient between pool and groundwater is at a maximum and there are no
confounding effects from direct solar radiation. This mapping can be conducted using point sensors or
thermal cameras, but in natural water bodies this method has primarily found success at thermal springs
where the temperature difference between surface waters and groundwater inflows is on the order of 10
°C (Briggs et al., 2016; Cardenas et al., 2011).
Vertical profiles of temperature can also be used to estimate vertical fluid fluxes but the application of
this approach in pools with coarse alluvial sediments (commonly through-flow pools) is likely to be
limited by lateral flow within the subsurface when $K_h>K_v$ (Rau et al., 2010; Lautz, 2010). Analytical
solutions for temperature-based flux estimates also break-down at low flux rates where the difference
between convection and conduction is difficult to determine (Stallman, 1965). Recently developed
instrumentation for measuring 3D flux fields (Banks et al., 2018) shows promise, but installation in
course alluvial sediments like those commonly found in arid streambeds remains a challenge. Point-
scale measurements also require up-scaling and these methods may not be applicable in fractured hard-
rock pools.

## 5 Application of this framework to persistent pools in the Hamersley Basin

In this section we demonstrate the application of this framework to persistent river pools in north-west Australia (Figure 5). Here we begin by providing an overview of the hydraulic mechanisms supporting persistent river-pools in the Hamersley Basin based on regional-scale tools; landscape position, geological and hydrogeological context. We then present three pool-scale case studies to demonstrate how this contextual understanding can be supported by time-series data from pools to further elucidate spatio-temporal variability in the key hydraulic mechanisms supporting pool persistence.

The Hamersley Basin has an arid-tropical climate with a wet season from October to April and a dry season from May to September (Sturman and Tapper, 1996). Average annual rainfall is less than 300 mm yr$^{-1}$ with most rain falling between December and April (www.bom.gov.au). Annual rainfall statistics can vary dramatically, depending on the influence of thunderstorms and cyclone activity. Thunderstorm activity is commonly highly localised, limiting the potential for spatial interpolation of data from individual monitoring sites. Annual evaporation is around 3000 mm yr$^{-1}$ (www.bom.gov.au), or about ten times annual rainfall, so that permanent surface water is rare. Ranges, spurs, and hills (consisting of bedrock and dykes described below) are separated by broad alluvial valleys with numerous deep gorges created by differential erosion. During large flood events, runoff creates sheet-flow along the main channel and the extensive floodplain can remain flooded for several weeks. In the absence of cyclonic rainfall, surface water is generally limited to a series of disconnected pools along the main channels. Bedrock consists of Archean basement rocks of the Weeli Wolli, Brockman, Wittenoom and Marra Mamamba Formations (youngest to oldest) that are extensively folded and intruded by dolerite dykes (Table 3) with unconsolidated Tertiary and Quaternary sediments overlying them (Dogramaci et al., 2015). The valleys are filled with up to 100 m of consolidated and unconsolidated Tertiary detrital material consisting of clays, gravels, and chemical precipitates. The Quaternary alluvial sediments are deposited along the creek-lines and incised channels (incised on the order of metres) and consist primarily of coarse, poorly sorted gravel and cobbles (thickness of up to tens of metres, widths of up to hundreds of metres) resulting in relatively high hydraulic conductivity

(see Table 3). The region is not considered to host regionally extensive, productive aquifers for water
supply (Brodie et al., 2019) but fresh groundwater is abundant throughout both within the Archean
basement rocks, where permeability is increased via weathering, fracturing or mineralisation, and
within the Tertiary and Quaternary sediments (Dogramaci et al., 2012). The geological basin multiple
surface water catchments with drainage flowing from the headwaters in the elevated areas towards the
Fortescue, Robe or Ashburton rivers.

**Table 3 Estimated horizontal hydraulic conductivities of geological units relevant to pool-scale case studies.**

| Age | Geological unit or formation | Description | *Hydraulic Conductivity (m d$^{-1}$) |
|---|---|---|---|
| Quaternary | Alluvium | Unconsolidated clay, silt, sand, gravel and cobbles | 1000 |
| Tertiary | Alluvium/colluvium | Unconsolidated clays, silts, sands and gravels. Some chemical precipitates | 0.2 - 5 |
| Proterozoic | Dykes | Dolerite | 0.001 |
| Archean | Weeli Wolli Formation | Banded Iron Formation (BIF), mudstone, siltstone, interlayered metadoleritic sills | 0.1 |
| | Brockman Iron Formation | BIF, chert, mudstone and siltstone | 0.3 - 12.4 |
| | Wittenoom Formation | Dolomite, chert, shale, sandstone | 0.001 - 3 |
| | Marra Mamba Formation | BIF, minor shale, stiltstone, mudstone | 0.001 - 10 |

* values from unpublished data and Dogramaci et al. 2015

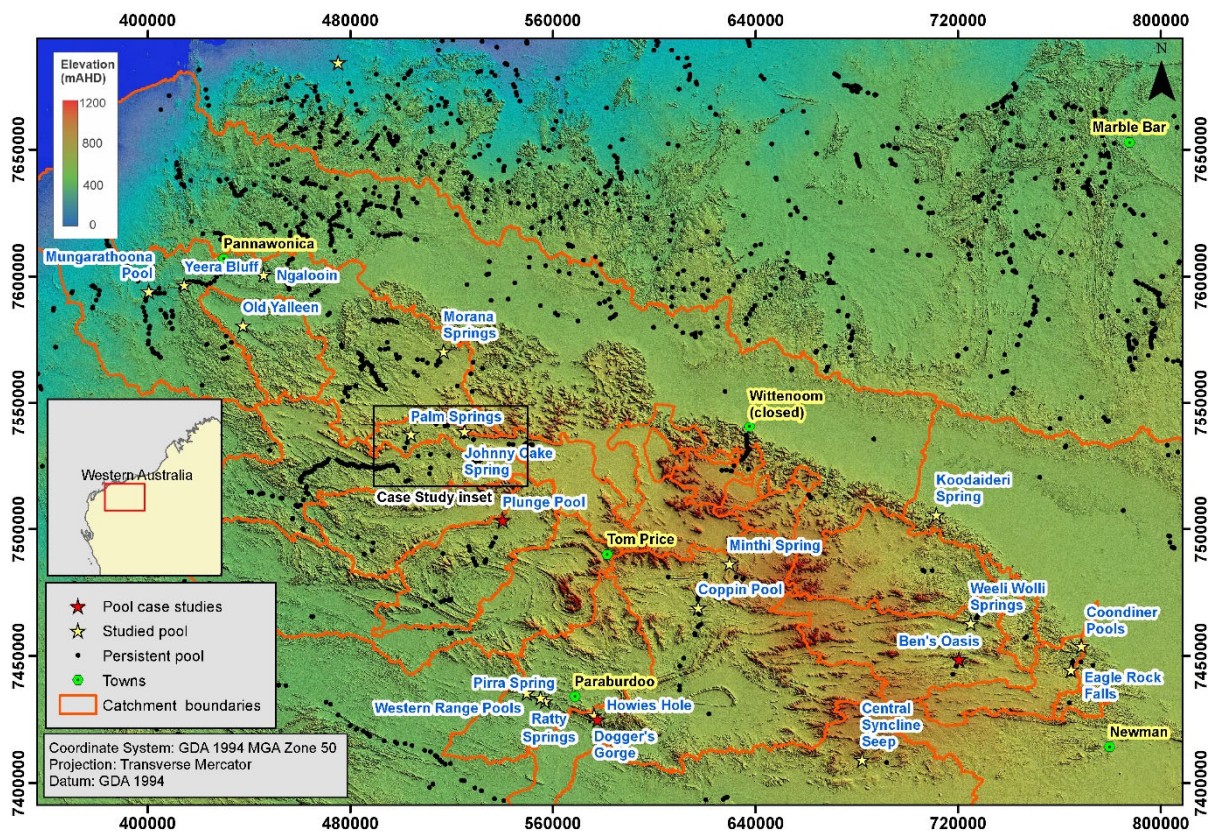


**Figure 5 Map of the Prevalence of persistent pools on watercourses in the Hamersley Basin and selected**

**pools examined in detail. Persistent pools based on "Waterholes" features from Geodata Topo 250K Series**

**3 data set, http://pid.geoscience.gov.au/dataset/ga/63999) Black rectangle indicates extent of Figure 7.**


**5.1 Regional-scale assessment of persistent pools within the Hamersley Basin**

Regional- scale mapping of known pools has provided valuable insights about the distribution of pools

and the likely hydraulic mechanisms supporting their persistence. Broad national-scale mapping of

"waterholes" as part of the publicly available topographic data set for this region identifies many pools

along drainage lines but does not capture all of the known pools (see Figure 5). National-scale

groundwater    dependent    ecosystem    mapping    is    also    available    across    Australia

(http://www.bom.gov.au/water/groundwater/gde/map.shtml). This data identifies the groundwater

dependence of some river reaches within the Hammersley Basin but does not readily allow

groundwater-dependent persistent pools to be differentiated from groundwater-dependent flowing

streams. Image analysis and local knowledge has allowed for the identification of additional pools that
were not mapped within the publicly available dataset.
Overlaying pool locations with topographic mapping allowed a number of pools located at points of
lateral catchment constriction to be identified suggesting that groundwater outflow (either alluvial or
regional groundwater) supports pool persistence. The presence of a topographic constriction was
confirmed using image analysis and direct observation (Figure 6).

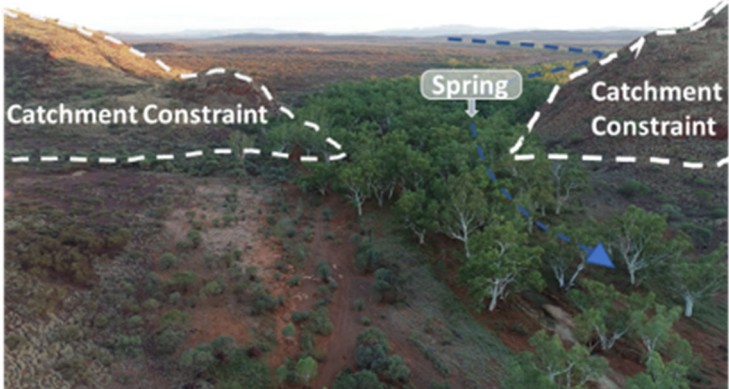


**Figure 6 Photo showing a river pool that persists where a surface catchment is constricted resulting in a**
**groundwater spring.**

By overlaying the locations of pools with available maps of surface geology we identified pools
overlying elevated basement rock in the absence of an extensive alluvial channel, indicating the
potential for these to be perched surface water (see un-named pools at southern extent of Fig. 7 for
example). For other pools their location at the likely edge of a groundwater flow system where low-
permeability basement intersected topography suggested regional groundwater outflow at geological
contacts as an important hydraulic mechanism supporting persistence (e.g. Johnny Cake spring on Fig
7). A number of pools were adjacent to mapped dykes; the persistence of these pools is potentially
influenced by regional groundwater outflow to the surface facilitated by the dyke acting as a hydraulic
barrier within the subsurface. Other pools were located on mapped river- channel alluvium, indicating
the likelihood that through-flow of alluvial groundwater at least partially supports pool persistence (e.g.
pools along the contemporary flow path of Caves Creek in Fig 7). The Hammersley Basin is a fractured-
rock province that does not host aquifers that are used for water supply (GSWA 2015). As such, publicly
available groundwater level data is sparse and regional-scale mapping of water table or depth to
groundwater contours that could further inform an assessment of the connectivity of pools to underlying
groundwater is not available. NDVI mapping was undertaken (Fig 7 inset); while this provides insights
into persistence, it did not allow for the further elucidation of hydrological processes that may be driving
this persistence.

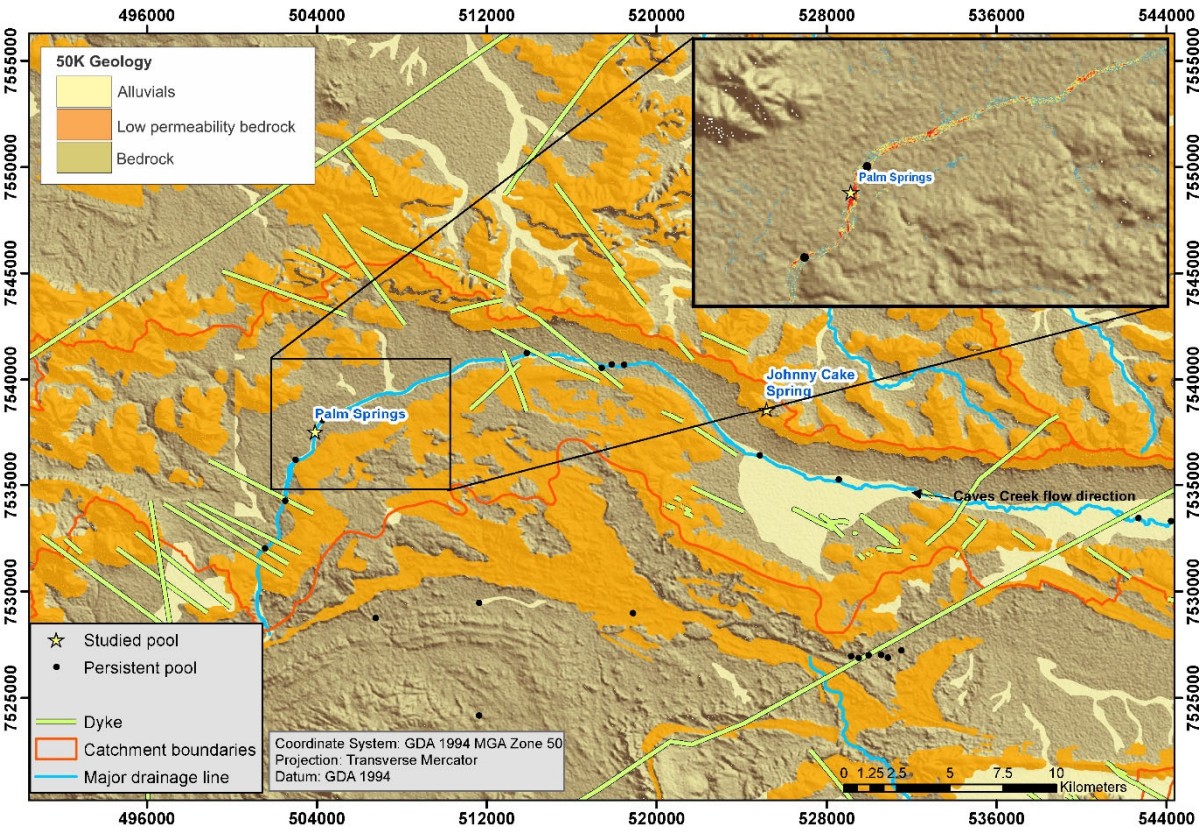


**Figure 7 Map showing locations of persistent pools along Caves Creek relative to bedrock, low-permeability**
**bedrock, dykes and alluvial river-channel sediments (Geological mapping 50K). Inset shows NDVI (on a**
**scale of 0 in blue to 100 in red).**

In-situ observation of the landscape position of pools and the qualitative duration or pool persistence
has also provided valuable insights into hydraulic mechanisms supporting pool persistence. For
example, there are approximately 20 pools that reside within the ephemeral drainage lines of the
Western Range that flow over hard-rock for a few days in response to rainfall and do not have extensive
alluvial deposits; a subset of these pools are deeply incised and shaded persist all year round (Figure
8a). The presence of an extensive alluvial deposits that would facilitate alluvial groundwater
throughflow as a hydraulic mechanism was also able to be inferred based on surface geological mapping
and direct visual observation (Figure 8b). In other cases, alluvial deposits are present, and this alluvium
is shaded within a gorge, so that alluvial through-flow is a major component of the water balance and
evaporation is reduced relative to the alluvium outside the gorge, allowing surface water to persist
(Figure 8c).

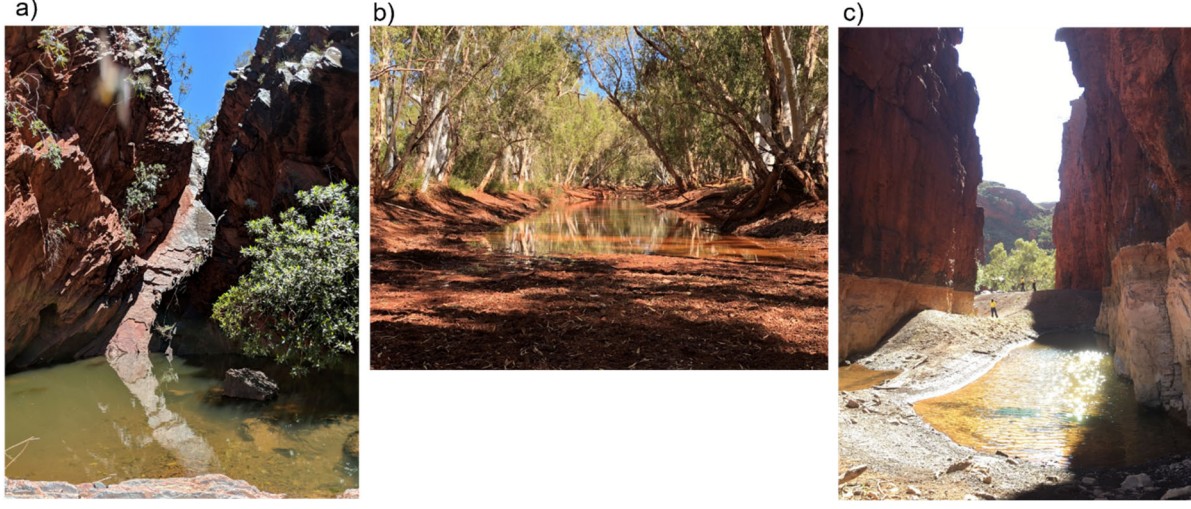


**Figure 8 Photos demonstrating presence or absence of alluvium indicating a) no alluvium and perched**
**surface water, b) extensive alluvium and pool supported by through-flow of alluvial groundwater, c)**
**alluvium within shaded gorge so that persistence of water sourced from outflow of alluvial groundwater is**
**enhanced.**

The exposure and outflow of regional groundwater as a mechanism supporting pool persistence was
also able to be inferred from regional-scale data sets and direct observation in some cases. For example,
some pools persist within deeply incised river gorges that do not contain extensive alluvial deposits and
are therefore likely to be supported by topographically controlled outflow of regional groundwater
(Figure 9a). In another case, groundwater was observed visibly seeping from an exposed rock-face
above a pool (Figure 9b) inferring that the exposure of that geological unit at the surface and subsequent
outflow of groundwater is important for supporting the persistence of that pool.

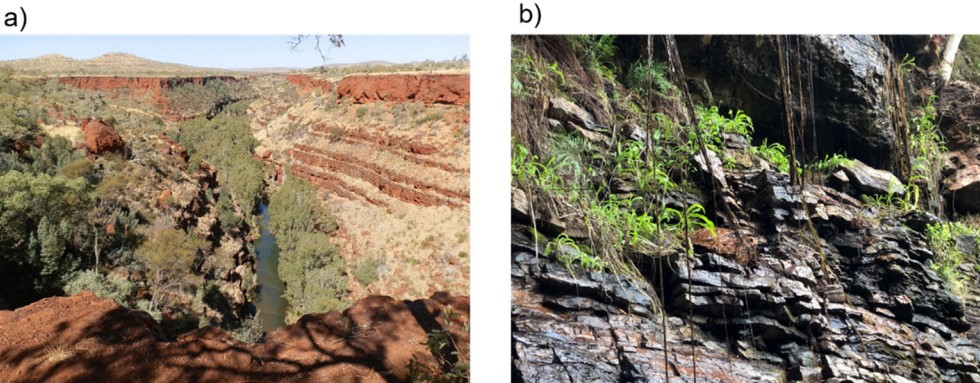


**Figure 9 Photos showing a) river pool persisting within a substantially incised gorge suggesting regional groundwater outflow supports pool persistence (*This Photo by Unknown Author is licensed under CC BY-ND), and b) visible groundwater outflow from an exposed rock face above a persistent river pool.**


## 5.2 Pool-scale case studies

The following three case studies demonstrate the application of this framework to three different pools (or pool systems) within the Hamersley Basin. To the best of our knowledge these pools have not been impacted by groundwater withdrawals or surface water diversions. These case-studies we aim to demonstrate a) the value of understanding the hydrogeological setting of each pool, and b) saptio-temporal variability pool water balances and hydrochemistry. Each case study begins by describing our understanding of the landscape position, geological and hydrogeological context of the pool. We then introduce time-series data; the first case study utilizes water levels and EC in the pool and groundwater; the second case study adds in time-series of stable isotopes values of pool water and groundwater; the third case study brings in radon-222 and temperature mapping. By combining time-series data with an understanding of landscape position and hydrogeological setting, we are able to infer hydraulic mechanisms supporting pool persistence. The implications of the identified hydraulic mechanisms for the susceptibility of the pools to groundwater withdrawals or changing climate are also discussed.


### 5.2.1 Case study 1: Plunge Pool

Plunge Pool (Fig. 10a) is located at the base of a steep topographic drop-off that exposes the Marra
Mamba Formation. The Wittenoom Formation and underlying Marra Mamba Formation are
hydraulically connected and form an unconfined regional aquifer where there has been sufficient
weathering and fracturing to generate secondary porosity. This aquifer is 50-100 m thick and divided
laterally by (sub-)vertical dykes on the order of 1 km apart (but as close as 100 m) that act as hydraulic
barriers within the groundwater system. The surface catchment has an area of approximately 26 km$^2$
and is storage limited. Regional groundwater in the adjacent aquifer has a hydraulic head of 547 m
above sea level (asl) at a distance of 200 m from the pool, increasing to 557 m asl 600 m from the pool,
indicating the presence of dyke that acts as a geological barrier between these two monitoring wells.
Seasonal variation in groundwater hydraulic heads is minimal (on the order of 0.2 m).
The pool is perennial with seasonal water level fluctuation between 541 and 543 m asl driven by
variation in streamflow, groundwater inflow and evapotranspiration (Fig. 10b). The varying proportions
of the pool water balance components are reflected in the temporal variation in the salinity of water in
the pool. At the onset of the first wet season flood the salinity in the pool spikes (up to 4171 mS/cm),
reflecting the flushing of surficial salts that were deposited during the previous dry through the
catchment. Subsequent rainfall events then cause a rapid freshening of the pool (to as low as 124 mS/cm
within 1 day). In the absence of rainfall, the salinity of the pool equilibrates to that of groundwater in
the regional aquifer (900 mS/cm). Given the consistency of groundwater levels, this inflow rate will be
relatively constant, so that (in the absence of streamflow) the variability in the salinity of the pool is
driven by seasonal variation in temperature and evapotranspiration (Bureau of Meteorology Station
#007185, Paraburdoo Aero). These seasonal weather patterns drive evapo-concentration of solutes in
the pool as water levels fall during the dry season and freshening of the pool as water levels rise when
evapotranspiration decreases in winter (May-Sep). Measurement of the relationship between water
levels and pool water volume will allow for these pool water balance components to be quantitatively
resolved.
Based on these data, the dominant hydraulic mechanism supporting the persistence of this pool is
attributed to groundwater inflow from the regional aquifer that is intersected by a topographic low
(Section 2.3.2). Despite the source being a regional aquifer, the spatial extent of the groundwater
reservoir supporting the pool is limited by the presence of geological dykes (Fig. 10c). The pool
effectively acts as a "drain" on the underlying/adjacent compartment of the unconfined aquifer with the
inflow rate to the pool controlled by the hydraulic conductivity of the aquifer (variation in groundwater
levels is negligible). The pool is also hydraulically connected to the alluvial aquifer and water from the
pool is likely to infiltrate into the alluvium on the down-gradient side, but this has not been measured
directly (alluvium is absent up-gradient of the pool – therefore alluvial through-flow is not a supporting
mechanism). The susceptibility of this pool to groundwater withdrawals is controlled by the
hydrogeological compartmentalization. The pool will be more susceptible to groundwater withdrawals
from the aquifer between the nearest dyke and the pool, and less susceptible groundwater withdrawals
outside of this compartment. Given that evaporation is an important component of the water balance
and contributes to the regulation of water levels, this pool is also susceptible to increases in
evapotranspiration that are predicted as temperatures increase under climate change (IPCC, 2021).

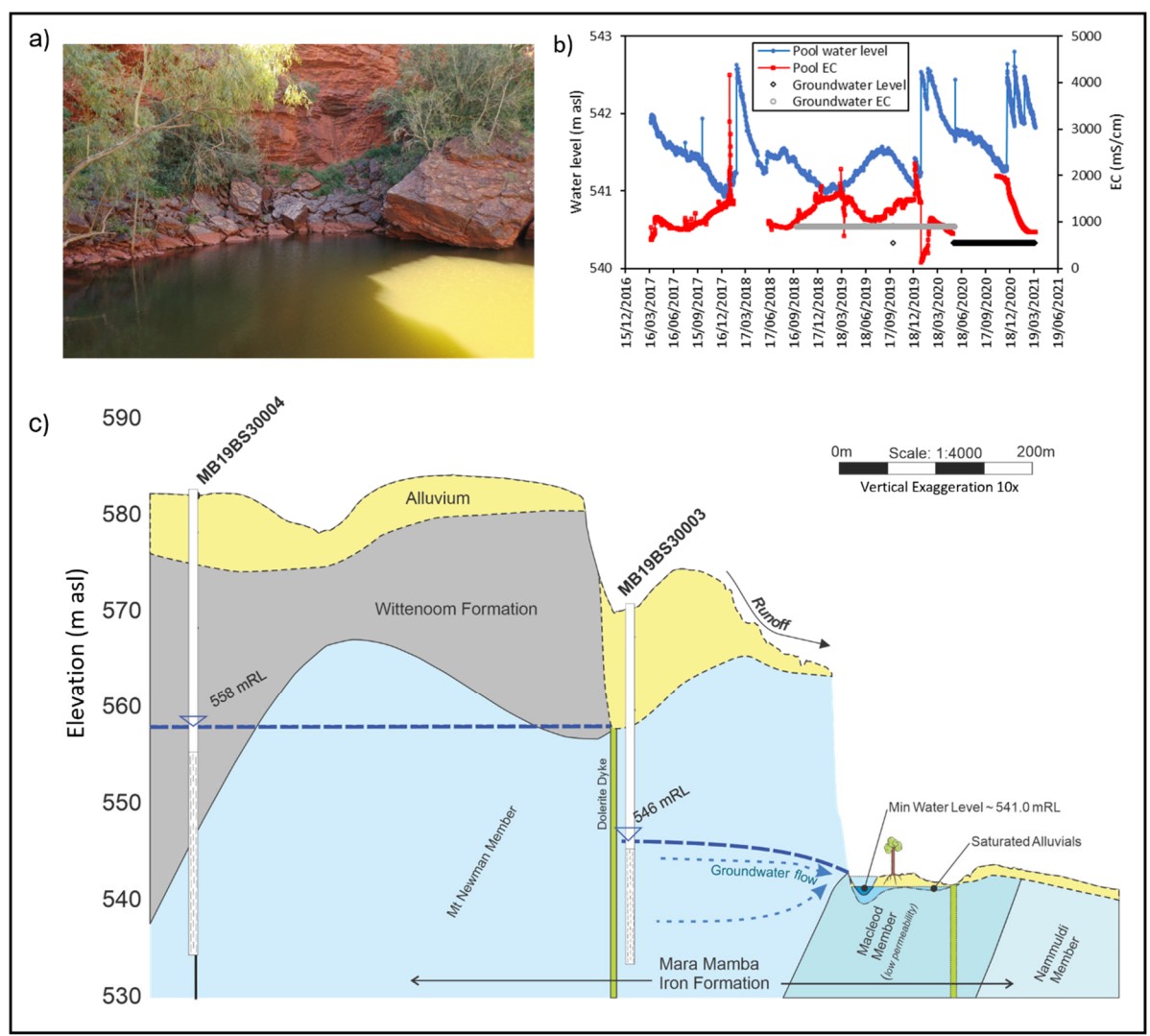


**Figure 10 a) photo of Plunge Pool, b) pool water level and electrical conductivity (EC), c) hydrogeological setting of the pool.**


### 5.2.2 Case Study 2: Howie's Hole

Howie's Hole is a pool within stream channel alluvium at the exit point of a short, narrow gorge (Fig. 11a). Immediately at the outlet of the gorge (approximately 30 m up-hydraulic gradient of the pool) there is also a seep where groundwater outflows to surface for most of the year (seep dries for approximately 2-3 months at end of dry season). The seep is supported by the regional unconfined aquifer hosted within the Marra Mamba Formation and the surficial sediments above it (including the alluvial channel sediments), which are hydraulically connected. At the seep, the Brockman

formation has become adjacent to the Marra Mamba due to faulting and this forms a relatively impermeable hydraulic barrier approximately 700 m wide (identified by the abrupt change in water table depth either side of the formation). The surface catchment upstream of Howie's Hole has an area of 33 km$^2$. The gorge restricts the stream channel from 30 m width down to a channel width of 10 m, enhancing the water depth and flow rate and resulting in scour and erosion of the Brockman formation. This area of scour during high-flow events has subsequently been filled by deposition of unconsolidated alluvial sediments, which are now at the base of the pool (sediments speculated to be 5-10 m deep).

The height of the regional water table is only known 1.5 km away from the seep, with seasonal fluctuations of 1-2 m (Fig. 11b). We assume that the water table declines towards the seep consistent with topographic elevation change ($\sim$ 20 m drop over 1.5 km), and that the seep reflects the height of the water table at that location (elevation of the groundwater seep is 405 m asl). During the period of observation, the groundwater seep dried up when the measured water table elevation dropped below ~418.4 m asl (water sample collected when the measured water table was at 418.5 m asl on 12th Nov 2018); the seep was dry when the measured water table was at 418.3 m asl on 7$^{th}$ Dec 2018. Pool water levels track groundwater elevations above 418 m asl, but data from 2019 shows the pool depth levelling off as the water table at the monitoring bore drops below 418 m asl, suggesting the cessation of significant groundwater inputs. The pool water levels have not been surveyed to the Australian Height Datum, but pool water level is consistently below the elevation of the seep (approximately 398 - 400 m asl).

Similar to Plunge Pool, the pool salinity spikes with the seasonal onset of rainfall, before freshening once the accumulated salts have flushed through (Fig. 11c). In the absence of rainfall, pool salinity is similar to groundwater at the water table (Marra Mamba EC 1140 uS/cm). Isotopic values were available for 2018 (which does not overlap with the data EC and water level data). During this dry-season isotope values of the seep and pool were relatively consistent until August, when the pool

isotopic values began to enrich suggesting decreased inputs from groundwater as the water table
receded (Figure 11d).
Based on these data we conclude that Howie's Hole reflects the water level in the alluvial aquifer
within the stream channel (Fig. 11e). The location of the groundwater seep is determined by the
geological contact between the permeable Marra Mamba Formation and impermeable Brockman
Iron Formation in the subsurface, which coincides with the catchment constriction (gorge) that
forms an outlet for surface and groundwater. As a result of the streamflow regime caused by this
catchment constriction, the Brockman Iron Formation has been eroded and subsequently filled with
unconsolidated stream channel sediments; water storage within these sediments now support the
persistence of this pool.
The water level and isotopic data indicate a threshold groundwater level for inflow of groundwater
to the pool, such that the pool water balance is primarily dominated by groundwater recharged
during the previous wet season. Below this threshold water level for groundwater inflow, the
persistence of the pool relies on local water storage within the streambed alluvium (supporting pool
depths of up to 0.2 m). The persistence of this pool is therefore susceptible to 1) wet season rainfall
that is inadequate to recharge the unconfined aquifer to above the threshold water level, or 2)
groundwater withdrawals that reduce seasonal peak groundwater levels to below the threshold level.
In the absence of this groundwater inflow, the pool is supported by water stored locally within the
streambed sediments (directly beneath the pool) and would be more susceptible to drying through
evapotranspiration (less inflow but the same amount of water loss through evapotranspiration).

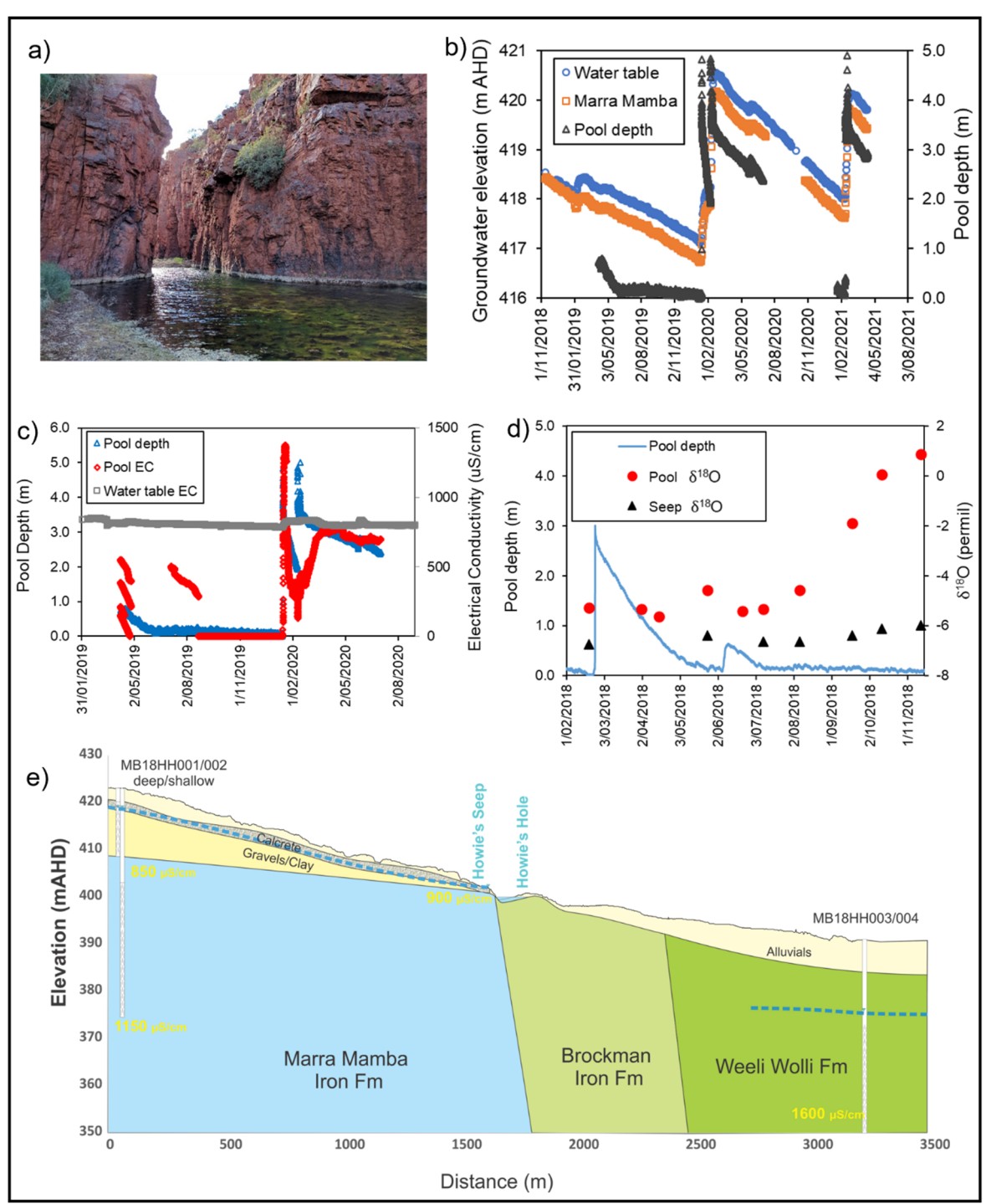

Figure 11 a) photo of Howie's Hole, b) groundwater elevations and pool depth, c) pool water levels and electrical conductivity of pool and groundwater, d) pool depth and δ¹⁸O showing stable isotopic composition at groundwater seep and evaporative enrichment down-gradient of seep during the dry season, e) conceptual diagram of pool occurrence.

### 5.2.3 Case study 3: Ben's Oasis

Ben's Oasis is a sequence of three pools (Pool 1, 2 and 3) that are hydraulically connected during peak water levels and subsequently disconnect during the dry season (Fig. 12a). The pools sit within a major drainage channel that consists of poorly sorted, fine to very coarse (gravel and boulders) unconsolidated alluvial sediments 10's of metres wide and on the order of metres in thickness. The regional water table is within the fractured dolomite of the Wittenoom Formation, which overlies the Marra Mamba Formation. The pool is 2 km up-hydraulic-gradient of two parallel dykes, with a regional water table decline of approximately 20 m across these dykes indicating that they act as a barrier within the groundwater system. Water levels in the upper pool have been monitored since 2016 and in 2019 a detailed study commenced using environmental tracers to assess the spatial variability of surface water – groundwater interaction along this pool sequence (Chapman, 2019).

Measured pool water levels show consistent seasonal trends with water level spikes of 2-3 m in response to cyclonic rainfall events during summer, followed by approximately five months of relatively steady water levels and then recession over approximately three months (Fig. 12b). These trends are consistent with the water level variation in the adjacent alluvium, which exhibits a similar period of steady water levels then recession following the cessation of summer rains. In contrast, regional groundwater levels increase by about 2 m in response to summer rainfall and then immediately begin to recede. Thus, although snapshot water level measurements indicate that pool water levels are consistent with the regional water table, transient water level data (that includes the water level in the alluvium) demonstrates that inflow of water from within the alluvial sediments within the drainage channel is the dominant driver of water level fluctuations in the upper pool (where the logger was installed). Spatial trends in the persistence of surface water and surface geology are also informative at this site. The regional Wittenoom aquifer is exposed at surface around Pools 2 (some alluvium present) and 3 (no alluvium, just bedrock), but not at Pool 1 (no bedrock, just alluvium). The upper, shallower section of Pool 1 and Pool 3 dried out as the dry season progressed, but the deeper parts of Pools 1 and 2 persisted throughout the dry-season (during 2019 and 2020). We interpret these spatial patterns of persistence as

reflecting spatially variable evaporation rates (i.e. more or less shading by vegetation), transpiration
rates and groundwater inputs (Chapman, 2019).
The results of longitudinal hydrochemical surveys ($^{222}$Rn and $\delta^{18}$O) along the pool sequence provide an
independent line of evidence to validate this interpretation (Fig. 12c). Alluvial water had a $^{222}$Rn activity
of 17.6 Bq L$^{-1}$ and $\delta^{18}$O of -6.3 ‰. The regional Wittenoom aquifer had a lower $^{222}$Rn activity of 8.1
Bq L-1 and more depleted $\delta^{18}$O of –7.26 ‰. At the top of Pool 1, $^{222}$Rn activity was 7 Bq L$^{-1}$. Given
that degassing of radon to the surface is rapid and the water level at the time of sampling was shallow,
the source of water inflows must have a much higher $^{222}$Rn activity than 7 Bq L$^{-1}$ and it is therefore
most likely that inflows here are dominated by the higher-Rn alluvial water. Isotopic $\delta^{18}$O values of
around –6 ‰, are also consistent with inflow of alluvial water. $^{222}$Rn activities then decrease along the
pool to around 0.5 Bq L$^{-1}$ (indicating degassing, and the absence of further groundwater inputs) as stable
isotopic values enrich to just over –5 ‰ (reflecting evaporation and the absence of further groundwater
inputs). Water at the up-stream end of Pool 2 had $^{222}$Rn of 2 Bq L$^{-1}$ (greater than at the down-stream
end of pool 1) and $\delta^{18}$O of -6.3 ‰ (more depleted than at the bottom of Pool 1). These data indicate
further water inflows from the subsurface, along this pool, with a lesser proportion of alluvial water,
and more regional groundwater, as well as through-flow from Pool 1 (inferred from relative water levels
in the pools). In Pool 3, $^{222}$Rn remains around 2 Bq L$^{-1}$ indicating further groundwater inputs, but the
stable isotopic values are more enriched (possibly due to the shallow water depth allowing for enhanced
evaporation).
Streambed temperatures within the pools were also mapped (temperatures measured every 0.2 - 1 m
along transects 1-10 m apart) in early September, when regional groundwater was 29 °C, and alluvial
water was 20 °C (Fig. 12d). Measured temperatures were recorded at dawn to reduce the effect of direct
solar radiation and pool depth variability (max pool depth was 0.5 m). Streambed temperatures in the
pools ranged from 17-23 °C, with the warmest water (>20 °C) at the top of Pool 1, and temperatures
between 19-20 °C in middle of Pool 2 and at the top of Pool 3. These results are broadly consistent with
the other results, but the approach is likely to be more conclusive in the presence of larger temperature
gradients. The application of vertical temperature profiles to infer water fluxes at this site was also
limited by the substantial lateral component of the subsurface flow-field (i.e. violating the assumption
of 1D flow) and flood events that removed or damaged monitoring infrastructure.
Based on these data we conclude that the persistence of Ben's Oasis throughout the dry season is
supported by regional groundwater inflows from the unconfined aquifer where it is exposed at surface
(see Section 2.3), but the water balance of Pool 1 is dominated by exchange with the alluvial water (see
Section 2.2). This importance of the alluvial water storage in supporting the largest of these pools is
only evident based on time-series water level data from the alluvium. Given only snapshot water level
measurements from the regional aquifer and one location in the pools, the similarity in water level
elevations would lead to the conclusion that regional groundwater discharge was the dominant
supporting mechanism. The substantial spatial variability captured in the longitudinal hydrochemical
survey also highlights the risks of making conclusions about surface water – groundwater interactions
from snapshot hydrochemistry measurements in just one location within a given pool or pool sequence.
Subsequent numerical modelling of the groundwater system indicates that the presence of the regional-
scale dykes east of the pool operates as a hydraulic barrier within the groundwater system, supporting
the regional water table west of the dykes, promoting regional groundwater outflow to the surface at
the pool (Jen Gleeson pers. comm).
These data allows us to infer that there are two hydraulic mechanisms supporting the water balance and
persistence of these pools; alluvial through-flow and regional groundwater discharge (Fig. 12e). The
persistence of these pools through the dry season is dependent on influx of water from the regional
unconfined aquifer. They will therefore be susceptible to groundwater withdrawals from the regional
aquifer if they reduce the hydraulic head to below the level of the ground surface at the pools. The water
balance of these pools is also controlled by the interaction with water stored in the alluvium (alluvial
through-flow). Therefore, the pools are also susceptible to reductions in rainfall or increases in
temperature (and evapotranspiration) that reduce the volume of water storage (and therefore water
levels) within the streambed alluvium. A reduction in the area of the surface catchment, which can result
from mining operations, could also similarly alter the water balance of these pools.

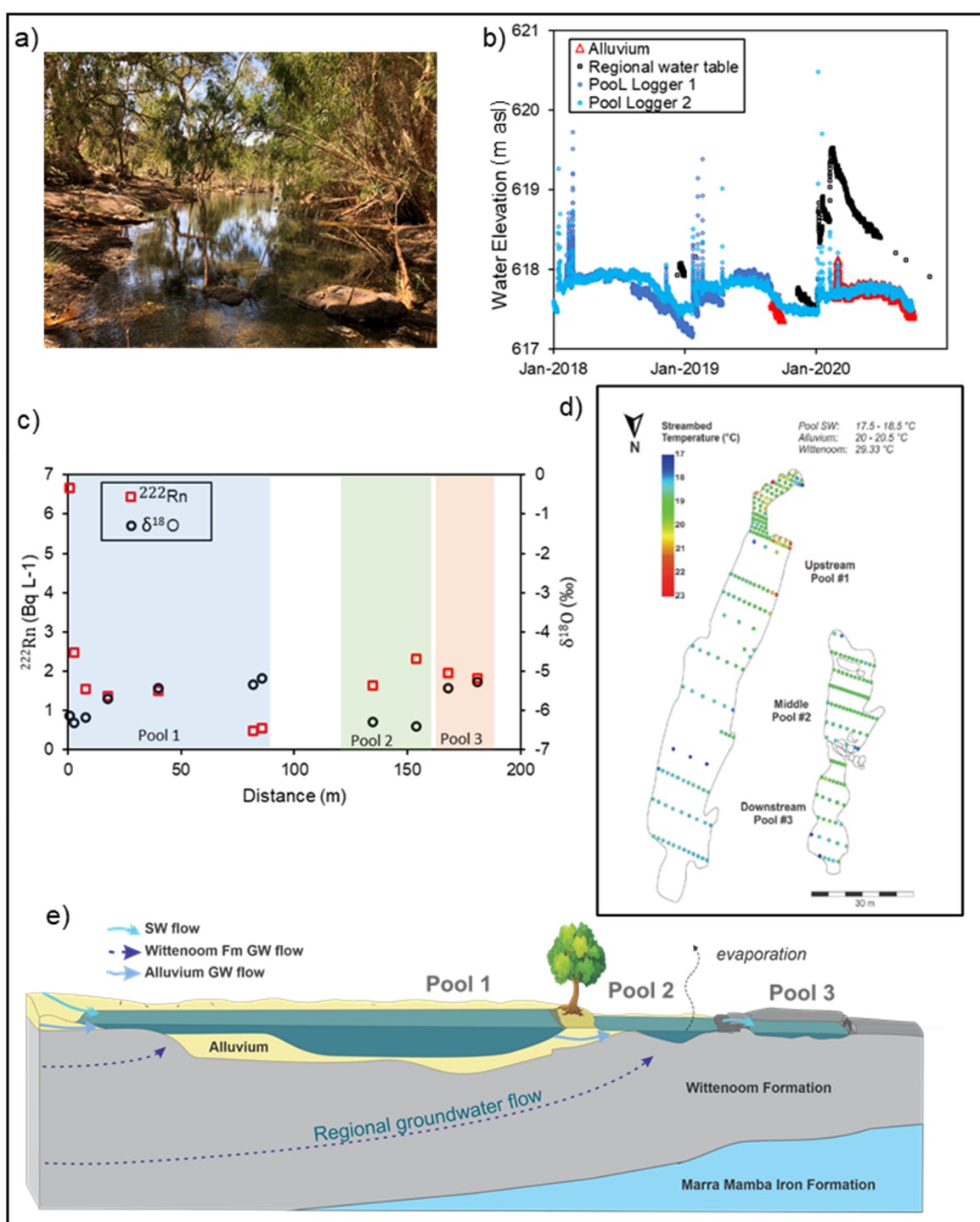

**Figure 12 a) photo of Ben's Oasis, b) water levels in the Pool 1 (logger 1 elevation was surveyed, logger 2 elevation was**
**approximated by matching data from logger 1), alluvium (DP1) and regional unconfined aquifer, c) spatial variation**
**in radon activities and δ¹⁸O along the pool sequence, d) temperature mapping of pool sediments and e) conceptual**
**diagram of mechanisms supporting pool persistence.**

## 6 Discussion

It has now been 100 years since groundwater springs were documented in published literature (Bryan,
1919; Meinzer, 1927; Meinzer, 1923). In that time, the literature on springs, surface water-groundwater
interactions, and non-perennial rivers have all expanded consderably. The goal of the present work has
been to synthesize concepts from all of those fields to aid in the identification of hydraulic mechanisms
that support in-stream pools. Thus in Section 2, we identified four primary pool types, discussing
hydraulic mechanisms for each conceptually and identifying relevant background literature to support
each. In section 4, we then provide a toolbox for use on individual pools and at the regional scale, and
show in section 5 how this toolbox can be used through a series of case studies. This identification of
the hydraulic mechanisms is essential for effective management of risks to pool ecosystems associated
with groundwater withdrawals, changes to the hydraulic properties of the catchment (e.g. land use
change) or climate change, as discussed in Section 3.
Across the three case studies, the persistence of all pools was related to geological contacts that resulted
in regional groundwater outflow. Plunge Pool and Howie's hole are both located where a low-
permeability geological unit results in groundwater outflow to the surface; the water cannot easily
continue to move in the subsurface, so it emerges at the point of contact. At Ben's Oasis, there is regional
groundwater outflow where saturated fractured rock is exposed at the surface and the hydraulic head is
above the land surface. In all of these cases, there were additional hydraulic mechanisms supporting
pool persistence. At plunge pool the geological contact coincides with a topographic low; at Howie's
Hole there is a catchment constriction; at Ben's Oasis alluvial through-flow is a key determinant of pool
water levels. Thus, the water balance of persistent pools can respond to a combination of hydraulic
mechanisms, and the dominant mechanisms can vary spatially and temporally within pools. In all three
cases, the pool(s) contain a mixture of water from streambed sediments (alluvial through-flow) and
regional groundwater during certain hydroperiods, but the pool likely wouldn't persist through the dry
season in that location without groundwater discharge from the regional aquifer. Thus, the maintenance
of the stream ecosystem in its current state would require preservation of in-stream water storage and
regional groundwater inflows. Such pools combining alluvial through-flow pools with some regional
groundwater input are likely common, but it can be difficult to definitively identify and/or quantify this
regional component of the water balance.
Given the potential for this complexity, we advocate for the use of multiple lines of evidence in
determining hydraulic mechanisms, in-line with the accepted paradigm in surface water-groundwater
interactions literature (Kalbus et al., 2006). Regional-scale tools provide a valuable method to make a
first estimate of hydraulic mechanisms; however, highly instrumented sites with robust geological
mapping, monitoring wells, and temporal hydrologic data are required to elucidate spatiotemporal
variability in the pool water balance. Likewise, snapshot data from multiple pools at one point in time
can help distinguish perched pools vs groundwater discharge pools (i.e. pool water hydrochemically
similar or different to rainfall or groundwater), but in some cases water-types (or end-members) are
difficult to distinguish based on easily measured parameters like electrical conductivity or stable
isotopes of water (Bourke et al., 2015).
However, we also acknowledge that direct measurement of water balances in arid and semi-arid regions
can be logistically difficult (Villeneuve et al., 2015). Rainfall (and therefore runoff) in arid and semi-
arid environments is commonly patchy and water fluxes can be either too large to measure (streamflow
during a cyclone) or too small to measure directly (dry-season groundwater seepage fluxes) (Shannon
et al., 2002; Shanafield and Cook, 2014). There are also potential logistical constraints that can apply
when installing any infrastructure for sampling and monitoring in-stream pools. Persistent pools in arid
landscapes are commonly sites of environmental and cultural significance (Finn and Jackson, 2011; Yu,
2000) so that appropriate approvals and permissions typically must be obtained prior to the installation
of monitoring infrastructure. This may restrict the types of data that can be collected. Moreover, some
sites may be sacred sites, limiting who is able to access them. Surface water features in general are a
draw for travellers and roaming livestock, so that any infrastructure must be secure from theft or
damage. Flood events and sudden, flashy streamflows are also potential threats to infrastructure, with
substantial sediment and vegetation (branches, trees) transported across the floodplain to heights of 2-
3 m that can (and have) destroyed sampling equipment. Infrastructure damage by unseasonal or early
rainfalls in particular can impact our ability to capture regional groundwater contributions, since this is
typically a relatively small (but important) component of the water balance of pools and is most readily
captured at the end of the dry season.
With limited resources and access to sites, trade-offs must therefore be made between detailed
characterization of one pool vs a minimal data set at many pools. In our experience, utilizing detailed
data from fewer pools, is more likely to provide a robust characterization of pool hydrology at a scale
required for management than snapshot data from many pools across a region, which can be open to
misinterpretation. For example, while there is no isotopic fractionation associated with the outflow of
water from a pool, there is mass removal of water (and solutes) from the pool this. As such, this water
loss term should be accounted for when interpreting measured isotopic values of pool. This is because
the change in isotopic value (or solute concentration, $C$) over time ($\delta C/\delta t$) of any water body is a
function of the water balance and the mass balance (Cook 2013, Bourke et al., 2014b). Errors in the
water balance will therefore propagate through to errors in the interpretation of measured concentration
data.
This point can be demonstrated using a simple synthetic model of a pool with a surface area of 200 m$^2$
and a depth of 2 m. The change in pool volume and $\delta^{18}O$ over 112 days was simulated using the method
and parameter values of Bourke et al. (2021) for four different water balance configurations (Figure
13). In the first scenario, evapotranspiration is the only loss (or gain) term in the water balance.
Evaporation is set at 0.007 m d$^{-1}$ and results in isotopic fractionation; transpiration is set at 0.003 m d$^{-1}$
and there is no associated fractionation (total loss rate 0.01 md$^{-1}$). In the second scenario, groundwater
inflows of 2.88 m$^3$ d$^{-1}$ are added with the $\delta^{18}O$ of groundwater set at -7‰. In the third scenario,
groundwater inflow and outflow are implemented (as well as evapotranspiration) with a net

920 groundwater inflow of 2.88 $m^3$ $d^{-1}$ (groundwater inflow of 11.52 $m^3$ $d^{-1}$ and outflow of 8.64 of $m^3$ $d^{-1}$.

921 The simulated pool volumes are identical in scenarios 2 and 3, but the isotopic values are different

922 because the water balance equations are different, and therefore the equation for $dC/dt$ is different in

923 scenario 2, as comparted to scenario 3. Thus, if the pool water outflow term was neglected when

924 interpreting measured stable isotope values from the pool, the results would be inaccurate. And although

925 the difference in isotopic enrichment may be small under different water balance scenarios, the

926 cumulative impact could still be important hydro-ecologically (the associated error is 8-30% of initial

927 pool water balance in scenarios shown below). The fourth scenario has the same evapotranspiration and

928 groundwater inflow as scenario 2, but the pool geometry is changed so that the pool area is halved and

929 the depth is doubled (A = 100 $m^2$, d = 4 m). This scenario demonstrates that the isotopic evolution of

930 this hypothetical pool is very sensitive to the surface area to volume ratio of the pool; and yet these

931 geometric properties can be difficult to characterize and are rarely reported in published literature (they

932 are also are lacking in the case studies presented herein).

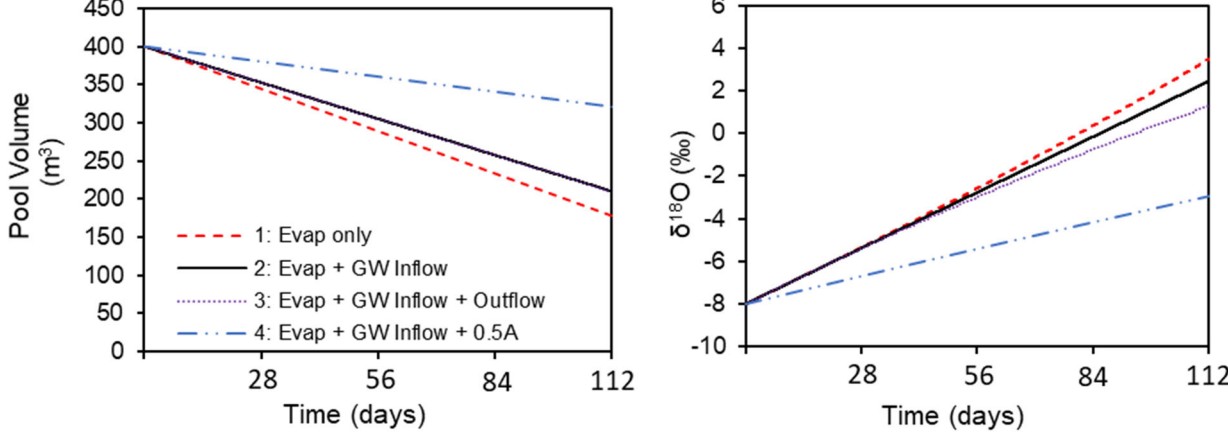


**934 Figure 13 Evolution of pool volume and values of stable isotopes of water in pools with varying water balance**

**935 components over approximately 4 months of dry season (Evap = Evapotranspiration, GW = groundwater, A = pool**

**936 area). Model modified after Bourke et al., (2021) to simulate a pool instead of a flowing stream.**


938 Such potential pitfalls can be found in all single methods. Thus, we suggest an initial regional-scale

939 assessment of landscape position and hydrogeological context that allows for pools to be grouped into

likely hydraulic mechanisms; a representative subset of these can be instrumented and sampled to
provide time series of water levels (groundwater and surface water) and hydrochemistry to understand
the pool water balance.
This was our approach in the Hammersley Basin and the above points can be seen in our results. We
were able to identify key hydraulic mechanisms supporting pool persistence at a number of pools at
regional and local scale. However, the saptio-temporally variable components of the water balance
remain difficult to constrain. Although there is a lot of data in the region overall, given the remote and
inaccessible nature of these pools, none of them have a complete data set of the kind advocated for here.
It should be noted that as with every field study, these case studies do not represent perfect examples
of the hypothetical cases but are instead limited by typical considerations found in the real world and
are subject to ongoing research efforts. In particular, Ben's Oasis provides an example of a pool that is
particularly difficult to characterise and cannot simply be linked to one hydraulic mechanism. Efforts
to characterise the bathymetry of Ben's Oasis have been fraught with challenges, and the relationship
between water level and pool volume remains uncertain, limiting our efforts to confidently determine
the water balance.
In this work, we have striven to provide a useful framework, based on a conceptual, first-principles
understanding, supported by both useful tools and case studies. However, this has also resulted in
limitations. Each of these topics could be presented as a full study. The list of field and regional-scale
methods is not exhaustive, but instead presents the most commonly used and accessible tools. Various
other tools, such as geophysical surveys and varied geochemical tracers, could easily be employed to
garner additional data useful in further understanding in-stream pool hydrology. Moreover, the review
of supporting literature, in particular from the field of groundwater-surface water interactions, has been
necessarily concise and more could be said. However, we feel that there is utility in presenting the basic
background in conjunction with field tools and considerations, allowing each reader can take the parts
that are most relevant to their own needs and seek out further background from the cited literature as
needed. We hope this work serves as a common platform for a deeper understanding of in-stream pools

globally, as non-perennial streams are increasingly recognised for both their importance and their vulnerability in our changing world.

The study of persistent river pools is a developing science and much remains to be done. Policy makers increasingly require accurate information on the mode of occurrence of surface water pools to put forward management plans to mitigate and/or minimise the adverse impacts of human activities (Leibowitz et al., 2008). This framework is subject to refinement as sufficient data becomes available to fully characterise pool water balances and mode of occurrence. Extension of this framework to facilitate the incorporation of biological and sedimentological processes is also desirable. Persistent river pools exist in all climates across the globe, and consistent data on geomorphology, hydrology and ecology should be collected at multiple features so that generalized patterns and processes can be elucidated. The nutrient and carbon transport between pools during flows and the effects of anthropogenic disruption to groundwater inputs or surface water flushes into these pools is also not well known. These disruptions can be detrimental to water quality if the anthropogenic inputs are contaminated (Jackson and Pringle, 2010), but may also support seasonal connectivity that benefits the ecosystem by distributing nutrients and organic matter between pools (Jaeger et al., 2014). Effects of climate change (e.g. lower groundwater levels, thermal loading, and altered storm cycles) also combine with geomorphological and biological factors to impact ecosystem function, but these mechanisms are not yet well understood.

## 7 Conclusion

Persistent pools are an important feature along non-perennial rivers and these types of systems are under increasing pressure from altered hydrology associated with shifting climates and anthropogenic activities (Steward et al., 2012). Three dominant hydraulic mechanisms that support the persistence of river pools were identified from literature on groundwater springs and groundwater - surface water interaction; perched surface water, through-flow of alluvial water, and regional groundwater discharge. Regional groundwater discharge can be further characterized into two types of control on groundwater

outflow; geological barrier vs topography. While the existing literature hints at the hydrologic and
geologic constraints imperative to pool persistence, the framework presented here provides cohesive
synthesis of hydraulic mechanisms supporting persistence, as required to sufficiently understand and
protect persistent river pools globally. Susceptibility to hydrological change depends on the
mechanism(s) of pool persistence and the spatial distribution of stressors relative to the pool. Further
research is required to resolve the impacts hydroclimatic stressors at the scale of individual pools.
A suite of diagnostic tools are available for understanding the hydrologic mechanisms that support the
persistence of a given river pool. A regional-scale assessment can be made based on an understanding
of the pool's landscape position and hydrogeological context, which may be supported by remote
sensing or image analysis. Time-series data of water levels and hydrochemistry are required to resolve
the spatiotemporal variability in pool water balances, as demonstrated in the three pool-scale case
studies presented. The suitability of each of these tools to any given pool or study will depend on the
data and resources available, and the requirement for a coarse or highly detailed resolution of the
mechanisms supporting pool persistence.

**Data Availability**
The data used in Section 5 of this paper are the property of Rio Tinto. Access to these data may be
requested by contacting Shawan Dogramaci (shawan.dogramaci@riotinto.com)

**Author Contribution**
SB and MS prepared the text of the manuscript with input from all co-authors. PH, SC, SD and SB
collected and analysed the data presented in Section 5. PH and SB prepared the figures.

**Competing Interests**
The authors declare that they have no conflict of interest.

**Acknowledgements**
This paper is based on data collection funded by Rio Tinto Iron Ore and funding from the Australian
Research Council, grant LP120100310. Author Shanafield's contribution was supported by funding
from the Australian Research Council, grant DE150100302. We thank the editor and our several
thorough reviewers, as well as the colleagues who have already found this framework useful and
encouraged us to finalise it.

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
