# Peer review of "A hydrological framework for persistent pools along non 2 perennial rivers"

_Hydrology and Earth System Sciences, 2021_

## Referee Comment (RC1)

[referee-annotated manuscript omitted]

---

## Community Comment (CC1)

**A hydrological framework for persistent river pools, Bourke et al., hess-2021-461, Leannah Sies**

The main objective of the study was to incorporate relevant literature on groundwater springs and surface-ground water interaction with a modern suite of diagnostic tools. This framework is meant to increase the understanding of the most important hydraulic mechanisms for the persistency of pools along non-perennial rivers and gain knowledge on their susceptibility to climate change and human activities. In accomplishing this, the hydraulic mechanisms that support river pools are classified as perched surface water, alluvial through flow and regional groundwater discharge controlled by geological features or topographic lows. A diagnostic tool suite to elucidate these hydraulic mechanism is presented. Landscape positioning, remote sensing, water balances and tracer techniques are discussed. Also, the susceptibility of each mechanism to climate change and human activities such as water abstraction is considered. Finally, the diagnostic tools presented are applied to three pools in the Hamersley basins in Australia to demonstrate their use and the difficulties related to their application in the real world.

As mentioned in the introduction of the paper persistent surface water pools along non-perennial rivers are an important water source for plants, animals an humans. However, their persistence is threatened by climate change and human activities such as water abstraction. In my opinion the authors did an interesting work by addressing this important but novel topic. I believe that creating a hydrological framework to understand the dominant hydraulic mechanisms supporting persistent river pools is relevant in effectively managing these pools and their cultural and ecological functions. Although the results are not surprising as they mostly summarize our existing knowledge, I believe the way it is presented is a valuable tool in future research. Because of the author's innovative hydrological look at river pools and the paper's contribution to the understanding of the hydrological system of river pools I believe the manuscript is appropriate for HESS. However, the message the paper tries to convey, the importance of river pool persistence and the usefulness of the diagnostic tool suite, is not as strong as could be. Therefore, some changes are needed to increase the readability and convincingness of the paper. Section 6, the conclusions and recommendations, need to be changed such that it better reflects the message conveyed by the rest of the paper. Besides this, I think some structural adaptations are needed and a stronger linkage between each section is necessary. Below, I have addressed these issues in more detail. After these are solved, I would recommend the manuscript for publication.

**Major Comments**

*Conclusions and recommendations*

The conclusions and recommendations in section 6 are well formulated and easy to read. However, I found that the conclusions and recommendations do not fully reflect the message that is conveyed in the rest of the paper. For example, in my opinion the fact that it has been 100 years since groundwater springs were documented is not important in this paper. Still, this is where the section starts with. In contrast, the dominant hydraulic mechanisms and their classification are not specifically mentioned whereas these are much more relevant in the scope of this paper. Also, the structure of the conclusions is not in line with the structure of the rest of the paper. For example, the conclusions on section 4, the susceptibility of the processes, is discussed before the diagnostic tool suite of section 3 is mentioned. Not only does section 6 not reflect the relevance of each topic properly, also new topics are addressed. For example, in the conclusions it is discussed whether more detailed data of less pools provides more insight than snapshots of many pools. This question is not introduced before and, in my opinion, distracts from the potential this paper has regarding the evaluation of the demonstrated diagnostic tool suite. Lastly, the recommendations done could use some more detail to better stress the use of this paper as stepping stone for future research. For example, it is mentioned that the biological and sedimentological processes need to be added to the framework. Why is this important? Are there hints this would massively change the systems known

from this framework? Furthermore, it is mentioned that this framework is subject to refinements as sufficient data becomes available. This makes me wonder, what is sufficient data? What does this mean for the current accuracy of the framework? To summarize, I believe that section 6 does not fully reflect the message of the paper as the focus is not solely on the relevant issues, its structure is different from the paper and the potential of the framework in identifying the knowledge gap of this research field is not worked out sufficiently.

I believe that due to these issues the conclusions drawn are less strong. Because the focus is not solely on the relevant issues the message conveyed by the paper is less clear and likely to have less impact. The lack of structure reduces the readability of the paper and, as mentioned before, the recommendations paragraph simply has more potential.

To solve these issues I would recommend changing the structure such that it is in line with the framework. Think of the objective of each of the sections written and reflect on whether this goal is reached. What is learned from this and what future research would you recommend? By doing this the readability is increased and it is easier for the authors to make sure everything is included and no new things are added without reason. If the readability remains an issue, I recommend adding a short introduction to the structure of the conclusion. As the framework in this paper contributes to understanding what knowledge gaps there are in this research field I would suggest stressing that these recommendations are  results, no shortcomings, of this framework. Also, I would suggest going into more detail. A question the authors may ask themselves to reach the appropriate level of detail is what the use of the framework is for the future research suggested? And also, what is the use of the future research for the framework? If the authors spend some more time on this I believe section 6 can from a strong and convincing  ending to this paper and can contribute to future management of river pools.

**Structural adaptations**

The structure of the paper is nicely introduced at the end of section 1. However, I found the structure of the paper illogical mainly due to the order of the sections. In section 4, as well as in section 2,  a more in depth understanding of the processes occurring is gained. However, already in section 3, tools are given to quantify these processes. I think this can be done better. An example of this can be found in a paper written by one of the co-authors and referred to several times. In "an overview of the hydrology of non-perennial rivers and streams"  first the processes are discussed (section 2,4), then common approaches and measuring challenges are highlighted (section 3), the hydraulic understanding is synthesized (in this situation the case studies in section 5) and finally future research directions are given (section 6).

I believe that the illogical structure reduces the readability of the paper. Also, I think this structure is one of the reasons the sections are rarely coupled (see "improve linkage between sections"). To solve this issue I suggest to simply switch section 3 and section 4.

**Improve linkage between each section**

Although I believe the paper is nicely separated in several sections, while reading the paper I found that the different sections are rarely linked to each other. This not only holds for section 6, the conclusions and recommendations but also for the other sections. For example, in section 3 the diagnostic tools are discussed. It is mentioned that these tools are used to distinguish the pools outlined in the previous section. However, I miss what information is needed to accomplish this. Also, I think the tools presented need to be better coupled to what is known from the previous section on the processes they elucidate. For example, for the first heading the focus is on remote sensing and landscape position. However, these methods are not related to the earlier made distinguishment between regional discharge controlled by geological features and regional discharge controlled by topographic lows. Not only is section 3 rarely coupled to the previous section, also the

coupling with the case studies is poorly done. For example, remote sensing is not used as a method in the case studies although it is discussed as part of the diagnostic tool suite. In contrast, a tracer technique with oxygen isotopes is used, but this method is not introduced before.

Because of this poor coupling of the sections, the paper comes across as less convincing than it could be. Also, especially for readers that aren't as experienced with the topic as the authors, this poor coupling causes the paper to be complex although this is definitely not needed. Furthermore, because of the poor coupling of especially section 3, I even doubt if the goal of the case studies is met. It is said, the goal is to demonstrate how the diagnostic tool suite infer the hydraulic mechanisms supporting pool resistance and to see the complexity of applying these methods in the real-world situation. Therefore, a coupling to a concrete diagnostic tool suite (section 3) inferring the hydraulic mechanism from section 2 and 5 is needed.

To solve this issue, I would suggest changing the overall approach. I recommend making a clear overview on what needs to be known in determining the dominant hydraulic mechanism and find the best diagnostic tools for each parameter/information. This might lead to a step-by-step plan on how to figure out what the most dominant hydraulic mechanism is and what the susceptibility of the pool is (of course this depends on the situation but this can be included). I advise making a table similar to table 1 but now for the diagnostic tools. This concrete diagnostic tool suite can then be applied to the case studies stepwise. If the conclusions on the mode of occurrence then agrees with what is found by Dogramaci the authors properly demonstrated the diagnostic tool suite worked. Furthermore, to better couple the sections I would recommend adding a paragraph to each section that links it to the previous section. By re-structuring as recommended in the previous paragraph, the linkage between the sections can be made stronger. I see the paper as follows:

- Section 1 is an introduction to why it is important that the river pools persist and this study is done.
- Section 2 gives insight in what hydraulic mechanism are important for this persistence.
- Section 3 gives insight in how these mechanisms/fluxes can be changed due to climate change or under a different water regime (the susceptibility of these fluxes).
- Section 4 starts with a paragraph on what fluxes/ mechanisms need to be quantified to understand the susceptibility of a pool after which a step-by-step plan is presented on how to access all the needed information while discussing the accuracy of the used methods.
- Section 5 applies the step-by-step plan on existing pools.
- Section 6 goes specifically deeper into the usefulness of this step-by-step plan as the end product of the paper and highlights several future research directions that will address missing information.

To summarize, to convey the message stronger and increase the readability of the paper section 6 needs to be changed such that it properly reflects the paper. Also, section 3 and 4 need to be switched giving the paper a more logic structure. Finally, the sections need to be stronger linked by adding a coupling paragraph and focussing on a more concrete diagnostic tool suite that links section 2 and 5 clearly to section 6.

**Minor comments**

*General comment 1: It is mentioned that the water balance of the pools is only considered after surface flows have ceased. That sounds good, however, shouldn't the period of time the pool is disconnected from the surface water play a key role in their susceptibility to changes? How long do the pools have to withstand on their own and when do they get a refill? This seems especially relevant in the case of perched surface water.*

*General comment 2: I found that the aims and objectives were not clearly stated. Because of this it is hard to figure what is the most important to the authors. By adding a sentence like " the aim of this*

*study is to contribute to the understanding of the persistency of river pools" and "the objectives of this study are to create a framework for understanding the hydraulic mechanisms supporting persistent river pools and demonstrate the use of this framework with help of three case studies." this can easily be solved. By adding these, the readability of the paper improves.*

*General comment 3: The classification of hydraulic mechanisms that are most important for the persistence of river pools is nicely done and may provide an important tool in future research. Also, table 1 adequately summarized the framework making it of good use.*

*General comment 4: Using the case studies to demonstrate the use of the tool suite is a interesting approach and does massively improve the applicability of the manuscript. An extra suggestion to make the case studies clearer is to be consistent in the use of arrows in the figures (8,9,10) explaining the direction of the flow.*

General comment 5- *This paper does not include a discussion. Because of this the completeness of the framework is not discussed. By adding a separate heading or paying more attention to this in section 6 the readers have no reason to question the completeness of the framework.*

*P1, title: The title is nicely formulated. The novel approach is stressed clear by the use of the word hydrological. The title makes clear that this paper is a framework including much of what is known. The use of the words "persistent river pools" tells us that the paper focusses on the persistence of pools along rivers. A small suggestion I'd like to make it to also include the non-perennial character of these rivers . An optional title could be "A hydrological framework of persistent pools along non-perennial rivers.*

*P1, line 24: As I understood it well, the case studies are also meant to see the difficulties of applying the diagnostic tool suite to a real situation. Consider adding this to be true to the readers.*

*P2, line 41: Consider adding "such as pumping/water abstraction" for the human activities.*

*P3, line 53: Is 'host to primary productivity" meant?*

*P12, line 257: extra "the" before "groundwater system".*

*P14 line 285-289: example might be to obvious to include.*

*P15, line 311:  It is stated that maps of geological contacts are readily available but information on hydraulic properties is not known in priori. Of course this depends on the location. However, if hydraulic properties are already known this Is probably a result of the method they described afterwards so a minor change in formulation would solve this.*

*P15, line 314: minor flaws in sentence structure.*

*P15, line 315-317: The information provided by deposition of geological precipitates (the occurrence of carbonates associated with ground water discharge) does not belong here as it is not directly linked to landscape positioning and/ or remote sensing. Instead I would put in under "tracer techniques" as in this subsection the geochemical properties of the water and streambeds are discussed.*

*P16, line 347: The reason why it is considered in absence of rain is missing. In my opinion rain can easily be included as ET is already considered. Also, in line 367 rain is included as refill for the alluvium water level.*

*P17, line 372: Make a clear distinguishment between regional ground water discharge, local groundwater discharge and alluvial through flow.*

*P17, line 379-383: The limitations of using the Darcy equations in a real world situation are nicely reported.*

*P18, line 401: Although I am acknowledged with the use of isotopes to quantify water sources I do not understand what is meant by the term "overlapping values". Therefore I would suggest changing this to "However, in a system where water is recirculated (irrigation, mining e.g.) stable isotopes have shown to be of limited use for constraining the relevant contribution of different recharge sources." (Bourke et al., 2015).*

*P18, line 405: minor flaws in sentence structure.*

*P19, line 426: "indicate" instead of "indicates".*

*P19, line 429: add "the" in front of $^{222}$Rn mass.*

*P21, line 471: "may" instead of "my".*

*P22, line 494: There is referred to (Cook et al., 2003). In the bibliography this paper is not found.*

*P25, line 534: Change "a subset of these (22 pools)" to "a subset (22 pools) of these" .*

*P25, line 534: Elaborate on why only 3 of the 22 pools are analysed. Do you think this suffices?*

*P25, line 551: change "permeability" to "permeable".*

*P28, line 578: claiming that the pools have not been impacted by human activities but considering climate change is, in my opinion, a contradiction. Specify the human activities (water abstraction etc.) to avoid this.*

*P29, line 603: "the that"*

*P35, line 732: measuring temperatures at dawn to reduce effect of direct solar radiation needs to be in section 3.*

*P38, line 771: include (Shanafield et al., 2021) as a source for frameworks of non-perennial streams.*

*P38, line 780-783: Not clear to me. Is it in addition to what was mentioned before on the novel hydrological approach?*

*P38, line 812: This paper deserves to end with a positive note.*

**Bibliography**

*Bourke, S. A., Cook, P. G., Dogramaci, S., and Kipfer, R. Partitioning sources of recharge in environments with groundwater recirculation using carbon-14 and CFC-12, Journal of Hydrology, 525, 418-428, 2015*

*Dogramaci S. Springs, pools and seeps in the Hamersley Basin, NW Australia, internal report for Rio Tinto Iron Ore, 2016*

*Shanafield, M., Bourke, S.A., Zimmer, M.A. Costigan, K.H. An overview of the hydrology of non-perennial rivers and streams. Wiley Interdisciplinary Reviews: Water, 8(2), e1504, 2021*

---

## Community Comment (CC2)

**Comment on hess-2021-461**

The paper "A hydrological framework for persistent river pools" by Sarah Bourke and colleagues presents a framework to classify the hydraulic mechanisms that support persistent pools. To do this, five mechanisms are defined: perched surface water, alluvial through-flow, groundwater outflow due to a regional aquifer pinching out, groundwater outflow at catchment constraints, and topography-controlled groundwater outflow. A suite of diagnostic tools is given and applied to the Hamersley basin (Australia) to demonstrate the framework can be used to distinguish between the different key hydraulic mechanisms. The susceptibility of the pools to groundwater withdrawals and climate shifts is determined and explained, as well.

As stated by the authors, there is a lack of understanding the mechanisms and water resources that support persistent river pools, which leads to limited effective water resource management. In my opinion, it is therefore relevant to set up a hydrological framework for persistent river pools as the hydrology is yet poorly understood. Especially in semi-arid regions it is important to know how the persistent pools are supported in order to predict the impact of changes in climate or groundwater abstraction.

The authors give a clear explanation of the different hydraulic mechanisms, and the accompanying figures help to understand them. In addition, the summary of the hydrological framework in Table 1 gives a direct overview of the different mechanisms. In the last section useful recommendations to improve the measurements and framework are listed. In general, the paper is clearly written and easy to understand, but the main focus is on the theorical framework itself instead of the application of it. These issues are explained below in more detail. Moreover, determining the susceptibility of the pools to improve effective water resource management seems to be the main reason to investigate their hydrology, but the explanation why this is important is missing in the introduction. Therefore, I would suggest the paper needs some major revisions and then a new review before it is approved.

The following major issues need to be considered.

(1)      Ideally, a paper introduces the topic, the importance of the research, and what will be investigated in the introduction. In their introduction, the authors state that a hydrological framework is needed that incorporates relevant literature, along with a modern suite of tools (lines 76-79). However, it is left out that these tools will be applied to identity the key supporting mechanisms in real world situations. This is a problem, because the authors want to set up a framework for persistent river pools that is also applicable in the real world, which is not mentioned in the introduction. Only in section 3 (lines 276-277) and section 5 (lines 512-514) is explained that these tools are needed to distinguish the key hydraulic mechanism(s) of a persistent pool. The consequence of not presenting the necessary applicability of the framework in the introduction, is that the application appears as an isolated part in the report. Since the aim of the paper is to understand the supporting mechanisms to improve effective water resource management (lines 47-48), it should be made clear in the introduction that the framework needs to be applicable to real world situations to identify the pools' supporting mechanism(s) and with that knowledge determine its susceptibility.

(2)      In section 5 the framework is applied to the Hamersley Basin, to see if the suite of tools can be used to identify the key hydraulic mechanism supporting a persistent pool. To show this, a subset of 22 pools is said to be investigated. However, only data of three of these is presented as case study. In the three case studies the data is explained and used to establish the dominant hydrologic mechanism supporting the pool. The 19 other pools are only mentioned at page 25 to make a general distribution of the mechanisms supporting pool persistence across the landscape. This is a problem, because no hard data is provided to prove that the five dominant mechanisms are all present in the Hamersley

Basin, as stated in Figure 5. Without this data, the reader cannot examine the measurement results and check the conclusions drawn by the authors. Moreover, the generalised explanation of the locations of the different type of pools appears as a repetition of section 2, mainly because no results are given to prove the statements made. As a result, the application does not convince the reader that the suite of tools can be used to distinguish all five different hydraulic mechanisms that support persistent pools. My recommendation is to expand the case studies with the data of the other 19 pools to at least show the five different supporting mechanisms. If similar data is present for different pools they can be compared and only then conclusions on the general location in the landscape can be drawn.

(3)      In the conclusion, the authors summarise the main hydraulic mechanisms supporting persistent pools and indicate the susceptibility to hydrological changes. However, the results of the application of the framework to the Hamersley Basin are not mentioned. Subsequently, as the conclusion is written now it focusses on the framework itself presenting it as being purely theoretical, while the application proves it to be practical, as well. Line 784 states that the presented suite of tools makes it possible to apply the framework to the real world. The authors explain the application of the framework in section 5, but do not mention this in the conclusion. The fact that the suite of tools can be applied to identify the different mechanisms, shows the value of the framework to determine the susceptibility which is a steppingstone to improve effective water resource management. Because of this, it should be made clear in the conclusion that the constructed framework and suite of tools have been shown to be applicable to real world situations.

In addition, the following issues should be considered.
- Section 2.3 divides groundwater influenced pools into two broad categories, however it is not specified based on what this decision made. Springer and Stevens (2009), as also referenced in this section, present twelve type of springs, but this already existing classification is not used in this new framework. So, please provide an explanation why this decision is made.
- In section 3.3 various tools are mentioned, but the methods of these tools are not given. Hamilton et al. (2005) and Siebers et al. (2016), for example, clearly explain the methods for $^{18}O$ isotope measurements and how to interpret the results. They also mention the locations of measurements, including the requirement to measure the isotopic composition in alluvial groundwater, which is not mentioned in the paper. Thus, please also provide the methods of the tools in section 3.3, including how to execute the measurements, which instruments to use, and which frequency is necessary for reliable results.
- The third paragraph of the case study of the Plunge Pool gives the conclusion, but it is not well substantiated. Why is the mechanism explained in section 2.3.1 not the key mechanisms in this case? Please give a better explanation based on what this conclusion was drawn. Actually indicating the position of the dykes in Figure 8c would assist in this explanation.
- Section 3.2 extensively explains different water balances per supporting mechanism, but these balances are not specifically mentioned or used in the application of the framework. Please use the water balance in the application or leave it out of section 3.
- The second paragraph of section 2.3.2 explains the difference between the two mechanisms; topography intersects regional aquifer and groundwater outflow at geological contacts. Please expand this explanation, as the difference is not clear to me now.
- The sentence "In the… aquifer (900 mS/cm)." in lines 602-604 on page 29 can be interpreted in two ways. Because the rainfall is not given in Figure 8b, "in the absence of rainfall" can reference to either a decreasing pool depth, and in these periods the EC increases (not

equilibrating around groundwater EC), OR to the moment shortly after a high pool depth is reached when the EC does gives values of approximately the groundwater EC. Please rewrite this sentence that it can be interpreted in only one way.

- The third paragraph of section 5.2.3 indicates measurements were done at the "top" of pool 1. Does this mean the upper part of the pool or the upstream part? Please elaborate if the measurements for the Radon-222 activity were done with a vertical or horizonal transect, or a combination.

List of minor issues:
- Page 5, lines 125-127 contain the sentence "For example…many pools". Please rewrite this sentence as it is lengthy and not formulated clearly.
- Page 14, line 299 mentions snapshot sampling. Please give a short explanation of what is meant with this.
- At page 15 the abbreviations NDVI, NDWI, and AEM are used. Please also give the full meaning of these abbreviations.
- Page 18, lines 404-407 contain the sentence "For example…flood event." Please rewrite this sentence as it is missing a verb and not formulated clearly. Also, put "value" before "re-equilibrated with the groundwater EC" instead of after it.
- Page 25, line 535 mentions the "mode of occurrence" of pools. Please specify what is meant with this, as I could not come up with a definition from the context.
- Page 27, Figure 6 gives the location of the persistent pools in the study area and the 22 pools chosen as subset, but it does not give the location of the three case studies specifically, which I think would be an addition to the map. Also, I think a digital elevation model (DEM) of the study area would be an addition since the topography of the area can determine the existence and persistence of pools. So, please indicate the location of the case studies on the map and add a DEM.
- Page 28, lines 586-587 mention the Marra Mamba Formation, but this subsurface layer is not indicated in Figure 8c. Please change the legend of Figure 8c to contain this name or change the name in the text to one of the layers that is mentioned in the legend. It could also be helpful to mention the hydraulic conductivity of the different formations, to give an indication of their permeability and the presence of aquicludes.
- Page 29 mentions the ADH. Please first give the meaning of this abbreviation, as this indication of a reference level is probably not known to non-Australians.
- Page 29, lines 592-595 mention the measured groundwater level at two monitoring wells. As their number is given in Figure 8c, it may be useful for the interpretation if the number is also mentioned in the text.
- Page 31, line 634 mentions the abbreviation BIF. Please give an explanation of this abbreviation or write it in full, since it is not used as abbreviation afterwards.
- Page 32, lines 658-659 mention "the water table drops below the pool" which references to Figure 8c, but the groundwater level is not given in this figure and cannot easily be compared to the water table in Figure 8b because of a different time scale. Please remove this part of the sentence as it cannot be checked now or add a graph of the water table to Figure 8c.
- Page 32, lines 673-675 state that a threshold groundwater level for inflow of groundwater to the pool can be found from the isotopic data. This cannot be concluded from the explanation of Figure 9d, as explained in the third paragraph in section 5.2.2. Please explain how this threshold value was found.

- Page 35, lines 710-711 states that pool 1 dries out as the dry season progressed, but the deeper parts of pool 1 persisted throughout the dry season. Please rewrite this sentence to resolve the contradiction for pool 1.

List of technical comments:
- Page 4, line 79: "mechanisms" instead of "mechanism"
- Page 10, line 198: "mechanisms" instead of "mechanism"
- Page 10, line 200: "purpose" instead of "purposes"
- Page 10, line 202: "Springer and Stevens (2009)" instead of "Springer (2009)"
- Page 15, line 315: add "as" before "a"
- Page 21, line 471: "may" instead of "my"
- Page 21, line 471: insert "the" before "subsurface"
- Page 29, line 598: move reference to Figure 8b to line 601 after "catchment"
- Page 29, line 601: add "period" after "dry"
- Page 29, line 603: substitute "that of the" instead of "the that of"
- Page 30, Figure 8c: What does "mRL" mean?
- Page 31, lines 650-651: move the closing bracket after "12[th] Nov 2018" to after "7[th] Dec 2018"
- Page 32, lines 665-672: move the sentence "Based on … (Fig. 9e)." to the end of the paragraph, so after the sentence "As a result ... this pool."
- Page 34, line 690: move the position of the reference of Figure 10a to after "Marra Mamba Formation" in line 693
- Page 37, Figure 10e: add a legend for the orange subsurface layer

References
Hamilton, S. K., Bunn, S. E., Thoms, M. C., & Marshall, J. C. (2005). Persistence of aquatic refugia between flow pulses in a dryland river system (Cooper Creek, Australia). *Limnology and Oceanography*, *50*(3), 743-754.
Siebers, A. R., Pettit, N. E., Skrzypek, G., Fellman, J. B., Dogramaci, S., & Grierson, P. F. (2016). Alluvial ground water influences dissolved organic matter biogeochemistry of pools within intermittent dryland streams. *Freshwater Biology*, *61*(8), 1228-1241.
Springer, A. E., & Stevens, L. E. (2009). Spheres of discharge of springs. *Hydrogeology Journal*, *17*(1), 83-93.

---

## Author Comment (AC1)

**HESS-2021-461: Response to Reviewer 1**

The manuscript entitled 'A hydrological framework for persistent river pools' by Sarah A. Bourke et al., propose a paper that describes a framework for characterizing the hydrology of semi-permanent river pools, as well as some examples of this kind of pools. Althoug I find interesting the overall idea of the manuscript, it is not adequate for publication in its present form.

Thank you for taking the time to review our manuscript, we appreciate your constructive comments and look forward to improving the manuscript in response to your review.

The description of the 'framework' (section 2) is rather overconfident, as this is more a revision of former descriptions than an original one.

While we agree with the reviewer that there are a small number of papers that mention some of the hydraulic mechanisms that sustain persistent river pools, these often have an ecological or management focus, and the treatment of hydrology is incomplete, flawed, or cites this manuscript under review. Therefore, there remains a need for this manuscript to be published as a rigorous hydrological synthesis of these different mechanisms so that future studies can be conducted in the context of a robust hydrological framework.

We thank the reviewer for introducing us to the Joque et al. (2010) paper freshwater on rock pools that we had not previously cited. As the title suggests, this paper describes the ecology of freshwater that persists over impermeable hard rock. There is a brief hydrological description (1 paragraph of hydrology, 1 paragraph of examples) within the section on "the rock pool habitat: definition and distribution". In the first paragraph authors mention that these features can be filled by precipitation, rivers and groundwater, but that the paper focusses on rain-fed rock pools which are the more typical freshwater habitat (presumably perched pools over impermeable bedrock). Thus, while identifying a relatively broad range of hydrological features that can exist (some of which may be within river channels, others which are not – gnammas for example), it does not detail the hydrological mechanisms that can support persistence of water in pools along rivers (groundwater discharge vs perched rainwater), which is the main focus of our present manuscript.

The reviewer also refers to Bonada et al. (2020), which is a paper on conservation and management of isolated pools in temporary rivers that we are aware of (and had cited). While this paper does provide a brief summary of hydrologic mechanisms that can support pools that is more rigorous than Joque et al. (2010), it cites the earlier version of our paper in HESS-D when doing so. As such, it is a circular argument to say that we are duplicating the work of Bonada (2020) given that they have applied the framework presented here in their manuscript. In revisiting the Bonada paper in response to this review, we have realized that we have not cited Leibowitz and Brooks (2008) chapter on vernal pools and will correct this omission in the revised manuscript. The 2008 chapter provides a summary of the water balance of pools that are not perched, which is consistent with the framework presented herein, but does not describe subsurface permeability features that control groundwater discharge.

The reviewer also directs us to Fellman et al., (2011), which we have discussed in the manuscript. This paper aims to characterize the hydrology of a particular set of pools as controls on dissolved organic matter biogeochemistry. While this manuscript does describe perched and alluvial through-flow pools along river channels, it does not robustly describe the hydrology of these features. It draws conclusions about the hydraulic mechanisms supporting pools based solely on stable isotope values of water (beginning and late dry-season), which are subject to uncertainty that has not been described. In their paper, the water balance of the pools is assumed to consist of inputs from rainfall and groundwater inflow and losses to evaporation. The calculation of evaporative loss from stable isotopic enrichment was made on the basis of a steady state model of evaporation divided by input (E/I). Perched and alluvial through-flow pools are then identified using this ratio (high E/I ratio implies perching). As such, although a subset of the pools are identified as through-flow pools, the conceptual model that underpins the analysis does not account for outflow of water from the pool back into the alluvium (Liebowitz and Brooks, 2008).

The stable isotopic enrichment of a pool with an initial volume of 400 m$^3$ can be simulated using the water balance equations presented in the current manuscript under review (Figure 1). The evolution of stable isotopic values is simulated using the approach of Bourke et al., (2021). A perched pool will have no inflow during the dry season and losses to ET only; a through-flow pool will have losses to ET, inflow of alluvial groundwater and loss via outflow (infiltration) of pool water back into the alluvium (ET + GW inflow + Outflow). For a perched pool with a volume of 400 m$^3$ at the beginning of the dry season, water volume over 112 days will reduce to 178 m$^3$ with $\delta^{18}$O enriching from -8 to 3.5 ‰. The addition of a groundwater inflow of 0.0002 m$^3$/min (0.3 m$^3$/d) results in similar end-point values (210 m$^3$ and $\delta^{18}$O of 2.4 ‰). In this example, using the line of thought presented in their paper, Fellman et al. would have concluded that groundwater in this second pool is not an important component of the water balance. However, over 112 days this groundwater inflow equates to 8% of the initial volume of the pool and may be important for hydrochemical parameters in the alluvial water (or regional groundwater) that have different values than the pool water. Furthermore, alluvial through-flow pools will usually have water losses associated with infiltration to the streambed sediments, which Fellman did not account for. Thus, the inflow of groundwater may be larger than otherwise thought, if it is balanced by infiltration from the pool of a similar magnitude. For example, groundwater inflow of 0.0008 m$^3$/min balanced by outflow via infiltration of 0.0006 m$^3$/min will result in the same pool water level as a groundwater inflow of 0.0002 m$^3$/min, but the isotopic enrichment will be slightly smaller ($\delta^{18}$O of 1.3 ‰). Over 112 days, the groundwater inflow in this third scenario adds up to 128 m$^3$, or 32% of the initial pool volume. A fourth scenario where the water balance is consistent with Fellman (ET and GW inflow terms are as per scenario 2), but the pool area is halved (initial volume remains the same) demonstrates that the water balance of the pool and stable isotopic enrichment are sensitive to the pool geometry (volume to area ratio), which Fellman et al. did not report on or consider explicitly in their analysis. Thus, the identification of hydraulic mechanism supporting pools was made on the basis of unsupported assumptions about pool water balances.

Their analysis approach, based on an incomplete water balance, led Fellman to conclude that many of the pools studied were isolated from the alluvium water table, but this conceptualization (see their Fig 1) is not hydrogeologically robust. All but one of the pools in their paper occur on permeable alluvial sediments with pools 1-12 shown overlying a similar thickness of alluvium. If the pools were not connected to the alluvial (and/or regional) water table, without the presence of a low-permeability layer beneath these pools, the pool water would infiltrate into the alluvium (Brunner et al., 2009) and

the pool would not persist. Thus, the inset diagram of "pools isolated from alluvium water table" is hydraulically implausible (as already discussed in the manuscript).

[Figure]

*Figure 1 Evolution of pool volume and values of stable isotopes of water in pools with varying water balance components over approximately 4 months of dry season (ET = evapotranspiration, GW = groundwater, A = pool area). Model modified after (Bourke et al., 2021)*

Sections 2, 3 and 4 are is too descriptive, too long and repetitive, the equations are obvious and the figures are of poor quality. Most of this part could be synthesized in the table 1 with appropriate references and some auxiliary text like that in section 5.1.

We are glad that the reviewer finds Table 1 useful. While the water balance equations may appear obvious, existing literature does not adequately or robustly describe the water balance of river pools (see discussion above), and so we feel that it is important that these are explicitly presented and explained so that water balances can be accurately accounted for in future studies. Similarly, hydrologic concepts that we may take for granted are often used or interpreted differently by practitioners in related fields. The reviewers' comment that the hyporheic zone as an ecotone or habitat relevant for aquatic life provides a great example of this. While this is true from an ecological perspective (Stubbington, 2012), there is also an extensive susbset of hydrology related to the hyporheic zone that focusses not on ecological properties, but on the scales and mechanisms of hydraulic fluxes, which are driven by *streamflow* and channel morphology (e.g. Stonedahl et al., 2010, Bourke et al., 2014). Thus, when the stream is not flowing, these in-and-out hyporheic exchange fluxes are not operating. In this manuscript (and others, e.g. Leibowitz and Brooks 2008), alluvial water is treated hydraulically as a groundwater storage, with fluxes from the capture zone into the pool considered groundwater inflow, and outflow via infiltration (to the release zone) back into the alluvial groundwater. These fluxes are driven by the hydraulic gradients between the pool and the alluvial groundwater and are not related to streamflow. Conceptually, this hydraulic exchange is most accurately described as analogous to the well-established concept of through-flow lakes found in literature on surface water – groundwater interaction (Winter et al., 1998). While this surface water – groundwater exchange process seems clearly distinct from the relatively short-time scale fluxes of hyporheic exchange associated with streambed contours, we can see that the distinction between this and longer timescale parafluvial flows may be unclear, particularly for non-perennial streams (Del Vecchia et al., under review), so we will revisit this text during revision to attempt to improve clarity for the reader.

With regard to Figures 1-4 in Section 5.1, these are presented as generalized conceptual diagrams of the hydraulic mechanisms that can support persistent river pools. Although we want to be geologically and geomorphologically plausible, they are not intended to represent particular settings or landscapes and are not to scale (this will be specified in the revised manuscript). These figures were always intended to be non-site-specific conceptual diagrams (even in the first submission of this manuscript they did not represent the settings of specific pools), and are broadly consistent with other published diagrams of incised river valleys and floodplains (e.g. Hayes et al. 2018). As this manuscript has a hydrological focus, we have drawn these figures so as to allow us to demonstrate the hydraulic processes that we are discussing, consistent with our experience of, primarily, Australia, but also North America. Some of these processes may not be obvious or common in geologically younger landscapes, or more humid climates, and we will endeavour to clarify and refine these figures in response to the reviewers' comments where possible.

In determining the geometries and labels used in these figures we consulted with colleagues who specialize in geology and geomorphology. We received a range of responses, from which we chose those that we thought were simplest, and most effective at conveying the hydraulic processes we were describing to a broad audience (for which this paper is intended). We thank the reviewer for highlighting the inconsistency in the implied permeability of the bedrock, we will be sure to avoid this in the revised manuscript. Similarly, the lack of a defined surface drainage line between the hillslope and river in Fig 1 will be rectified. The reviewer has suggested "Alluvium" as a replacement for Alluvial channel – we are not sure of the basis for this suggestion, but are happy to make the replacement. We have used the term "valley-fill" to refer to any sediments within the geological river channel (as distinct from the flowing channel that a hydrologist may consider) and do not intend this to make any reference to a particular age of sedimentary deposition – hydraulically, the time of sediment deposition is not of primary importance. Unfortunately, a more suitable alternative has not been suggested by the reviewer and no supporting citations for this comment were provided so we are unable to determine a suitable replacement term. The reviewer also suggests that the water table in Fig 2b is too far from the surface of the floodplain. This figure represents the case of a perched water table beneath a river that resides in an arid or semi-arid climate where the regional water table can be tens of metres below the surface (Villeneuve et al., 2015). Perhaps the reviewer is suggesting that the regional water table should be within the flood plain rather than the bedrock? If so, this point is well taken and we will revisit this diagram during revision to ensure it is consistent with our understanding of the hydrology of these systems. The reviewer has also made a comment about the lower boundary of the aquifer in Fig 3b. This figure depicts the generalized case where valley fill sediments are relatively thin and the lower boundary of the regional aquifer is determined by the lower boundary of weathering in the bedrock, which is hydraulically connected to the valley fill. It is unfortunately not clear what issue the reviewer has with this depiction, which is consistent with our experience.

In my opinion, section 5.2 is of value and deserve publication if some aspects are improved. Mostly, the paper should be readable for everybody not used with Australian geologic units, map coordinates and elevation datum.

This section was added in response to reviewer comments on the previous submission of this work and we are pleased that this reviewer finds value in it. We can very easily ensure all coordinates are standard map grids. Presumably the reviewer would be more comfortable with elevations in meters

above sea level (m asl), which is equivalent to m AHD (Australian Height Datum) that we had used and we will update the revised manuscript accordingly.

The map in Figure 6 should represent more information than just the location of unknown pools and the figures should be of better quality.
Thank you for the useful suggestions. We will add towns to this map (e.g. Tom Price) during revision. The grid coordinates shown are standard UTM values for Zone 50K, which are used by Google Earth, a statement of this and a north arrow (up) will be added to the revised figure. The pools used as case studies will also be identified.

The assumptions and interpretations should be better separated from observations.
Each of these case studies is currently structured as beginning with a description of the hydro(geo)logical setting, followed by the data collected and the resulting interpretation of mechanisms supporting pool persistence, and finally the implications for management. Perhaps sub-headings would make this clearer? The reviewer has not provided any specific guidance on how to improve the structure.

Section 6 is rather a discussion than a conclusion, but some discussion is necessary not for showing the interest of 'framework' but for identifying research gaps and further research goals, not necesarily using heavy instrumentation.
We are happy to work on this section during revision and can easily present a separate conclusion rather than the combined section currently presented. We are intrigued by the reviewers' assertion that it is only in intricate places that extensive instrumentation and multiple data sets are required to determine the hydraulic mechanisms supporting persistent river pools. This has not been our experience; perhaps persistent pools on Australian rivers are exceptionally complex? We would have been grateful for any specific papers the reviewer could suggest that could give us insight into how we can robustly understand the hydrology of river pools using a simplified (and therefore cheaper and less time-consuming) approach; however, none were provided

Many detailed comments are annotated in the manuscript. Please also note the supplement to this comment: https://hess.copernicus.org/preprints/hess-2021-461/hess-2021-461-RC1-supplement.pdf

Further response to individual comments in the supplement provided will be made when submitting the revised manuscript.

**References**

Bonada, N., Cañedo-Argüelles, M., Gallart, F., von Schiller, D., Fortuño, P., Latron, J., Llorens, P., Murria, C., Soria, M., Vinyoles, D., Cid, N. Conservation and management of isolated pools in temporary rivers, Water, 12 (10) 2870, https://doi.org/10.3390/w12102870, 2020.

Bourke, S.A., Cook, P.G., Shanafield, M., Dogramaci, S. and Clark, J.F., 2014. Characterisation of hyporheic exchange in a losing stream using radon-222. Journal of hydrology, 519, pp.94-105.

Bourke, S.A., Degens, B., Searle, J., Tayer, T., Rothery, J. Geological permeability controls streamflow generation in a remote, ungauged, semi-arid drainage system, Journal of Hydrology: Regional Studies, 38. 2021. https://doi.org/10.1016/j.ejrh.2021.100956

Brunner, P., Cook, P., and Simmons, C. Hydrogeologic controls on disconnection between surface water and groundwater, Water Resources Research, 45, W01422, 2009.

Del Vecchia, A, Shanafield, M., Zimmer, M., Datry, T., et al. Reconceptualizing the hyporheic zone of non-perennial rivers and streams. Submitted to Freshwater Science, June 2021.

Fellman, J. B., Dogramaci, S., Skrzypek, G., Dodson, W., and Grierson, P. F. Hydrologic control of dissolved organic matter biogeochemistry in pools of a subtropical dryland river, Water Resour. Res., 47, W06501, 10.1029/2010wr010275, 2011.

Hayes, Daniel & Braendle, Julia & Seliger, Carina & Zeiringer, Bernhard & Ferreira, Maria & Schmutz, Stefan. Advancing towards functional environmental flows for temperate floodplain rivers. Science of The Total Environment. 633. 1089–1104. 10.1016/j.scitotenv.2018.03.221. 2018.

Jocque, M., Vanschoenwinkel, B. and Brendonck, L.U.C., 2010. Freshwater rock pools: a review of habitat characteristics, faunal diversity and conservation value. Freshwater Biology, 55(8), pp.1587-1602.

Leibowitz, S.G.; Brooks, R.T. Hydrology and landscape connectivity of vernal pools. In Science and Conservation of Vernal Pools in Northeastern North America; Calhoun, A.J.K., deMaynadier, P.G., Eds.; CRC Press: Boca Raton, FL, USA, 2008; pp. 31–53.

Stonedahl, S.H., Harvey, J.W., Wörman, A., Salehin, M. and Packman, A.I., 2010. A multiscale model for integrating hyporheic exchange from ripples to meanders. Water Resources Research, 46(12).

Stubbington, R., The hyporheic zone as an invertebrate refuge; a review of variability in space, time, taxa and behaviour. Marine and Freshwater Research, 63, 294-311, 2012.

Villeneuve, S., Cook, P.G., Shanafield, M., Wood, C. White, N. Groundwater recharge via infiltration through an ephemeral riverbed, central Australia. Journal of Arid Environments, 117, 47-58, 2015.

Winter, T.C., Harvey,J.W., Franke, O.L., Alley, W.M. Groundwater and surface water: a single resource. USGS Circular 1139, 1998.

---

## Author Comment (AC4)

**A hydrological framework for persistent river pools, Bourke et al., hess-2021-461, Leannah Sies**

The main objective of the study was to incorporate relevant literature on groundwater springs and surface-ground water interaction with a modern suite of diagnostic tools. This framework is meant to increase the understanding of the most important hydraulic mechanisms for the persistency of pools along non-perennial rivers and gain knowledge on their susceptibility to climate change and human activities. In accomplishing this, the hydraulic mechanisms that support river pools are classified as perched surface water, alluvial through flow and regional groundwater discharge controlled by geological features or topographic lows. A diagnostic tool suite to elucidate these hydraulic mechanism is presented. Landscape positioning, remote sensing, water balances and tracer techniques are discussed. Also, the susceptibility of each mechanism to climate change and human activities such as water abstraction is considered. Finally, the diagnostic tools presented are applied to three pools in the Hamersley basins in Australia to demonstrate their use and the difficulties related to their application in the real world.

As mentioned in the introduction of the paper persistent surface water pools along non-perennial rivers are an important water source for plants, animals an humans. However, their persistence is threatened by climate change and human activities such as water abstraction. In my opinion the authors did an interesting work by addressing this important but novel topic. I believe that creating a hydrological framework to understand the dominant hydraulic mechanisms supporting persistent river pools is relevant in effectively managing these pools and their cultural and ecological functions. Although the results are not surprising as they mostly summarize our existing knowledge, I believe the way it is presented is a valuable tool in future research. Because of the author's innovative hydrological look at river pools and the paper's contribution to the understanding of the hydrological system of river pools I believe the manuscript is appropriate for HESS. However, the message the paper tries to convey, the importance of river pool persistence and the usefulness of the diagnostic tool suite, is not as strong as could be. Therefore, some changes are needed to increase the readability and convincingness of the paper. Section 6, the conclusions and recommendations, need to be changed such that it better reflects the message conveyed by the rest of the paper. Besides this, I think some structural adaptations are needed and a stronger linkage between each section is necessary. Below, I have addressed these issues in more detail. After these are solved, I would recommend the manuscript for publication.

*Thank you for taking the time to review our manuscript and making constructive comments to help us improve it.*

**Major Comments**

*Conclusions and recommendations*

The conclusions and recommendations in section 6 are well formulated and easy to read. However, I found that the conclusions and recommendations do not fully reflect the message that is conveyed in the rest of the paper. For example, in my opinion the fact that it has been 100 years since groundwater springs were documented is not important in this paper. Still, this is where the section starts with. In contrast, the dominant hydraulic mechanisms and their classification are not specifically mentioned whereas these are much more relevant in the scope of this paper. Also, the structure of the conclusions is not in line with the structure of the rest of the paper. For example, the conclusions on section 4, the susceptibility of the processes, is discussed before the diagnostic tool suite of section 3 is mentioned. Not only does section 6 not reflect the relevance of each topic properly, also new topics are addressed. For example, in the conclusions it is discussed whether more detailed data of less pools provides more insight than snapshots of many pools. This question is not introduced before and, in my opinion, distracts from the potential this paper has regarding the evaluation of the demonstrated diagnostic tool suite. Lastly, the recommendations done could use some more detail to better stress the use of this paper as stepping stone for future research. For example, it is mentioned that the biological and sedimentological processes need to be added to the

framework. Why is this important? Are there hints this would massively change the systems known

from this framework? Furthermore, it is mentioned that this framework is subject to refinements as sufficient data becomes available. This makes me wonder, what is sufficient data? What does this mean for the current accuracy of the framework? To summarize, I believe that section 6 does not fully reflect the message of the paper as the focus is not solely on the relevant issues, its structure is different from the paper and the potential of the framework in identifying the knowledge gap of this research field is not worked out sufficiently.

I believe that due to these issues the conclusions drawn are less strong. Because the focus is not solely on the relevant issues the message conveyed by the paper is less clear and likely to have less impact. The lack of structure reduces the readability of the paper and, as mentioned before, the recommendations paragraph simply has more potential.

To solve these issues I would recommend changing the structure such that it is in line with the framework. Think of the objective of each of the sections written and reflect on whether this goal is reached. What is learned from this and what future research would you recommend? By doing this the readability is increased and it is easier for the authors to make sure everything is included and no new things are added without reason. If the readability remains an issue, I recommend adding a short introduction to the structure of the conclusion. As the framework in this paper contributes to understanding what knowledge gaps there are in this research field I would suggest stressing that these recommendations are results, no shortcomings, of this framework. Also, I would suggest going into more detail. A question the authors may ask themselves to reach the appropriate level of detail is what the use of the framework is for the future research suggested? And also, what is the use of the future research for the framework? If the authors spend some more time on this I believe section 6 can from a strong and convincing ending to this paper and can contribute to future management of river pools.

*We acknowledge that there are weaknesses in the conclusion as written. Thank you for these constructive suggestions about ways in which we can improve the conclusions, we will refine the conclusion in response during the manuscript revision.*

**Structural adaptations**

The structure of the paper is nicely introduced at the end of section 1. However, I found the structure of the paper illogical mainly due to the order of the sections. In section 4, as well as in section 2, a more in depth understanding of the processes occurring is gained. However, already in section 3, tools are given to quantify these processes. I think this can be done better. An example of this can be found in a paper written by one of the co-authors and referred to several times. In "an overview of the hydrology of non-perennial rivers and streams" first the processes are discussed (section 2,4), then common approaches and measuring challenges are highlighted (section 3), the hydraulic understanding is synthesized (in this situation the case studies in section 5) and finally future research directions are given (section 6).

I believe that the illogical structure reduces the readability of the paper. Also, I think this structure is one of the reasons the sections are rarely coupled (see "improve linkage between sections"). To solve this issue I suggest to simply switch section 3 and section 4.

*Thanks for the suggestion to switch the order of these two sections, we are happy to do this during revision.*

**Improve linkage between each section**

Although I believe the paper is nicely separated in several sections, while reading the paper I found that the different sections are rarely linked to each other. This not only holds for section 6, the

conclusions and recommendations but also for the other sections. For example, in section 3 the diagnostic tools are discussed. It is mentioned that these tools are used to distinguish the pools outlined in the previous section. However, I miss what information is needed to accomplish this. Also, I think the tools presented need to be better coupled to what is known from the previous section on the processes they elucidate. For example, for the first heading the focus is on remote sensing and landscape position. However, these methods are not related to the earlier made distinguishment between regional discharge controlled by geological features and regional discharge controlled by topographic lows. Not only is section 3 rarely coupled to the previous section, also the coupling with the case studies is poorly done. For example, remote sensing is not used as a method in the case studies although it is discussed as part of the diagnostic tool suite. In contrast, a tracer technique with oxygen isotopes is used, but this method is not introduced before.

*While we are keen to ensure that the manuscript flows nicely the specific issues described here are not entirely clear to us. For example, the text in 3.1 on remote sensing and landscape position does explicitly describe the ways in which these techniques may (and may not) help you elucidate the hydraulic mechanisms outlined in Section 2.*

*In the case study presented in Section 5 it is not our intention to demonstrate all of the possible techniques, but the techniques we present are discussed within Section 3. Stable isotopes of water are discussed in Section 3.3. Perhaps the reader was not aware that 18O is a stable isotope of water, and so we will clarify this during revisions.*

Because of this poor coupling of the sections, the paper comes across as less convincing than it could be. Also, especially for readers that aren't as experienced with the topic as the authors, this poor coupling causes the paper to be complex although this is definitely not needed. Furthermore, because of the poor coupling of especially section 3, I even doubt if the goal of the case studies is met. It is said, the goal is to demonstrate how the diagnostic tool suite infer the hydraulic mechanisms supporting pool resistance and to see the complexity of applying these methods in the real-world situation. Therefore, a coupling to a concrete diagnostic tool suite (section 3) inferring the hydraulic mechanism from section 2 and 5 is needed.

*We believe that the case studies do demonstrate the inference of hydraulic mechanisms supporting pools (regional groundwater and alluvial water) based on a subset of the methods described in the paper (water levels, EC, stable isotopes of water, temperature, radon). We can see that there is room to be more explicit in attributing the role of topographic lows vs barriers in driving regional groundwater discharge and we will try to better highlight this distinction during review.*

To solve this issue, I would suggest changing the overall approach. I recommend making a clear overview on what needs to be known in determining the dominant hydraulic mechanism and find the best diagnostic tools for each parameter/information. This might lead to a step-by-step plan on how to figure out what the most dominant hydraulic mechanism is and what the susceptibility of the pool is (of course this depends on the situation but this can be included). I advise making a table similar to table 1 but now for the diagnostic tools. This concrete diagnostic tool suite can then be applied to the case studies stepwise. If the conclusions on the mode of occurrence then agrees with what is found by Dogramaci the authors properly demonstrated the diagnostic tool suite worked. Furthermore, to better couple the sections I would recommend adding a paragraph to each section that links it to the previous section.

*The approach suggested here seems to imply that there is a "truth" about the hydrological function of these pools that was previously elucidated by Dogramaci 2016, but this is not the case. The description of Dogramaci 2016 was based on the best understanding available at the time. With new data this understanding is sometimes revised and updated.*

By re-structuring as recommended in the previous paragraph, the linkage between the sections can be made stronger. I see the paper as follows:

- Section 1 is an introduction to why it is important that the river pools persist and this study is done.
- Section 2 gives insight in what hydraulic mechanism are important for this persistence.
- Section 3 gives insight in how these mechanisms/fluxes can be changed due to climate change or under a different water regime (the susceptibility of these fluxes).
- Section 4 starts with a paragraph on what fluxes/ mechanisms need to be quantified to understand the susceptibility of a pool after which a step-by-step plan is presented on how to access all the needed information while discussing the accuracy of the used methods.
- Section 5 applies the step-by-step plan on existing pools.
- Section 6 goes specifically deeper into the usefulness of this step-by-step plan as the end product of the paper and highlights several future research directions that will address missing information.

To summarize, to convey the message stronger and increase the readability of the paper section 6 needs to be changed such that it properly reflects the paper. Also, section 3 and 4 need to be switched giving the paper a more logic structure. Finally, the sections need to be stronger linked by adding a coupling paragraph and focussing on a more concrete diagnostic tool suite that links section 2 and 5 clearly to section 6.

*As per our above comment we have no problem swapping the order of 3 and 4 and working to improve the conclusion during revision.*

**Minor comments**

*General comment 1:* *It is mentioned that the water balance of the pools is only considered after surface flows have ceased. That sounds good, however, shouldn't the period of time the pool is disconnected from the surface water play a key role in their susceptibility to changes? How long do the pools have to withstand on their own and when do they get a refill? This seems especially relevant in the case of perched surface water.*

*While we agree that the duration without surface water inflows may be important for the pool ecosystem we consider this to be outside the scope of this manuscript, which is focussed specifically on the hydraulic mechanisms that support pool persistence in the absence of surface inflows.*

*General comment 2:* *I found that the aims and objectives were not clearly stated. Because of this it is hard to figure what is the most important to the authors. By adding a sentence like " the aim of this*

study is to contribute to the understanding of the persistency of river pools" and "the objectives of this study are to create a framework for understanding the hydraulic mechanisms supporting persistent river pools and demonstrate the use of this framework with help of three case studies." this can easily be solved. By adding these, the readability of the paper improves.

*The final paragraph of the Introduction outlines the aim and objectives of the study (simply swap "Here we"….with "The aim of this study is"…). This seems a question of writing style but if the editor feels that these specific words should be used then was can do so.*

General comment 3: The classification of hydraulic mechanisms that are most important for the persistence of river pools is nicely done and may provide an important tool in future research. Also, table 1 adequately summarized the framework making it of good use.

*Thank you.*

General comment 4: Using the case studies to demonstrate the use of the tool suite is a interesting approach and does massively improve the applicability of the manuscript. An extra suggestion to make the case studies clearer is to be consistent in the use of arrows in the figures (8,9,10) explaining the direction of the flow.

*Fair enough, can do.*

General comment 5- This paper does not include a discussion. Because of this the completeness of the framework is not discussed. By adding a separate heading or paying more attention to this in section 6 the readers have no reason to question the completeness of the framework.

*The point is well taken that the concluding remarks should be revised.*

*Minor comments below will be addressed during manuscript revision.*

P1, title: The title is nicely formulated. The novel approach is stressed clear by the use of the word hydrological. The title makes clear that this paper is a framework including much of what is known. The use of the words "persistent river pools" tells us that the paper focusses on the persistence of pools along rivers. A small suggestion I'd like to make it to also include the non-perennial character of these rivers . An optional title could be "A hydrological framework of persistent pools along non-perennial rivers.

P1, line 24: As I understood it well, the case studies are also meant to see the difficulties of applying the diagnostic tool suite to a real situation. Consider adding this to be true to the readers.

P2, line 41: Consider adding "such as pumping/water abstraction" for the human activities.

P3, line 53: Is 'host to primary productivity" meant?

P12, line 257: extra "the" before "groundwater system".

P14 line 285-289: example might be to obvious to include.

P15, line 311: It is stated that maps of geological contacts are readily available but information on hydraulic properties is not known in priori. Of course this depends on the location. However, if hydraulic properties are already known this Is probably a result of the method they described afterwards so a minor change in formulation would solve this.

P15, line 314: minor flaws in sentence structure.

P15, line 315-317: The information provided by deposition of geological precipitates (the occurrence of carbonates associated with ground water discharge) does not belong here as it is not directly linked to landscape positioning and/ or remote sensing. Instead I would put in under "tracer

*techniques" as in this subsection the geochemical properties of the water and streambeds are discussed.*

*P16, line 347: The reason why it is considered in absence of rain is missing. In my opinion rain can easily be included as ET is already considered. Also, in line 367 rain is included as refill for the alluvium water level.*

*P17, line 372: Make a clear distinguishment between regional ground water discharge, local groundwater discharge and alluvial through flow.*

*P17, line 379-383: The limitations of using the Darcy equations in a real world situation are nicely reported.*

*P18, line 401: Although I am acknowledged with the use of isotopes to quantify water sources I do not understand what is meant by the term "overlapping values". Therefore I would suggest changing this to "However, in a system where water is recirculated (irrigation, mining e.g.) stable isotopes have shown to be of limited use for constraining the relevant contribution of different recharge sources." (Bourke et al., 2015).*

*P18, line 405: minor flaws in sentence structure.*

*P19, line 426: "indicate" instead of "indicates".*

*P19, line 429: add "the" in front of $^{222}$Rn mass.*

*P21, line 471: "may" instead of "my".*

*P22, line 494: There is referred to (Cook et al., 2003). In the bibliography this paper is not found.*

*P25, line 534: Change "a subset of these (22 pools)" to "a subset (22 pools) of these" .*

*P25, line 534: Elaborate on why only 3 of the 22 pools are analysed. Do you think this suffices?*

*P25, line 551: change "permeability" to "permeable".*

*P28, line 578: claiming that the pools have not been impacted by human activities but considering climate change is, in my opinion, a contradiction. Specify the human activities (water abstraction etc.) to avoid this.*

*P29, line 603: "the that"*

*P35, line 732: measuring temperatures at dawn to reduce effect of direct solar radiation needs to be in section 3.*

*P38, line 771: include (Shanafield et al., 2021)  as a source for frameworks of non-perennial streams.*

*P38, line 780-783: Not clear to me. Is it in addition to what was mentioned before on the novel hydrological approach?*

*P38, line 812: This paper deserves to end with a positive note.*

**Bibliography**

*Bourke, S. A., Cook, P. G., Dogramaci, S., and Kipfer, R. Partitioning sources of recharge in environments with groundwater recirculation using carbon-14 and CFC-12, Journal of Hydrology, 525, 418-428, 2015*

*Dogramaci S. Springs, pools and seeps in the Hamersley Basin, NW Australia, internal report for Rio Tinto Iron Ore, 2016*

*Shanafield, M., Bourke, S.A., Zimmer, M.A. Costigan, K.H. An overview of the hydrology of non-perennial rivers and streams. Wiley Interdisciplinary Reviews: Water, 8(2), e1504, 2021*

---

## Author Comment (AC5)

**Comment on hess-2021-461**

The paper "A hydrological framework for persistent river pools" by Sarah Bourke and colleagues presents a framework to classify the hydraulic mechanisms that support persistent pools. To do this, five mechanisms are defined: perched surface water, alluvial through-flow, groundwater outflow due to a regional aquifer pinching out, groundwater outflow at catchment constraints, and topography-controlled groundwater outflow. A suite of diagnostic tools is given and applied to the Hamersley basin (Australia) to demonstrate the framework can be used to distinguish between the different key hydraulic mechanisms. The susceptibility of the pools to groundwater withdrawals and climate shifts is determined and explained, as well.

As stated by the authors, there is a lack of understanding the mechanisms and water resources that support persistent river pools, which leads to limited effective water resource management. In my opinion, it is therefore relevant to set up a hydrological framework for persistent river pools as the hydrology is yet poorly understood. Especially in semi-arid regions it is important to know how the persistent pools are supported in order to predict the impact of changes in climate or groundwater abstraction.

The authors give a clear explanation of the different hydraulic mechanisms, and the accompanying figures help to understand them. In addition, the summary of the hydrological framework in Table 1 gives a direct overview of the different mechanisms. In the last section useful recommendations to improve the measurements and framework are listed. In general, the paper is clearly written and easy to understand, but the main focus is on the theorical framework itself instead of the application of it. These issues are explained below in more detail. Moreover, determining the susceptibility of the pools to improve effective water resource management seems to be the main reason to investigate their hydrology, but the explanation why this is important is missing in the introduction. Therefore, I would suggest the paper needs some major revisions and then a new review before it is approved.

*Thank you for taking the time to review our manuscript and give detailed comments to help us improve it during revision.*

The following major issues need to be considered.

(1)     Ideally, a paper introduces the topic, the importance of the research, and what will be investigated in the introduction. In their introduction, the authors state that a hydrological framework is needed that incorporates relevant literature, along with a modern suite of tools (lines 76-79). However, it is left out that these tools will be applied to identity the key supporting mechanisms in real world situations. This is a problem, because the authors want to set up a framework for persistent river pools that is also applicable in the real world, which is not mentioned in the introduction. Only in section 3 (lines 276-277) and section 5 (lines 512-514) is explained that these tools are needed to distinguish the key hydraulic mechanism(s) of a persistent pool. The consequence of not presenting the necessary applicability of the framework in the introduction, is that the application appears as an isolated part in the report. Since the aim of the paper is to understand the supporting mechanisms to improve effective water resource management (lines 47-48), it should be made clear in the introduction that the framework needs to be applicable to real world situations to identify the pools' supporting mechanism(s) and with that knowledge determine its susceptibility.

*Thanks for the suggestion. It seemed self-evident that the framework and methods needed to be applied in the real world. But if there is value in explicitly stating this then it is a minor correction and we can easily do so.*

(2)     In section 5 the framework is applied to the Hamersley Basin, to see if the suite of tools can be used to identify the key hydraulic mechanism supporting a persistent pool. To show this, a subset of 22 pools is said to be investigated. However, only data of three of these is presented as case study. In

the three case studies the data is explained and used to establish the dominant hydrologic mechanism supporting the pool. The 19 other pools are only mentioned at page 25 to make a general distribution of the mechanisms supporting pool persistence across the landscape. This is a problem, because no hard data is provided to prove that the five dominant mechanisms are all present in the Hamersley Basin, as stated in Figure 5. Without this data, the reader cannot examine the measurement results and check the conclusions drawn by the authors. Moreover, the generalised explanation of the locations of the different type of pools appears as a repetition of section 2, mainly because no results are given to prove the statements made. As a result, the application does not convince the reader that the suite of tools can be used to distinguish all five different hydraulic mechanisms that support persistent pools. My recommendation is to expand the case studies with the data of the other 19 pools to at least show the five different supporting mechanisms. If similar data is present for different pools they can be compared and only then conclusions on the general location in the landscape can be drawn.

*Figures 5 and 6 were intended to familiarize the reader with the study area, which is likely to be a part of the world that most reader's will not have experience first-hand. It is not our intention to present empirical data to show that all of these mechanisms operate. Rather, we felt that a set of photos that span the types of pools present would be helpful for the reader to orient themselves to the landscape. Similarly, it is not feasible to present the detailed data from all 19 pools, the map is presented to give the reader a sense of the abundance of persistent pools within the study area. We can delete these elements if they are not of value to the reader, but we do not feel that presenting more empirical data is warranted.*

(3)     In the conclusion, the authors summarise the main hydraulic mechanisms supporting persistent pools and indicate the susceptibility to hydrological changes. However, the results of the application of the framework to the Hamersley Basin are not mentioned. Subsequently, as the conclusion is written now it focusses on the framework itself presenting it as being purely theoretical, while the application proves it to be practical, as well. Line 784 states that the presented suite of tools makes it possible to apply the framework to the real world. The authors explain the application of the framework in section 5, but do not mention this in the conclusion. The fact that the suite of tools can be applied to identify the different mechanisms, shows the value of the framework to determine the susceptibility which is a steppingstone to improve effective water resource management. Because of this, it should be made clear in the conclusion that the constructed framework and suite of tools have been shown to be applicable to real world situations.

*No problem, we can add a sentence to this effect during revision.*

In addition, the following issues should be considered.
- Section 2.3 divides groundwater influenced pools into two broad categories, however it is not specified based on what this decision made. Springer and Stevens (2009), as also referenced in this section, present twelve type of springs, but this already existing classification is not used in this new framework. So, please provide an explanation why this decision is made.

*We believe that this is already described in sufficient detail at the beginning of Section 2.3*

- In section 3.3 various tools are mentioned, but the methods of these tools are not given. Hamilton et al. (2005) and Siebers et al. (2016), for example, clearly explain the methods for $^{18}O$ isotope measurements and how to interpret the results. They also mention the locations of measurements, including the requirement to measure the isotopic composition in alluvial groundwater, which is not mentioned in the paper. Thus, please also provide the

methods of the tools in section 3.3, including how to execute the measurements, which instruments to use, and which frequency is necessary for reliable results.

*Thank you for the suggestion. Additional references to specific methods will be added during revision where appropriate.*

- The third paragraph of the case study of the Plunge Pool gives the conclusion, but it is not well substantiated. Why is the mechanism explained in section 2.3.1 not the key mechanisms in this case? Please give a better explanation based on what this conclusion was drawn. Actually indicating the position of the dykes in Figure 8c would assist in this explanation.

*We will revisit this for clarity during revision*

- Section 3.2 extensively explains different water balances per supporting mechanism, but these balances are not specifically mentioned or used in the application of the framework. Please use the water balance in the application or leave it out of section 3.

*It is impractical, and not our intention, to demonstrate all of the methods described in Section 3 in the case studies presented in Section 5. We will clarify this in the revised manuscript.*

- The second paragraph of section 2.3.2 explains the difference between the two mechanisms; topography intersects regional aquifer and groundwater outflow at geological contacts. Please expand this explanation, as the difference is not clear to me now.

*We will revisit this for clarity during revision*

- The sentence "In the… aquifer (900 mS/cm)." in lines 602-604 on page 29 can be interpreted in two ways. Because the rainfall is not given in Figure 8b, "in the absence of rainfall" can reference to either a decreasing pool depth, and in these periods the EC increases (notequilibrating around groundwater EC), OR to the moment shortly after a high pool depth is reached when the EC does gives values of approximately the groundwater EC. Please rewrite this sentence that it can be interpreted in only one way.

*We will revisit this for clarity during revision*

- The third paragraph of section 5.2.3 indicates measurements were done at the "top" of pool 1. Does this mean the upper part of the pool or the upstream part? Please elaborate if the measurements for the Radon-222 activity were done with a vertical or horizontal transect, or a combination.

*We will revisit this for clarity during revision*

*The following minor issues will be addressed during revision.*

List of minor issues:
- Page 5, lines 125-127 contain the sentence "For example…many pools". Please rewrite this sentence as it is lengthy and not formulated clearly.
- Page 14, line 299 mentions snapshot sampling. Please give a short explanation of what is meant with this.

- At page 15 the abbreviations NDVI, NDWI, and AEM are used. Please also give the full meaning of these abbreviations.
- Page 18, lines 404-407 contain the sentence "For example…flood event." Please rewrite this sentence as it is missing a verb and not formulated clearly. Also, put "value" before "re-equilibrated with the groundwater EC" instead of after it.
- Page 25, line 535 mentions the "mode of occurrence" of pools. Please specify what is meant with this, as I could not come up with a definition from the context.
- Page 27, Figure 6 gives the location of the persistent pools in the study area and the 22 pools chosen as subset, but it does not give the location of the three case studies specifically, which I think would be an addition to the map. Also, I think a digital elevation model (DEM) of the study area would be an addition since the topography of the area can determine the existence and persistence of pools. So, please indicate the location of the case studies on the map and add a DEM.
- Page 28, lines 586-587 mention the Marra Mamba Formation, but this subsurface layer is not indicated in Figure 8c. Please change the legend of Figure 8c to contain this name or change the name in the text to one of the layers that is mentioned in the legend. It could also be helpful to mention the hydraulic conductivity of the different formations, to give an indication of their permeability and the presence of aquicludes.
- Page 29 mentions the ADH. Please first give the meaning of this abbreviation, as this indication of a reference level is probably not known to non-Australians.
- Page 29, lines 592-595 mention the measured groundwater level at two monitoring wells. As their number is given in Figure 8c, it may be useful for the interpretation if the number is also mentioned in the text.
- Page 31, line 634 mentions the abbreviation BIF. Please give an explanation of this abbreviation or write it in full, since it is not used as abbreviation afterwards.
- Page 32, lines 658-659 mention "the water table drops below the pool" which references to Figure 8c, but the groundwater level is not given in this figure and cannot easily be compared to the water table in Figure 8b because of a different time scale. Please remove this part of the sentence as it cannot be checked now or add a graph of the water table to Figure 8c.
- Page 32, lines 673-675 state that a threshold groundwater level for inflow of groundwater to the pool can be found from the isotopic data. This cannot be concluded from the explanation of Figure 9d, as explained in the third paragraph in section 5.2.2. Please explain how this threshold value was found.

- Page 35, lines 710-711 states that pool 1 dries out as the dry season progressed, but the deeper parts of pool 1 persisted throughout the dry season. Please rewrite this sentence to resolve the contradiction for pool 1.

List of technical comments:
- Page 4, line 79: "mechanisms" instead of "mechanism"
- Page 10, line 198: "mechanisms" instead of "mechanism"
- Page 10, line 200: "purpose" instead of "purposes"
- Page 10, line 202: "Springer and Stevens (2009)" instead of "Springer (2009)"
- Page 15, line 315: add "as" before "a"
- Page 21, line 471: "may" instead of "my"
- Page 21, line 471: insert "the" before "subsurface"
- Page 29, line 598: move reference to Figure 8b to line 601 after "catchment"
- Page 29, line 601: add "period" after "dry"
- Page 29, line 603: substitute "that of the" instead of "the that of"
- Page 30, Figure 8c: What does "mRL" mean?
- Page 31, lines 650-651: move the closing bracket after "12$^{th}$ Nov 2018" to after "7$^{th}$ Dec 2018"
- Page 32, lines 665-672: move the sentence "Based on … (Fig. 9e)." to the end of the paragraph, so after the sentence "As a result ... this pool."
- Page 34, line 690: move the position of the reference of Figure 10a to after "Marra Mamba Formation" in line 693
- Page 37, Figure 10e: add a legend for the orange subsurface layer

References

Hamilton, S. K., Bunn, S. E., Thoms, M. C., & Marshall, J. C. (2005). Persistence of aquatic refugia between flow pulses in a dryland river system (Cooper Creek, Australia). *Limnology and Oceanography*, *50*(3), 743-754.

Siebers, A. R., Pettit, N. E., Skrzypek, G., Fellman, J. B., Dogramaci, S., & Grierson, P. F. (2016). Alluvial ground water influences dissolved organic matter biogeochemistry of pools within intermittent dryland streams. *Freshwater Biology*, *61*(8), 1228-1241.

Springer, A. E., & Stevens, L. E. (2009). Spheres of discharge of springs. *Hydrogeology Journal*, *17*(1), 83-93.

---

## Referee Report (RR1)

The revised version of the manuscript "A hydrological framework for persistent pools along non- perennial rivers", by Sarah A. Bourke *et al*. clearly improves the previous versions.

There are still a number of small issues that should be improved before publication.

Line 95: "... where the shallow, unconfined aquifer does not support year-round flow" this is unclear and does not apply to all the types of persistent pools.

Lines 96-98: "the general case of a non-perennial river along an alluvial channel (inundated and/or flowing during contemporary flood events) within valley-fill sediments deposited over bedrock" this excludes the perched pools directly carved in impermeable bedrock.

Table 1:

- Hydrochemical characteristics of perched water: enrichment in nutrients such as nitrogen, phosphorus, and dissolved organic matter, which are attributed to both the concentration by water evaporation and the accumulation of leaves and other types of organic matters, may contribute to the eutrophication of the pool water.

- Susceptibility to stressors of perched water: Drinking animals (cattle) or riparian vegetation transpiration can dry out the pool

- Caption: The meaning of the variables in the equations is not indicated here. Transpiration by the riparian vegetation should be taken into account or mentioned, particularly where the extent of this vegetation is large in comparison with the pool area.

Line 114: "if the pool is directly carved in the impervious bedrock or there is a low-permeability layer..."

Line 149: The regional gradient of the river (from the headwaters to the catchment outlet) is not relevant for the pool hydraulics.

Line 383: Transpiration by the riparian vegetation should be taken into account or mentioned, particularly where the extent of this vegetation is large in comparison with the pool area.

Line 387: conversion of water levels to both pool water volume and area requires knowledge of pool bathymetry.

Line 388: between $h_p$, A and V will change during...

Line 449: Evaporation is the only output that involves isotopic fractionation; transpiration and outflow do not modify the isotopy of the pool water.

Line 510: Some short description of the geology of these reliefs (Ranges, spurs and hillslopes) would be necessary here.

Line 573: are these perched pools?

Line 591: Figure 10a

Line 592 Figure 10b

Line 617: Figure 11a

Line 618: This topography and the section on figure 11 suggest that this pool could also be considered a 'topographic low' one.

Line 629: Figure 11b

Line 645: Figure 11c

Line 650: may through-flow from alluvium in the upper part of the water fall seep along it?

Figure 11a: The horizontal scale of the graph is lacking.

Line 666: What is BIF? Brockman Iron Formation?

Line 672: Channel scour physically depends on gradient and depth of the flowing water.

Line 677: Figure 12b

Line 689: Figure 12c

Line 693: "... isotopic values began to enrich suggesting that evaporation became less compensated by decreased inputs from groundwater..."

Line 694: Figure 12d

Line 696: Figure 12e

Figure 11: This should be Figure 12. Dots are too small in the legends of figures b) and c)

Line 744: evaporation and riparian vegetation transpiration rates.

Figure 13: This example is insufficiently explained and it is not necessary for the paper. $m^3 d^{-1}$ are not volume but flow units.

Figure 14: This figure is inconsistent and redundant. The 'Topographically... pool' is too small for observing the structure. The 'Throughflow... pool' looks really as a 'Topographically controlled...' pool.

---

## Author Response (AR2)

**Major revisions to HESS-461-2021: A hydrological framework for persistent pools along non-perennial rivers, Bourke et al.**

Dear editor,

Thank you for the opportunity to revise this manuscript. We have made substantial revisions to the manuscript in response to the reviewers' comments. The comments from reviewers and community members were largely consistent, and we are hopeful that this revised manuscript will have addressed their key concerns. Although we haven't explicitly responded to their comments here, the community (students) reviews were particularly helpful and we have implemented their suggestions consistent with our response during the discussion phase. These reviews provided thoughtful and constructive suggestions for how to improve clarity of organisation – we thank them for taking the time, and hope that they receive similarly useful reviews on their own work in the future.

We hope that the manuscript is now suitable for publication (minor revisions not withstanding). In addition to the conceptual framework this paper was originally conceived as, this manuscript now includes an extensive review of literature and field methods relevant to the hydrology of persistent river pools, as well as a substantial demonstration of the application of this framework in the Hammersley Basin using both regional-scale and pool-scale techniques (comprising a substantial novel-data component). While reviewers may pick out small deficiencies in individual components, we believe that this now forms a comprehensive (and long) manuscript that presents a framework of ideas and their application with links to find more information and further considerations on each sub-component that will continue to be useful for the scientific community.

**Summary of revisions**

**Manuscript structure and integration of sections**

The order of sections has been revised in-line with the reviewer's recommendation so that the management implications with respect to susceptibility of pools is now before the description of available tools for identifying hydraulic mechanisms.

The description and critique of available tools is now separated into regional-scale and pool-scale tools in acknowledgement that detailed sampling at specific pools is not always possible or required depending on needs of a given study or assessment.

We have also added a separate discussion section after the case study as suggested.

The case study section itself (Section 5) now includes a section explicitly describing the application of regional-scale tools to Hammersley Basin region, followed by the three pool-scale case studies. This division is between regional- and pool-scale tools is consistent with the revised section on available tools (Section 4).

**Manuscript Text**

A substantial portion of the text has been updated in line with specific comments made by the reviewers (and community members). These changes have focussed on improving clarity and accuracy, as well as adding citations where suggested.

The aim and objectives described in the Introduction have also been updated to reflect the new structure of the manuscript.

The manuscript now refers to m asl (above sea level) rather than the equivalent commonly used in Australia, m AHD.

A new discussion has been added which begins by summarising what has been achieved in the manuscript. It then discusses some additional considerations (e.g., that although we have striven to go back to the basic concepts, there may be more than one mechanism contributing to pool persistence) and suggests that multiple methods from Table 2 be used, demonstrating with the synthetic example how the interpretation of just one type of data may lead to an erroneous conclusion. We also acknowledge the difficulties of conducting fieldwork in the environments that host non-perennial rivers and how limitations of time and funding may call for trade-offs between detail and quantity in data. We then tie all these considerations back to what we learned from the case studies, thus linking the paper together more fully as requested by the reviewers.

The conclusion has also been substantially revised to better reflect the key outcomes of the paper, consistent with the revised objectives in the Introduction.

**Figures and Tables**

All of the figures in the manuscript have been revised in line with the reviewers' comments to improve clarity and consistency and we have also created new figures and tables.

The conceptual diagrams in Fig 1-4 have been updated to have alluvium labelled rather than alluvial channel, as suggested. We have also used consistent colours for geological layers and clarified impermeably vs permeable basement/bedrock as well as improving the placement and formatting of arrows and surface drainage features.

The map of the Hammersley Basin (Fig 5) has also been updated in line with reviewer comments, and there is now a map showing the locations of pools relative to geological strata (Figure 7), which also includes an inset of NDVI results. The previous figure of photos showing the different types of pools has now been unpacked (see Figs 6, 8, 9) and is accompanied by text describing the regional-scale assessment in a structure that links to the regional-scale tools described in section 4.

We have also moved Table 1 further up the manuscript and created a new table (Table 2) summarizing the available tools so that the reader can rapidly assess which they might find useful for a given study, as suggested. We have also added a table (Table 3) of hydraulic conductivities of the geological formations described in the case study section (as requested).

Following reviewer comments questioning the importance of the water balance equations, we have now included text and a figure (Figure 13) in the discussion on the importance of an accurate understanding of pool water balance components when interpreting measured data.

**Additional comments**

We considered all of the reviewers' comments and suggestions thoroughly; there were a small minority that we have not made changes in response to:

- We believe that the word "framework", defined as a hypothetical description of a complex entity or process, accurately describes the nature of this manuscript and have therefore not altered the title and continue to use this word in the manuscript.
- The hydrological framework presented in Table 1 was developed primarily from first principles. As such we believe that it is most appropriate to retain the references provided within the accompanying text, rather than adding references to the table.

**HESS-2021-461: Author changes in response to Reviewer 1**

The manuscript entitled 'A hydrological framework for persistent river pools' by Sarah A. Bourke et al., propose a paper that describes a framework for characterizing the hydrology of semi-permanent river pools, as well as some examples of this kind of pools. Althoug I find interesting the overall idea of the manuscript, it is not adequate for publication in its present form.

*Thank you for taking the time to review our manuscript, we appreciate your constructive comments and have done our best to improve the manuscript in response to your comments.*

The description of the 'framework' (section 2) is rather overconfident, as this is more a revision of former descriptions than an original one.

*While we agree with the reviewer that there are a small number of papers that mention some of the hydraulic mechanisms that sustain persistent river pools, these often have an ecological or management focus, and the treatment of hydrology is incomplete, flawed, or cites this manuscript under review. Therefore, there remains a need for this manuscript to be published as a rigorous hydrological synthesis of these different mechanisms so that future studies can be conducted in the context of a robust hydrological framework. Individual papers mentioned are discussed in more detail in our response to the comments on the pdf below.*

*We have considered whether the word "framework" is suitable as a description of the work presented in this manuscript, and we believe that it is. A "framework" can be defined as "a hypothetical description of a complex entity or process". We believe that this accurately describes what we are presenting; the hydrologic mechanisms supporting persistent river pools are complex and we have endeavoured to provide a theoretical (or hypothetical) description of them. Table 1, which summarizes our framework, was developed largely from first principles, in the context of our understanding of existing literature and our own experience working on persistent river pools. We therefore continue to use this word in the title and throughout the manuscript.*

*We have also considered what adjustments could be made to the Introduction given the reviewers comment. We agree with the reviewer that many of the elements described in the framework are mentioned within published literature if one knows where to look. We are not asserting that we are the first people to identify that water can persist in pools that form over impermeable rock. However, as described above, there is no complete, hydrologically robust, synthesis of the key hydraulic mechanisms that support the persistence of river pools, which is what we have aimed to achieve within this manuscript. We have made some adjustments to the Introduction to try to ensure that this point is clear to the reader (L63-85).*

Sections 2, 3 and 4 are is too descriptive, too long and repetitive, the equations are obvious and the figures are of poor quality. Most of this part could be synthesized in the table 1 with appropriate references and some auxiliary text like that in section 5.1.

*We are glad that the reviewer finds Table 1 useful. While the water balance equations may appear obvious, existing literature does not adequately or robustly describe the water balance of river pools*

*(see discussion above), and so we feel that it is important that these are explicitly presented and explained so that water balances can be accurately accounted for in future studies. Our work, including data shown in the Case Study Section 5, demonstrates with measured data the spatiotemporally variable nature of pool water balance components. We believe that the importance of comprehensively identifying the components of persistent pool water balances is an important message from our paper. As such, we have retained the water balance equations but changed them equations to in-line so that there is less emphasis placed on them. The importance of understanding spatiotemporal variability is also now discussed within a new Discussion section, which includes (Figure 13) the isotopic modelling results below (see comments on pdf) to demonstrate our point more clearly to the reader.*

*Similarly, hydrologic concepts that we may take for granted are often used or interpreted differently by practitioners in related fields. The reviewers' comment that the hyporheic zone as an ecotone or habitat relevant for aquatic life provides a great example of this. While this is true from an ecological perspective (Stubbington, 2012), there is also an extensive subset of hydrology related to the hyporheic zone that focusses not on ecological properties, but on the scales and mechanisms of hydraulic fluxes, which are driven by streamflow and channel morphology (e.g. Stonedahl et al., 2010, Bourke et al., 2014). Thus, when the stream is not flowing, these in-and-out hyporheic exchange fluxes are not operating. In this manuscript (and others, e.g. Leibowitz and Brooks 2008), alluvial water is treated hydraulically as a groundwater storage, with fluxes from the capture zone into the pool considered groundwater inflow, and outflow via infiltration (to the release zone) back into the alluvial groundwater. These fluxes are driven by the hydraulic gradients between the pool and the alluvial groundwater and are not related to streamflow. Conceptually, this hydraulic exchange is most accurately described as analogous to the well-established concept of through-flow lakes found in literature on surface water – groundwater interaction (Winter et al., 1998). While this surface water – groundwater exchange process seems clearly distinct from the relatively short-time scale fluxes of hyporheic exchange associated with streambed contours, we can see that the distinction between this and longer timescale parafluvial flows may be unclear, particularly for non-perennial streams (Del Vecchia et al., 2022). Section 2.2 has now been substantially revised to clarify how the conceptual models of hyporheic exchange and through-flow lakes can be applied to persistent river pools (S2.2, L161-213).*

*With regard to Figures 1-4 in Section 5.1, these are presented as generalized conceptual diagrams of the hydraulic mechanisms that can support persistent river pools. Although we want to be geologically and geomorphologically plausible, they are not intended to represent particular settings or landscapes and are not to scale. This is now clearly stated in the revised manuscript (L110-112).*

*These figures were always intended to be non-site-specific conceptual diagrams (even in the first submission of this manuscript they did not represent the settings of specific pools), and are broadly consistent with other published diagrams of incised river valleys and floodplains (e.g. Hayes et al. 2018). As this manuscript has a hydrological focus, we have drawn these figures so as to allow us to demonstrate the hydraulic processes that we are discussing, consistent with our experience of, primarily, Australia, but also North America. In determining the geometries and labels used in these figures we consulted with colleagues who specialize in geology and geomorphology. We received a range of responses, from which we chose those that we thought were simplest, and most effective at conveying the hydraulic processes we were describing to a broad audience (for which this paper is intended).*

In my opinion, section 5.2 is of value and deserve publication if some aspects are improved. Mostly, the paper should be readable for everybody not used with Australian geologic units, map coordinates and elevation datum.

*This section was added in response to reviewer comments on the previous submission of this work and we are pleased that this reviewer finds value in it. Presumably the reviewer would be more comfortable with elevations in meters above sea level (m asl), which is equivalent to m AHD (Australian Height Datum) that we had used. We have updated the revised manuscript accordingly and now use m asl for elevations.*

*It is not our intention to assume that the reader is already aware of the Hammersley Basin. As such, the geology of the Hammersley Basin is complex, but is described generally at the beginning of Section 5 (L546-556). We have also now added a new table describing the hydraulic conductivities of key units (Table 3). Regionally, the Hammersley Basin is a fractured rock province that is not considered to host productive aquifers for water supply (Jacobsen & Lau, 1987), so we have not delineated aquifers and aquitards.*

The map in Figure 6 should represent more information than just the location of unknown pools and the figures should be of better quality.
*Thank you for the useful suggestions. This map (now Figure 5) has been substantially revised to show the locations of towns (Pannawonica, Wittenoom, Tom Price, Paraburdoo, and Newman), as well as the boundaries of surface water catchments. The grid coordinates shown are standard UTM values for Zone 50K, which are used by Google Earth. A statement of this and a north arrow (up) has also been added to the revised figure (S5, Figure 5). The pools used as case studies are also now clearly identified.*

The assumptions and interpretations should be better separated from observations.
*Each of these case studies is currently structured as beginning with a description of the hydro(geo)logical setting, followed by the data collected and the resulting interpretation of mechanisms supporting pool persistence, and finally the implications for management. The reviewer has not provided any specific guidance on how to improve the structure, but it seems that the existing structure does separate observations (data) from interpretations. No change made*

Section 6 is rather a discussion than a conclusion, but some discussion is necessary not for showing the interest of 'framework' but for identifying research gaps and further research goals, not necesarily using heavy instrumentation.
*There is now a separate Discussion section and the Conclusion has been thoroughly revised and rewritten. We acknowledge that regional-scale assessments of hydraulic mechanisms supporting pools are possible without detailed, pool-specific data; thank you for pointing out that this was not clear from the previously submitted manuscript. We have now separated the description of available diagnostic tools into (i) regional scale (S4.1), and (ii) pool-scale tools (S4.2). Similarly, the case study now begins by demonstrating the application of regional-scale tools in the Hammersley Basin (S5.1), followed by three pool-scale case studies that incorporate water level and hydrochemical time-series data (S5.2).*

Many detailed comments are annotated in the manuscript. Please also note the supplement to this comment: https://hess.copernicus.org/preprints/hess-2021-461/hess-2021-461-RC1-supplement.pdf

*Further response to individual comments in the supplement provided is as follows. Where a number of comments were made on one page or section these are addressed collectively as appropriate.*

**Comments Reviewer 1 made on the pdf**

P1 This is not new, but an update of already described schemes (e.g. Fellman et al. 2011; Bonada et al., 2020). Also Jocque, M.; Vanschoenwinkel, B.; Brendonck, L. Freshwater rock pools: A review of habitat characteristics, faunal diversity and conservation value. Freshw. Biol. **2010**, 55, 1587–1602.

*While there are publications that describe some aspects persistent pools, we do not believe that there is an existing publication that presents a comprehensive scheme or framework for understanding the hydraulic mechanisms supporting persistent river pools. Specific elements of our framework that are not clearly articulated or present within in existing literature include:*
- *the applicability of the theoretical model of through-flow lakes to persistent river pools*
- *the role of catchment constriction in determining the location and persistence of river pools*
- *the need for an impermeable layer for persistent pools in the absence of a connection to groundwater or alluvial through-flow*
- *review and critique of available tools for characterizing the hydrology of persistent pools*
- *demonstration of regional-scale tools for characterizing the hydrology of persistent pools*
- *time-series data demonstrating spatial and temporal variability in pool hydrochemistry and water levels,*
- *the application of timeseries data to characterize the hydrology of persistent pools*

*That said, we thank the reviewer for introducing us to the Joque et al. (2010) paper freshwater on rock pools that we had not previously cited and we have added this paper to the Introduction (L79) and the section on perched pools (2.1, L125). As the title suggests, this paper describes the ecology of freshwater that persists over impermeable hard rock. There is a brief hydrological description (1 paragraph of hydrology, 1 paragraph of examples) within the section on "the rock pool habitat: definition and distribution". In the first paragraph authors mention that these features can be filled by precipitation, rivers and groundwater, but that the paper focusses on rain-fed rock pools which are the more typical freshwater habitat (presumably perched pools over impermeable bedrock). Thus, while identifying a relatively broad range of hydrological features that can exist (some of which may be within river channels, others which are not – gnammas for example), it does not detail the hydrological mechanisms that can support persistence of water in pools along rivers (groundwater discharge vs perched rainwater), which is the main focus of our present manuscript.*

*The reviewer also refers to Bonada et al. (2020), which is a paper on conservation and management of isolated pools in temporary rivers that we are aware of (and had cited). While this paper does provide a brief summary of hydrologic mechanisms that can support pools that is more rigorous than Joque et al. (2010), it cites the earlier version of our paper in HESS-D when doing so. As such, it is a circular argument to say that we are duplicating the work of Bonada (2020) given that they have applied the framework presented here in their manuscript. In revisiting the Bonada paper in response to this review, we have realized that we have not cited Leibowitz and Brooks (2008) chapter on vernal pools and will correct this omission in the revised manuscript; this citation has now been added in the revised manuscript (S2.2 L157). This 2008 book chapter provides a summary of the water balance of*

*pools that are not perched, which is consistent with the framework presented herein, but does not describe subsurface permeability features that control groundwater discharge.*

*The reviewer also directs us to Fellman et al., (2011), which we discussed in the early versions of this manuscript. This paper aims to characterize the hydrology of a particular set of pools as controls on dissolved organic matter biogeochemistry. While this manuscript does describe perched and alluvial through-flow pools along river channels, it does not robustly describe the hydrology of these features. It draws conclusions about the hydraulic mechanisms supporting pools based solely on stable isotope values of water (beginning and late dry-season), which are subject to uncertainty that has not been described. In their paper, the water balance of the pools is assumed to consist of inputs from rainfall and groundwater inflow and losses to evaporation. The calculation of evaporative loss from stable isotopic enrichment was made on the basis of a steady state model of evaporation divided by input (E/I). Perched and alluvial through-flow pools are then identified using this ratio (high E/I ratio implies perching). As such, although a subset of the pools are identified as through-flow pools, the conceptual model that underpins the analysis does not account for outflow of water from the pool back into the alluvium (Liebowitz and Brooks, 2008).*

*The stable isotopic enrichment of a pool with an initial volume of 400 $m^3$ can be simulated using the water balance equations presented in the current manuscript under review (Figure 1 below - now Figure 13 in the paper). The evolution of stable isotopic values is simulated using the approach of Bourke et al., (2021). A perched pool will have no inflow during the dry season and losses to ET only; a through-flow pool will have losses to ET, inflow of alluvial groundwater and loss via outflow (infiltration) of pool water back into the alluvium (ET + GW inflow + Outflow). For a perched pool with a volume of 400 $m^3$ at the beginning of the dry season, water volume over 112 days will reduce to 178 $m^3$ with $\delta^{18}O$ enriching from -8 to 3.5 ‰. The addition of a groundwater inflow of 0.0002 $m^3$/min (0.3 $m^3$/d) results in similar end-point values (210 $m^3$ and $\delta^{18}O$ of 2.4 ‰). In this example, using the line of thought presented in their paper, Fellman et al. would have concluded that groundwater in this second pool is not an important component of the water balance. However, over 112 days this groundwater inflow equates to 8% of the initial volume of the pool and may be important for hydrochemical parameters in the alluvial water (or regional groundwater) that have different values than the pool water. Furthermore, alluvial through-flow pools will usually have water losses associated with infiltration to the streambed sediments, which Fellman did not account for. Thus, the inflow of groundwater may be larger than otherwise thought, if it is balanced by infiltration from the pool of a similar magnitude. For example, groundwater inflow of 0.0008 $m^3$/min balanced by outflow via infiltration of 0.0006 $m^3$/min will result in the same pool water level as a groundwater inflow of 0.0002 $m^3$/min, but the isotopic enrichment will be slightly smaller ($\delta^{18}O$ of 1.3 ‰). Over 112 days, the groundwater inflow in this third scenario adds up to 128 $m^3$, or 32% of the initial pool volume. A fourth scenario where the water balance is consistent with Fellman (ET and GW inflow terms are as per scenario 2), but the pool area is halved (initial volume remains the same) demonstrates that the water balance of the pool and stable isotopic enrichment are sensitive to the pool geometry (volume to area ratio), which Fellman et al. did not report on or consider explicitly in their analysis. Thus, the identification of hydraulic mechanism supporting pools was made on the basis of unsupported assumptions about pool water balances.*

*Their analysis approach, based on an incomplete water balance, led Fellman to conclude that many of the pools studied were isolated from the alluvium water table, but this conceptualization (see their*

*Fig 1) is not hydrogeologically robust. All but one of the pools in their paper occur on permeable alluvial sediments with pools 1-12 shown overlying a similar thickness of alluvium. If the pools were not connected to the alluvial (and/or regional) water table, without the presence of a low-permeability layer beneath these pools, the pool water would infiltrate into the alluvium (Brunner et al., 2009) and the pool would not persist. Thus, the inset diagram of "pools isolated from alluvium water table" is hydraulically implausible (as previously discussed in the manuscript).*

[Figure]

*Figure 1 Evolution of pool volume and values of stable isotopes of water in pools with varying water balance components over approximately 4 months of dry season (ET = evapotranspiration, GW = groundwater, A = pool area). Model modified after (Bourke et al., 2021)*

P5 Deep perched pools may persist without groundwater flow, ?,

*This text has now been modified and moved to the Discussion section.*

This kind of pools was earlier described, as e.g. Fellman et al., **2011**, 47, W06501.

*We agree that Fellman does mention perched pools and we have cited and discussed it in this section. However, for the reasons described above in detail we do not consider this to be a robust or complete hydrological discussion of the key elements of perched surface water as they relate to persistent river pools. The literature on disconnection of surface water and groundwater is also important here (e.g. Brunner et al., 2009), and therefore in our manuscript we combine these elements (as well as Joque 2010) in our discussion of perched surface water.*

The negative role of riparian vegetation transpiration may be more important than its shading effect.

*We agree that evapotranspiration is an important element of the pools water balance, and already state this in that same sentence. This paragraph is outlining the theoretical considerations, and we do not feel that it is appropriate at this point in the manuscript to make an assertion that either shading or ET loss will be more or less important in controlling pool water balances – this will depend on the characteristics of individual pools and the types of vegetation they support. No change made.*

P6 Is this necessary?

*The reviewer refers here to identifying a low-conductivity layer to support perching in sandy sediments. A low-conductivity layer is essential for supporting perched surface water in otherwise permeable sediments (Brunner et al., 2009). We are not saying definitively that it wasn't there, but we do think this is an important feature of perched surface water within permeable sediments (as opposed to overlying*

*impermeable bedrock) that was not acknowledged or discussed by Fellman et al. (2011). No change made.*

Bedrock is assumed of low permeability but this is not stated (otherwise the upper pool could not exist).

*We thank the reviewer for highlighting the inconsistency in the implied permeability of the bedrock. We have updated Figs 1-4 to consistently differentiate between impermeable (or low perm) bedrock and permeable bedrock.*

The picture is too simplistic/naive. (exaggerated relief, disconnection between the main stream and the tributary, "valley fill" of poor geomorphic meaning).'Alluvial channel' should be 'alluvium' It seems that the authors tried to generalize the pictures of their early manuscript in HESSD, but 'Tertiary detritals' cannot be directly converted into 'valley fill' because the last one can be assumed as of Quaternary age and therefore having a form related to the recent and present-day alluvial landscape (flood plain with Pleistocene terraces)

*The reviewer has suggested "Alluvium" as a replacement for "Alluvial channel" – we are not sure of the basis for this suggestion but have made the replacement in Figs 1-4. Similarly, the lack of a defined surface drainage line between the hillslope and river in Fig 1 has been rectified.*

*We have used the term "valley-fill" to refer to any sediments within the geological river channel (as distinct from the flowing channel that a hydrologist may consider) and do not intend this to make any reference to a particular age of sedimentary deposition – hydraulically, the time of sediment deposition is not of primary importance. This is consistent with general definitions of valley-fill (unconsolidated sedimentary deposit which fills, or partly fills a valley) that do not refer to a specific time of deposition. A more suitable alternative has not been suggested by the reviewer and no supporting citations for this comment were provided so we are unable to determine a replacement term for the generalized conditions that we are trying to show in these conceptual diagrams. No change made.*

Water means here water saturating the alluvium.

P7 The occurrence of pools depend on much detailed scales (10 - 1000 m) whereas this concept of gradient is valid for 10-1000 kilometers.

Bedrock must also be of low permeability.

Evapotranspiration by riparian vegetation may deplete relevant volumes of water. Brooks, R. T., & Hayashi, M. (2002). Depth-area-volume and hydroperiod relationships of ephemeral (vernal) forest pools in southern New England. Wetlands, 22(2), 247-255.

P8 shallow

This lacks of scientific meaning. Hyporheic zone defines an ecotone or habitat relevant for aquatic life; the water and life in shallow groundwater in exchange with those in surface water. Hyporheos refers usually to the shallow groundwater in the alluvium.
https://en.wikipedia.org/wiki/Hyporheic_zone

*This section on through-flow of alluvial groundwater (S2.2) has been significantly modified to improve clarity. Thank you for drawing our attention to Brooks and Hayashi, which we had omitted when*

*discussing the importance of ET as part of the water balances of persistent river pools and groundwater throughout the manuscript. We have now cited it at multiple suitable points in S2.2.*

p9 These pictures are rather naive and of very poor geomorphic quality. The 'valley fill' should take the form of a flood plain and/or a terraced system and the water table should be not so far from the surface of the floodplain.

*Figures 1-4 are conceptual diagrams, which are not to scale. Figure 2b represents the case of a perched water table beneath a river that resides in an arid or semi-arid climate where the regional water table can be tens of metres below the surface (Villeneuve et al., 2015). The key feature of this diagram is that the regional water table is below the valley fill, within the weathered basement. In our view, this diagram reasonably represents a valley fill on the order of say 10-20 m and a depth to water table of 15-30 m, which are plausible values in our experience. No change made.*

P10 this discussion lacks of interest

*This text was added in response to reviewers of our previous submission of this manuscript who were of the view that the hydraulic processes relevant to persistent river pools were all covered in literature on groundwater springs. Given that the term "sphere of influence" is quite vague, we have unpacked this framework or schema further so that it is evident to the reader why it is not helpful for persistent river pools. We are pleased that the reviewer finds this self-evident, but we would be deleting the text to satisfy the current reviewer, at the expense of not addressing comments from previous reviewers. Thus, we suggest retaining the text. No change made.*

Some hard rocks may be permeable, such as  sandstones.

*Agreed. Sentence modified to refer to effective hydraulic conductivity rather than fracture aperture.*

P12 Again, the generalization of the figures from the former paper is not successful, particularly for the 'valley fill' and the lower boundary of the aquifer, that there was the lower boundary of the weathered rock

*See comments above regarding use of the term valley fill. The reviewer has also made this comment about the lower boundary of the aquifer in Fig 3b. This figure depicts the generalized case where valley fill sediments are relatively thin and the lower boundary of the regional aquifer is determined by the lower boundary of weathering in the bedrock, which is hydraulically connected to the valley fill. It is unfortunately not clear what issue the reviewer has with this depiction, which is consistent with our experience. No change made.*

P13 Unlike in the other figures, here 'bedrock' is permeable.

*Thank you for bringing the inconsistent implied permeability of bedrock to our attention, we have updated Figs 1-4 to ensure clarity and consistency.*

'Valley fill' fully lacks of sense in this setting.

*These unconsolidated sediments that are below and adjacent to the stream are now described as alluvium.*

None of the former aquifers was confined, so this adjective is unnecessary here.

*Deleted.*

P15 This is excessive, more adequate for a textbook than for a research paper

*Text deleted.*

P17 and transpiration if riparian vegetation is present (as frequently occurs)

*Evaporation changed to evapotranspiration. Transpiration was already mentioned but not explicitly in this term, thank you for bringing this to our attention.*

P19 and along the river path (Dogramaci, S., Firmani, G., Hedley, P., Skrzypek, G., & Grierson, P. F. (2015). Evaluating recharge to an ephemeral dryland stream using a hydraulic model and water, chloride and isotope mass balance. Journal of Hydrology, 521, 520-532).

*Thank you for the encouragement to self-cite; citation added and sentence updated.*

Unclear

*"Concentration" replaced by "strontium values".*

This discussion is not well founded here; Fellman et al (2011) seem to assume that bedrock is impervious, as the Authors do in Fig. 1.

*This comment was referring to pools that overly unconsolidated alluvium; the permeablity of bedrock is irrelevant. Nevertheless, this sentence has been deleted.*

P22 Phreatophytes can abstract relevant volumes of water from the alluvial aquifer. New tree plantations may have similar effects than pumping.

*True. We have replaced "abstraction" with the more general term "withdrawals" which may include by pumping or ET as stated at L332.*

P23 This table can substitute most of the previous text. But this is not original, so appropriate references are needed.

*The hydrological framework presented in Table 1 was developed primarily from first principles in the context of our understanding of published literature and our experience working on the hydrology of persistent river pools. As such we believe that it is most appropriate to retain the references provided within the accompanying text, rather than adding references to the table.*

Solutes and heavy water isotopes may increase downstream due to cumulated evaporation.

Of a pool? You must mean in the alluvium, yes of course. But here we are talking about the pool.

Phreatophyte stands or plantations may cause relevant water abstraction

In the absence of precipitation.

*Thank you for these helpful suggestions, they have all been added to Table 1.*

The units for the variables in the equations are welcome.

*The terms are defined in the text with units given.*

P24 Before explaining the 'valley fills', a description of the diverse bedrock (basement?) units is necessary, and a small table defining their names and hydrological properties would be very welcome.

More details are needed, as stated before

*The description of geology of the region has been updated in the text and a new summary table has been added that reports hydraulic conductivities (Table 3).*

P25 Which is the size of the total set?

New paragraph

shaded, so evaporation is minimised

Which percentage from the total amount of pools?

Bat? I expected aquatic life

How many? which percentage?

New paragraph

How many? which percentage?

*We recognize that this summary was not particularly useful for the reader. This text has been deleted and replaced with a new section on the application of regional-scale tools to understand hydraulic mechanisms supporting persistent river pools in the Hammersley Basin (S5.1).*

P27 As shown, this map is not useful to the reader and it should be deleted. If a location appearing in Google Earth is shown on the map, the visualization of satellite images of the area would be easier to the reader. Some identification of the pools must appear in the map; pool type or the names of the 3 pools in section5.2.  Coordinates should be adequate for a reader not used with the GDA94 ones.

*This map has now been updated to include towns, surface water catchment boundaries and identify the 3 pool-scale case studies. All spatial reference points are reported as projected coordinates so that linear distances are obvious from their values. These UTM projections are a global standard, and they are readily available within Google Earth.*

P28 The features at the pools are too small. The 'alluvial channel' is not visible in the figure, so the throughflow pool looks as a regional groundwater one. The relief seems very exaggerated. Hammersley Group and Fortescue Group are not described in the text. 'Valley fill' is too ambiguous concept. Why aquifers disappear in depth?

*This figure has been removed.*

Which is the altitude of the pool AHD?

*Pool water level elevations now reported in the text (data shown in Fig 10b)*

P29 Some adjective would be welcome: dykes of intrusive, or porphyritic rocks or...

*The dykes consist of dolerite, as previously described in the text and now also stated in Table 3.*

AHD (Australian Heigh Datum?) is not necessarily known to readers from other countries

*AHD has now been replaced with m asl (metres above sea level). These are effectively equivalent (https://www.ga.gov.au/scientific-topics/positioning-navigation/geodesy/datums-projections/australian-height-datum-ahd).*

Presumably a dyke (as assumed below)

*Now stated*

is attributed to

*text added*

although the occurrence of barriers (dikes) may force the groundwater to emerge (?)

*In this instance the dykes compartmentalize the aquifer, but because they don't continue up to surface they do not force the groundwater to emerge at surface.*

P30 Is this dyke the barrier cited above?

*Yes. 'nearest' added for clarity*

The water table between the two wells should not be shown as an horizontal solid line. The text about the pool depth should be moved to the text. The geologic units are shown in too similar colours and should be described in the text. V.E. should be fully stated as vertical scale exaggeration (?). The coordinates are not universal. Lettering font is too small.

*VE is now stated and annotations added to clarify permeability relationships. These x and y coordinates were UTM coordinates are a global standard that is available within Google Earth. Nonetheless, they were not required on this conceptual diagram and have been removed.*

P31 BIF?

*BIF is now defined as banded iron formation in Table 3.*

P32 All the information on the geological units and their permeability should be shown in a table, otherwise it is impossible to the reader to follow their relevance

*This is now done in Table 3, and we have also revised the figure annotations for clarity.*

P34 Issues like in other figures. Yellow letters are not legible, dashed blue line is not defined

*Yellow letters converted to black. Dashed blue line is water table, as now indicated by the standard inverted triangle symbol.*

P35 over

*Fixed*

values at the bottom of pool 1 are not shown

*'top' and 'bottom' replaced with up-stream end and down-stream end for clarity*

P36 Some less overconfident expression is suggested, e.g: These data allow us to infer …

*Done.*

Is this feasible??

*This is common in mining impacted areas like the Hammersley Basin. The removal of overburden and ore can substantially reduce area of surface catchments. This is now clarified in the text*

P37 c) Please, use a similar notation for both isotopes: 18O and 222Rd, or Oxygen-18 and Radon-222. What means Sub-pool? d) Lettering too small. Temperature scale would be better with the higher values on the upper part. e) Some lettering is too small. Alluvium GW flow arrow is located within the bedrock. What means 'Alluvial Bore? What is the orange colour below Wittenoom Fm?

*Sub-pool deleted, isotopic notation now consistent. Sub-fig e) revised.*

P38 Mols of this is rather a discussion than a conclusion

Summarizes

This is really overconfident!

Gather

Only in intricate places all this is necessary.

*We have now created a separate Discussion section and the Conclusion has been substantially revised. We acknowledge that some inferences about the hydraulic mechanisms supporting pools from regional-scale information. As such we have separated the methods into regional- and pool-scale tools and present a new section (5.1) applying the regional scale tools in the Hammersley Basin.*

**HESS-2021-461: Author changes in response to Reviewer 2**

Anonymous Referee #2
Referee comment on "A hydrological framework for persistent river pools" by Sarah A. Bourke et al., Hydrol. Earth Syst. Sci. Discuss.,
https://doi.org/10.5194/hess-2021-461-RC2, 2021

This paper contains a lot of information and is in a way a literature review with a proposed methodology for diagnostic and an application to study sites. I believe the contents are appropriate and people interested in the topic will find this a useful guide. Because of its nature, the paper has very little quantitative results, so the authors struggle to find a common synthesis or a final message.

I believe readers will benefit from a more succinct treatment in some of the sections. For example, sections 2 and 3 have lengthy introductory paragraphs that tell the reader what the authors are going to do next...I wonder if that is really necessary or if it is better to just mention what points are going to be touched upon and why. A better connection between sections 2,3 and 4 can be provided, with a bit of a synthesis and perhaps based on Table 1.

*Thank you for this suggestion, the extended text at the beginning of sections 2 and 3 has been deleted, edited or moved as appropriate. The link between sections 2-4 has also been enhanced by re-ordering the sections so that management implications are discussed before diagnostic tools. We have also re-structured the section on diagnostic tools (Section 3) to better align with the case studies (Section 5), in both cases clearly distinguishing between regional-scale tools and more data intensive local-scale approaches.*

Table 1 should be moved up the manuscript, within section 2.

*Table 1 was originally cited in the first sentence of Section 2. As such we have not moved the initial citation, but we have now placed the Table immediately before 2.1 so it is closer to the in-text citation.*

The end of the manuscript is rather abrupt. After going through the application sites the authors go straight to the conclusions, which is most of a summary. There is no discussion of differences or similitudes between sites, lessons learned or future work. I am sure that there are elements of all of this somewhere in the manuscript, but they need to be clearly synthesised at the end. This is, I believe, the main weakness of the manuscript.d

*A separate Discussion has now been included as Section 6 that discusses the similarities and/or differences across the case studies, summarizes the lessons learned from the field sites and suggest future work.*

*The Conclusion presented in Section 7 has been re-written to provide a more suitable concluding statement in line with the study aims.*

**References**

Bonada, N., Cañedo-Argüelles, M., Gallart, F., von Schiller, D., Fortuño, P., Latron, J., Llorens, P., Murria, C., Soria, M., Vinyoles, D., Cid, N. Conservation and management of isolated pools in temporary rivers, Water, 12 (10) 2870, https://doi.org/10.3390/w12102870, 2020.

Bourke, S.A., Cook, P.G., Shanafield, M., Dogramaci, S. and Clark, J.F., 2014. Characterisation of hyporheic exchange in a losing stream using radon-222. Journal of hydrology, 519, pp.94-105.

Bourke, S.A., Degens, B., Searle, J., Tayer, T., Rothery, J. Geological permeability controls streamflow generation in a remote, ungauged, semi-arid drainage system, Journal of Hydrology: Regional Studies, 38. 2021. https://doi.org/10.1016/j.ejrh.2021.100956

Brunner, P., Cook, P., and Simmons, C. Hydrogeologic controls on disconnection between surface water and groundwater, Water Resources Research, 45, W01422, 2009.

Del Vecchia, A, Shanafield, M., Zimmer, M., Datry, T., et al. Reconceptualizing the hyporheic zone of non-perennial rivers and streams. Submitted to Freshwater Science, June 2021.

Fellman, J. B., Dogramaci, S., Skrzypek, G., Dodson, W., and Grierson, P. F. Hydrologic control of dissolved organic matter biogeochemistry in pools of a subtropical dryland river, Water Resour. Res., 47, W06501, 10.1029/2010wr010275, 2011.

Hayes, Daniel & Braendle, Julia & Seliger, Carina & Zeiringer, Bernhard & Ferreira, Maria & Schmutz, Stefan. Advancing towards functional environmental flows for temperate floodplain rivers. Science of The Total Environment. 633. 1089–1104. 10.1016/j.scitotenv.2018.03.221. 2018.

Jacobson, G., Lau, J.E. 1987. Hydrogeology Map of Australia - 1:5m. Geoscience Australia, Canberra. http://pid.geoscience.gov.au/dataset/ga/15629

Jocque, M., Vanschoenwinkel, B. and Brendonck, L.U.C., 2010. Freshwater rock pools: a review of habitat characteristics, faunal diversity and conservation value. Freshwater Biology, 55(8), pp.1587-1602.

Leibowitz, S.G.; Brooks, R.T. Hydrology and landscape connectivity of vernal pools. In Science and Conservation of Vernal Pools in Northeastern North America; Calhouh, A.J.K., deMaynadier, P.G., Eds.; CRC Press: Boca Raton, FL, USA, 2008; pp. 31–53.

*Stonedahl, S.H., Harvey, J.W., Wörman, A., Salehin, M. and Packman, A.I., 2010. A multiscale model for integrating hyporheic exchange from ripples to meanders. Water Resources Research, 46(12).*

*Stubbington, R., The hyporheic zone as an invertebrate refuge; a review of variability in space, time, taxa and behaviour. Marine and Freshwater Research, 63, 294-311, 2012.*

*Villeneuve, S., Cook, P.G., Shanafield, M., Wood, C. White, N. Groundwater recharge via infiltration through an ephemeral riverbed, central Australia. Journal of Arid Environments, 117, 47-58, 2015.*

*Winter, T.C., Harvey,J.W., Franke, O.L., Alley, W.M. Groundwater and surface water: a single resource. USGS Circular 1139, 1998.*

---

## Author Response (AR3)

The revised version of the manuscript "A hydrological framework for persistent pools along non- perennial rivers", by Sarah A. Bourke *et al*. clearly improves the previous versions. There are still a number of small issues that should be improved before publication.

Thank you for taking the time to review or revised manuscript. We are pleased that you consider there to be only minor revisions required prior to publication. Note that the line numbers provided by the reviewer do not directly match with either the clean or marked-up versions of the revised manuscript. The reviewer also mentions Figure 14, which doesn't exist in the current draft. It seems the reviewer may have been looking at a superseded version. We have nonetheless done our best to address these comments and thereby improve the manuscript prior to publication.

Line 95: "... where the shallow, unconfined aquifer does not support year-round flow" this is unclear and does not apply to all the types of persistent pools.

Text removed.

Lines 96-98: "the general case of a non-perennial river along an alluvial channel (inundated and/or flowing during contemporary flood events) within valley-fill sediments deposited over bedrock" this excludes the perched pools directly carved in impermeable bedrock.

L100-101: Text added to account for perched pools over impermeable bedrock: …"where tributaries flow either across the bedrock or the valley-fill sediments"

Table 1:
- Hydrochemical characteristics of perched water: enrichment in nutrients such as nitrogen, phosphorus, and dissolved organic matter, which are attributed to both the concentration by water evaporation and the accumulation of leaves and other types of organic matters, may contribute to the eutrophication of the pool water.

Text to this effect added in Table 1 Row 1, Column 2, Hydrochemical characteristics of perched pools.

- Susceptibility to stressors of perched water: Drinking animals (cattle) or riparian vegetation transpiration can dry out the pool.

Text to this effect added in Table 1 Row 1, Column 3, Susceptibility to stressors of perched pools.

- Caption: The meaning of the variables in the equations is not indicated here. Transpiration by the riparian vegetation should be taken into account or mentioned, particularly where the extent of this vegetation is large in comparison with the pool area.

These variables are now defined in the text at the bottom below Table 1. *E* is the evapotraspiration rate, and therefore includes transpiration directly from the pool. Transpiration by riparian vegetation from the alluvium adjacent to the pool would be considered as a withdrawal from the alluvial groundwater store rather than a water loss from the pool directly. The possibility that this may need to be accounted for is now explicitly described in Section 4.2 L433-435.

Line 114: "if the pool is directly carved in the impervious bedrock or there is a low-permeability layer..."

Text updated.

Line 149: The regional gradient of the river (from the headwaters to the catchment outlet) is not relevant for the pool hydraulics.

It is not clear what the reviewer is referring to here, the regional hydraulic gradient is not explicitly mentioned in this version of the manuscript as controlling the pool water balance directly. There is mention of the recession of groundwater in the alluvium be controlled in part by hydraulic gradients at Line 172, which is accurate. No change made.

Line 383: Transpiration by the riparian vegetation should be taken into account or mentioned, particularly where the extent of this vegetation is large in comparison with the pool area.

As mentioned above, transpiration by extensive riparian vegetation constitutes a water loss from the groundwater store. The possibility that this may need to be accounted for is now explicitly described in Section 4.2 L428-429.

Line 387: conversion of water levels to both pool water volume and area requires knowledge of pool bathymetry.
Text added accordingly (L419)

Line 388: between $h_p$, A and V_will change during...
The relationship between $h_p$, A and V is controlled by bathymetry and doesn't change unless the bathymetry changes (for example due to a major flood event that redistributes sediments) The reviewer is likely referring to the role of $A$ in controlling the relationship between $h_p$ and V as water levels recede. The reduction in A as water levels recede is now explicitly identified at L421.

Line 449: Evaporation is the only output that involves isotopic fractionation; transpiration and outflow do not modify the isotopy of the pool water.
This is an unfortunate misconception. It is true that transpiration and outflow are not associated with fractionation, and the isotopic composition (or solute concentrations) of these loss terms will be the same as the pool itself at the time and location of water loss. However, these terms are important components of the pool water balance that need to be accounted for to avoid errors in the interpretation of isotopic data from pools. This point is described in detail in the discussion (Figure 13 and associated text).

Line 510: Some short description of the geology of these reliefs (Ranges, spurs and hillslopes) would be necessary here.
Text added at L547: …(consisting of bedrock and dykes described below)…

Line 573: are these perched pools?
It is unclear what the reviewer is referring to here at this line number (or in adjacent text). No change made.

Line 591: Figure 10a
Line 592 Figure 10b
Line 617: Figure 11a
It is not clear what issue the reviewer is attempting to identify here, these figure references appear to be accurate in the current version of the paper. No change made.

Line 618: This topography and the section on figure 11 suggest that this pool could also be considered a 'topographic low' one.

Figure 11 is describes Howie's Hole. The dominant hydraulic controls on the location of this pool are the contact between permeable and impermeable rock and the catchment constriction, as described. No change made.

Line 629: Figure 11b
Line 645: Figure 11c
This figure references appear correct, no change made

Line 650: may through-flow from alluvium in the upper part of the water fall seep along it?
It is not clear what the reviewer is referring to here. No change made.

Figure 11a: The horizontal scale of the graph is lacking.
Figure 11a is a photo. No change made.

Line 666: What is BIF? Brockman Iron Formation?

In the current draft, BIF is only mentioned in Table 3 where it is defined as Banded Iron Formation. No change made.

Line 672: Channel scour physically depends on gradient and depth of the flowing water.
The meaning of this comment is not entirely clear. Hydraulic gradients are important controls on scour because they control the flow rate. Flow rates can also increase if the area is restricted, as is the case where there is catchment constriction (which is what we are describing). The importance on enhanced flow rates for creating scour is described at L714. Text here updated to explicitly mention the increase in depth.

Line 677: Figure 12b
Line 689: Figure 12c
These figure references appear to be correct. No change made.

Line 693: "... isotopic values began to enrich suggesting that evaporation became less compensated by decreased inputs from groundwater..."
This suggested text is grammatically unclear. Existing text identifies the decreased inputs in groundwater, consistent with the reviewers suggestion. Evaporation didn't change and we see no reason to complicate the sentence by mentioning it. No change made.

Line 694: Figure 12d
Line 696: Figure 12e
These figure references appear to be correct. No change made.

Figure 11: This should be Figure 12. Dots are too small in the legends of figures b) and c)
Figures appear to be numbered correctly. Symbologies enlarged in Figs 11b and 11c.

Line 744: evaporation and riparian vegetation transpiration rates.
Meaning of comment unclear. May be referring to comments on Ben's Oasis at L786 – transpiration rates now explicitly mentioned in this sentence.

Figure 13: This example is insufficiently explained and it is not necessary for the paper. $m^3 d^{-1}$ are not volume but flow units.

As per our response to the reviewers comment above, the point here is that water balance components should be adequately accounted for to avoid misinterpretation of measured hydrochemical data. It is clear from the reviewer comments that our point did not come across with the text as written. The accompanying text has therefore been extensively revised for clarity and completeness (L905-931).

Volume units on Fig 31a have been corrected to $m^3$.

Figure 14: This figure is inconsistent and redundant. The 'Topographically... pool' is too small for observing the structure. The 'Throughflow... pool' looks really as a 'Topographically controlled...' pool.

There is no Figure 14 in the current version of the paper